# PHYSICS-REGULATED DEEP REINFORCEMENT LEARNING: INVARIANT EMBEDDINGS

**Hongpeng Cao**[1,*]**, Yanbing Mao**[2*,†]**, Lui Sha**[3] **& Marco Caccamo**[1]
[1]TUM, Germany, [2]WSU, United States, [3]UIUC, United States

## ABSTRACT

This paper proposes the Phy-DRL: a physics-regulated deep reinforcement learning (DRL) framework for safety-critical autonomous systems. The Phy-DRL has three distinguished invariant-embedding designs: i) residual action policy (i.e., integrating data-driven-DRL action policy and physics-model-based action policy), ii) automatically constructed safety-embedded reward, and iii) physics-model-guided neural network (NN) editing, including link editing and activation editing. Theoretically, the Phy-DRL exhibits 1) a mathematically provable safety guarantee and 2) strict compliance of critic and actor networks with physics knowledge about the action-value function and action policy. Finally, we evaluate the Phy-DRL on a cart-pole system and a quadruped robot. The experiments validate our theoretical results and demonstrate that Phy-DRL features guaranteed safety compared to purely data-driven DRL and solely model-based design while offering remarkably fewer learning parameters and fast training towards safety guarantee.

## 1 INTRODUCTION

### 1.1 MOTIVATIONS

Machine learning (ML) technologies have been integrated into autonomous systems, defining learning-enabled autonomous systems. These have succeeded tremendously in many complex tasks with high-dimensional state and action spaces. However, the recent incidents due to the deployment of ML models overshadow the revolutionizing potential of ML, especially for the safety-critical autonomous systems Zachary & Helen (2021). Developing safe ML is thus more vital today. In the ML community, deep reinforcement learning (DRL) has demonstrated breakthroughs in sequential decision-making in broad areas, ranging from autonomous driving Kendall et al. (2019) to games Silver et al. (2018). This motivates us to develop a DRL-based safe learning framework for achieving safe and complex tasks of safety-critical autonomous systems

Despite the tremendous success of DRL in many autonomous systems for complex decision-making, applying DRL to safety-critical autonomous systems remains a challenging problem. It has a deep root to the action policy of DRL being parameterized by deep neural networks (DNN), whose behaviors are hard to predict Huang et al. (2017) and verify Katz et al. (2017), raising the first safety concern. The second safety concern stems from the purely data-driven DNN that DRL adopts

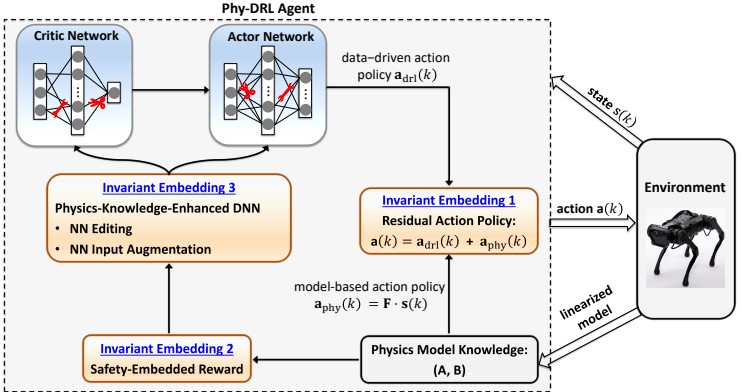

Figure 1: Phy-DRL Framework, applied to a quadruped robot.

---

*Equal contribution.
†Correspondence to Yanbing Mao: maoyanbing.eth@gmail.com.

for powerful function approximation and representation learning of action-value function, action policy, and environment states Mnih et al. (2015); Silver et al. (2016). Specifically, recent studies revealed that purely data-driven DNN applied to physical systems can infer relations violating physics laws, which sometimes leads to catastrophic consequences (e.g., data-driven blackout owning to violation of physical limits Zachary & Helen (2021)).

## 1.2 CONTRIBUTIONS

To address the aforementioned safety concerns, we propose the **Phy-DRL**: a physics-regulated deep reinforcement learning framework with enhanced safety assurance. Depicting in Figure 1, Phy-DRL has three novel (invariant-embedding) architectural designs:

- **Residual Action Policy**, which integrates data-driven-DRL action policy and physics-model-based action policy.
- **Safety-Embedded Reward**, in conjunction with the Residual Action Policy, empowers the Phy-DRL with a mathematically provable safety guarantee and fast training.
- **Physics-Knowledge-Enhanced Critic and Actor Networks**, whose neural architectures have two key components: i) NN input augmentation for directly capturing hard-to-learn features, and ii) NN editing, including link editing and activation editing, for guaranteeing strict compliance with available knowledge about the action-value function and action policy.

## 1.3 RELATED WORK AND OPEN PROBLEMS

**Residual Action Policy**. The recent research on DRL for controlling autonomous systems has shifted to integrating data-driven DRL and model-based decision-making, leading to a residual action policy diagram. In this diagram, the model-based action policy can guide the exploration of DRL agents during training. Meanwhile, the DRL policy learns to effectively deal with uncertainties and compensate for the model mismatch of the model-based action policy. The aims of existing residual frameworks Rana et al.; Li et al. (a); Cheng et al. (2019b); Johannink et al. (2019) mainly focus on stability guarantee, with the exception being Cheng et al. (2019a) on safety guarantee. Moreover, physics models are nonlinear, which poses difficulty in empowering analyzable and verifiable behavior. Furthermore, the model knowledge has not yet been explored to regulate the construction of DRL towards safety guarantee. In summary, the open problems in this domain are

**Problem 1.1.** *How to design a residual action policy to be a best-trade-off between the model-based action policy and the data-driven-DRL action policy?*

**Problem 1.2.** *How can the knowledge of the physics model be used to construct a DRL's reward towards a safety guarantee?*

**Safety-Embedded Reward**. A safety-embedded reward is crucial for DRL to search for policies that are safe. To achieve a safety guarantee, control Lyapunov function (CLF) is the potential safety-embedded reward Perkins & Barto (2002); Berkenkamp et al. (2017); Chang & Gao (2021); Zhao et al. (2023; 2024). Meanwhile, the seminal work Westenbroek et al. (2022) discovered that if DRL's reward is CLF-like, the systems controlled by a well-trained DRL policy can retain a mathematically-provable stability guarantee. Moving forward, the question of how to construct such a CLF-like reward for achieving a safety guarantee remains open, i.e.,

**Problem 1.3.** *What is the systematic guidance for constructing the safety-embedded reward (e.g., CLF-like reward) for DRL?*

**Knowledge-Enhanced Neural Networks**. The critical flaw of purely data-driven DNN, i.e., violation of physics laws, motivates the emerging research on physics-enhanced DNN. Current frameworks include physics-informed NN Wang & Yu; Willard et al. (2021); Jia et al. (2021); Lu et al. (2021); Chen et al. (2021); Xu & Darve (2022); Karniadakis et al. (2021); Wang et al. (2020); Cranmer et al. (2020) and physics-guided NN architectures Muralidhar et al. (2020); Masci et al. (2015); Monti et al. (2017); Horie et al.; Wang (2021); Li et al. (b). Both use compact partial differential equations (PDEs) for formulating training loss functions and/or architectural components. These frameworks improve consistency degree with prior physics knowledge but remain problematic in applying to DRL. For example, we define DRL's reward in advance. The critic network of DRL is to learn or estimate the expected future reward, also known as the action-value function. Because the

action-value function involves unknown future rewards, its compact governing equation is unavailable for physics-informed networks and physics-guided architectures. In summary, only partial knowledge about the action-value function and action policy is available, which thus motivates the open problem:

**Problem 1.4.** *How do we develop end-to-end critic and actor networks that strictly comply with partially available physics knowledge about the action-value function and action policy?*

### 1.4 SUMMARY: ANSWERS TO PROBLEM 1.1–PROBLEM 1.4

The proposed Phy-DRL answers Problem 1.1–Problem 1.4 simultaneously. As shown in Figure 1, the residual diagram of Phy-DRL simplifies the model-based action policy to be an analyzable and verifiable linear one, while offering fast training towards safety guarantee. Meanwhile, the linear model knowledge (leveraged for computing model-based policy) works as a model-based guidance for constructing the safety-embedded reward for DRL towards mathematically provable safety guarantee. Lastly, the proposed NN editing guarantees the strict compliance of critic and actor networks with partially available physics knowledge about the action-value function and action policy.

## 2 PRELIMINARIES

Table 1 in Appendix A summarizes notations that are used throughout the paper.

### 2.1 DYNAMICS MODEL OF REAL PLANT

The generic dynamics model of a real plant can be described by
$$\mathbf{s}(k+1) = \mathbf{A} \cdot \mathbf{s}(k) + \mathbf{B} \cdot \mathbf{a}(k) + \mathbf{f}(\mathbf{s}(k), \mathbf{a}(k)), \quad k \in \mathbb{N} \tag{1}$$
where $\mathbf{f}(\mathbf{s}(k), \mathbf{a}(k)) \in \mathbb{R}^n$ is the underlined{unknown} model mismatch, $\mathbf{A} \in \mathbb{R}^{n \times n}$ and $\mathbf{B} \in \mathbb{R}^{n \times m}$ denote known system matrix and control structure matrix, respectively, $\mathbf{s}(k) \in \mathbb{R}^n$ is the system state, $\mathbf{a}(k) \in \mathbb{R}^m$ is the applied control action. The available model knowledge pertaining to real plant (1) is represented by $(\mathbf{A}, \mathbf{B})$.

### 2.2 SAFETY DEFINITION

The considered safety problem stems from safety regulations or constraints on system states, which motives the following definition of safety set $\mathbb{X}$.

$$\textbf{Safety Set:} \quad \mathbb{X} \triangleq \left\{ \mathbf{s} \in \mathbb{R}^n \,|\, \underline{\mathbf{v}} \le \mathbf{D} \cdot \mathbf{s} - \mathbf{v} \le \overline{\mathbf{v}}, \quad \mathbf{D} \in \mathbb{R}^{h \times n}, \ \mathbf{v}, \overline{\mathbf{v}}, \underline{\mathbf{v}} \in \mathbb{R}^h \right\}. \tag{2}$$
where $\mathbf{D}$, $\mathbf{v}$, $\overline{\mathbf{v}}$ and $\underline{\mathbf{v}}$ are given in advance for formulating $h$ safety conditions. Considering the safety set, we present the definition of a safety guarantee.

**Definition 2.1.** Consider the safety set $\mathbb{X}$ in Equation (2) and its subset $\Omega$. The real plant (1) is said to be safety guaranteed, if for any $\mathbf{s}(1) \in \Omega \subseteq \mathbb{X}$, then $\mathbf{s}(k) \in \Omega \subseteq \mathbb{X}, \forall k > 1 \in \mathbb{N}$.

*Remark* 2.2 (**Role of** $\Omega$). The subset $\Omega$ is called the safety envelope, whose details will be explained in Section 5. The $\Omega$ will bridge many (i.e., high-dimensional) safety conditions in safety set Equation (2) and one-dimensional safety-embedded reward. Meanwhile, Definition 2.1 indicates that safety guarantee means the Phy-DRL successfully searches for a policy that renders $\Omega$ invariant (i.e., operating from any initial sample inside $\Omega$, system state never leaves $\Omega$ at any time).

## 3 DESIGN OVERVIEW: INVARIANT EMBEDDINGS

In this paper, an 'invariant' refers to a prior policy, prior knowledge, or a designed property independent of DRL agent training. As shown in Figure 1, the proposed Phy-DRL can address Problem 1.1–Problem 1.4 because of three invariant-embedding designs. Specifically, 1) residual action policy, which integrates data-driven action policy with an invariant model-based action policy that completely depends on prior model knowledge $(\mathbf{A}, \mathbf{B})$. ii) Safety-embedded reward, whose off-line-designed inequality (shown in Equation (8)) for assistance in delivering the mathematically provable safety guarantee is also completely independent of agent training. iii) Physics-knowledge-enhanced DNN, whose NN editing embeds the prior invariant knowledge about the action-value function and action policy to the critic and actor networks, respectively. The following Section 4, Section 5 and Section 6 detail the three designs, respectively.

## 4 INVARIANT EMBEDDING 1: RESIDUAL ACTION POLICY

Showing in Figure 1, the applied control action $\mathbf{a}(k)$ from Phy-DRL is given in the residual form:

$$\mathbf{a}(k) = \underbrace{\mathbf{a}_{\mathrm{drl}}(k)}_{\text{data-driven}} + \underbrace{\mathbf{a}_{\mathrm{phy}}(k)}_{\text{invariant: model-based}}, \tag{3}$$

where the $\mathbf{a}_{\mathrm{drl}}(k)$ denotes a date-driven action from DRL, while the $\mathbf{a}_{\mathrm{phy}}(k)$ is a model-based action, computed according to the invariant policy:

$$\mathbf{a}_{\mathrm{phy}}(k) = \mathbf{F} \cdot \mathbf{s}(k), \tag{4}$$

where the computation of $\mathbf{F}$ is based on the model knowledge $(\mathbf{A}, \mathbf{B})$, carried out in Section 5.

The developed Phy-DRL is based on actor-critic architecture in DRLLillicrap et al. (2016); Schulman et al.; Haarnoja et al. (2018) for searching an action policy $\mathbf{a_{drl}}(k) = \pi(\mathbf{s}(k))$ that maximizes the expected return from the initial state $\mathbf{s}(k)$:

$$\mathcal{Q}^{\pi}(\mathbf{s}(k), \mathbf{a}_{\mathrm{drl}}(k)) = \mathbf{E}_{\mathbf{s}(k)\sim\rho,\ \mathbf{a_{drl}}(k)\sim\pi}\left[\sum_{t=k}^{\infty}\gamma^{t-k}\cdot\mathcal{R}\left(\mathbf{s}(t), \mathbf{a}_{\mathrm{drl}}(t)\right)\right], \tag{5}$$

where $\rho$ represents the initial state distribution, $\mathcal{R}(\cdot)$ maps a state-action to a real-value reward, $\gamma \in [0,1]$ is the discount factor. The expected return (5) and action policy $\pi(\cdot)$ are parameterized by the critic and actor networks, respectively.

## 5 INVARIANT EMBEDDING 2: SAFETY-EMBEDDED REWARD

The current safety formula (2) is not ready for constructing the safety-embedded reward yet, since it has multiple safety conditions while the reward $\mathcal{R}(\cdot)$ in Equation (5) is one-dimensional. To bridge the gap, we introduce the following safety envelope, which converts multi-dimensional safety conditions into a scalar value.

$$\text{Safety Envelope: } \Omega \triangleq \left\{ \mathbf{s} \in \mathbb{R}^n \,|\, \mathbf{s}^\top \cdot \mathbf{P} \cdot \mathbf{s} \le 1,\ \mathbf{P} \succ 0 \right\}. \tag{6}$$

The following lemma builds a connection: safety envelope is a subset of the safety set (also required in Definition 2.1). Its formal proof appears in Appendix C.

**Lemma 5.1.** *Consider the sets defined in Equation* (2) *and Equation* (6). *We have* $\Omega \subseteq \mathbb{X}$, *if*

$$\left[\overline{\mathbf{D}}\right]_{i,:}\cdot\mathbf{P}^{-1}\cdot\left[\overline{\mathbf{D}}^\top\right]_{:,i} \le 1 \text{ and } [\underline{\mathbf{D}}]_{i,:}\cdot\mathbf{P}^{-1}\cdot\left[\underline{\mathbf{D}}^\top\right]_{:,i} = \begin{cases} \ge 1, & [\mathbf{d}]_i = 1 \\ \le 1, & [\mathbf{d}]_i = -1 \end{cases}, \quad i \in \{1, 2, \ldots, h\} \tag{7}$$

*where* $\overline{\mathbf{D}} = \frac{\mathbf{D}}{\overline{\Lambda}}$, $\underline{\mathbf{D}} = \frac{\mathbf{D}}{\underline{\Lambda}}$, *and* $\mathbf{d}$, $\overline{\Lambda}$ *and* $\underline{\Lambda}$ *are defined below for* $i, j \in \{1, 2, \ldots, h\}$:

$$[\mathbf{d}]_i \triangleq \begin{cases} 1, & [\underline{\mathbf{v}}+\mathbf{v}]_i > 0 \\ 1, & [\overline{\mathbf{v}}+\mathbf{v}]_i < 0 \\ -1, & \text{otherwise} \end{cases}, \quad [\overline{\Lambda}]_{i,j} \triangleq \begin{cases} 0, & i \ne j \\ [\overline{\mathbf{v}}+\mathbf{v}]_i, & [\underline{\mathbf{v}}+\mathbf{v}]_i > 0 \\ [\underline{\mathbf{v}}+\mathbf{v}]_i, & [\overline{\mathbf{v}}+\mathbf{v}]_i < 0 \\ [\overline{\mathbf{v}}+\mathbf{v}]_i, & \text{otherwise} \end{cases}, \quad [\underline{\Lambda}]_{i,j} \triangleq \begin{cases} 0, & i \ne j \\ [\underline{\mathbf{v}}+\mathbf{v}]_i, & [\underline{\mathbf{v}}+\mathbf{v}]_i > 0 \\ [\overline{\mathbf{v}}+\mathbf{v}]_i, & [\overline{\mathbf{v}}+\mathbf{v}]_i < 0 \\ [-\underline{\mathbf{v}} - \mathbf{v}]_i, & \text{otherwise} \end{cases}.$$

Referring to model knowledge $(\mathbf{A}, \mathbf{B})$, Equation (6) and Equation (4), the proposed reward is

$$\mathcal{R}(\mathbf{s}(k), \mathbf{a}_{\mathrm{drl}}(k)) = \underbrace{\mathbf{s}^\top(k) \cdot \overline{\mathbf{A}}^\top \cdot \mathbf{P} \cdot \overline{\mathbf{A}} \cdot \mathbf{s}(k) - \mathbf{s}^\top(k+1) \cdot \mathbf{P} \cdot \mathbf{s}(k+1)}_{\triangleq\, r(\mathbf{s}(k),\, \mathbf{s}(k+1)):\ \text{invariant property } \mathbf{P} - \overline{\mathbf{A}}^\top\cdot\mathbf{P}\cdot\overline{\mathbf{A}} \succ 0} + w(\mathbf{s}(k), \mathbf{a}_{\mathrm{drl}}(k)), \tag{8}$$

where the sub-reward $r(\mathbf{s}(k), \mathbf{s}(k+1))$ is safety-embedded, and we define:

$$\overline{\mathbf{A}} \triangleq \mathbf{A} + \mathbf{B} \cdot \mathbf{F}. \tag{9}$$

*Remark* 5.2 (Sub-rewards). In Equation (8), the safety-embedded sub-reward $r(\mathbf{s}(k), \mathbf{s}(k+1))$ is critical for keeping a system safe, such as avoiding car crashes, and preventing car sliding and slipping in an icy road. The sub-reward $w(\mathbf{s}(k), \mathbf{a}_{\mathrm{drl}}(k))$ aims at high-performance operations, such as minimizing energy consumption of resource-limited robots Yang et al. (2022), which can be optional in some time- and safety-critical environments.

Moving forward, we present the following theorem, which states the conditions on matrices $\mathbf{F}$, $\mathbf{P}$ and reward for safety guarantee, whose proof is given in Appendix D.1.

**Theorem 5.3 (Mathematically Provable Safety Guarantee).** *Consider the safety set $\mathbb{X}$ (2), the safety envelope $\Omega$ (6), and the system (1) under control of Phy-DRL. The matrices $\mathbf{F}$ and $\mathbf{P}$ involved in the model-based action policy (4) and the safety-embedded reward (8) are computed according to*

$$\mathbf{F} = \mathbf{R} \cdot \mathbf{Q}^{-1}, \quad \mathbf{P} = \mathbf{Q}^{-1}, \tag{10}$$

*where $\mathbf{R}$ and $\mathbf{Q}^{-1}$ satisfy the inequalities (7) and*

$$\begin{bmatrix} \alpha \cdot \mathbf{Q} & \mathbf{Q} \cdot \mathbf{A}^\top + \mathbf{R}^\top \cdot \mathbf{B}^\top \\ \mathbf{A} \cdot \mathbf{Q} + \mathbf{B} \cdot \mathbf{R} & \mathbf{Q} \end{bmatrix} \succ 0, \quad \text{with a given } \alpha \in (0,1). \tag{11}$$

*Given any $\mathbf{s}(1) \in \Omega$, the system state $\mathbf{s}(k) \in \Omega \subseteq \mathbb{X}$ holds $\forall k \in \mathbb{N}$ (i.e., the safety of system (1) is guaranteed), if the sub-reward $r(\mathbf{s}(k), \mathbf{s}(k+1))$ in (8) satisfies $r(\mathbf{s}(k), \mathbf{s}(k+1)) \geq \alpha - 1$, $\forall k \in \mathbb{N}$.*

*Remark* 5.4 (**Solving Optimal $\mathbf{R}$ and $\mathbf{Q}$**). Equation (4), Equation (8) and Equation (9) imply that the designs of model-based action policy and safety-embedded reward equate the computations of $\mathbf{F}$ and $\mathbf{P}$. While Equation (10) shows the computations depend on $\mathbf{R}$ and $\mathbf{Q}$ only. So, the remaining work is obtaining $\mathbf{R}$ and $\mathbf{Q}$. There are multiple toolboxes for solving $\mathbf{R}$ and $\mathbf{Q}$ from linear matrix inequalities (LMIs) (7) and (11), such as MATLAB's LMI Solver Boyd et al. (1994). What we are more interested in is finding optimal $\mathbf{R}$ and $\mathbf{Q}$ that can maximize the safety envelope. We note the volume of a safety envelope (6) is proportional to $\sqrt{\det(\mathbf{P}^{-1})}$, the interested optimal problem is thus a typical analytic centering problem, formulated as given a $\alpha \in (0,1)$,

$$\underset{\mathbf{Q}, \mathbf{R}}{\arg\min} \left\{ \log\det\left(\mathbf{Q}^{-1}\right) \right\} = \underset{\mathbf{Q}, \mathbf{R}}{\arg\max} \left\{ \log\det\left(\mathbf{P}^{-1}\right) \right\}, \text{ subject to LMIs (7) and (11)} \tag{12}$$

from which, optimal $\mathbf{R}$ and $\mathbf{Q}$ can be solved via CVX toolbox Grant et al. (2009).

*Remark* 5.5 (**$\mathbf{F}$ is given**). Equation (12) also works in the scenario of a given model-based action policy (i.e., $\mathbf{F}$), which is carried out in Section 7.2 as an example.

*Remark* 5.6 (**Provable Stability Guarantee**). Following the same proof path of Theorem 5.3, Phy-DRL also exhibits the mathematically provable stability guarantee, which is presented in Appendix E.

*Remark* 5.7 (**Fast Training**). The proof path of Theorem 5.3 is leveraged to reveal the driving factor of Phy-DRL's fast training towards safety guarantee, which is presented in Appendix D.2.

*Remark* 5.8 (**Obtaining $(\mathbf{A}, \mathbf{B})$**). For a system with an available nonlinear dynamics model, the model knowledge $(\mathbf{A}, \mathbf{B})$ can be directly obtained by simplifying the nonlinear model to a linear one. While for a system whose dynamics model is not available, $(\mathbf{A}, \mathbf{B})$ can be obtained via system identification Oymak & Ozay (2019), as used in social systems Mao et al. (2022).

# 6 INVARIANT EMBEDDING 3: PHYSICS-KNOWLEDGE-ENHANCED DNN

The Phy-DRL is built on the *actor-critic* architecture, where a critic and an actor network are used to approximate the action-value function (i.e., $\mathcal{Q}(\mathbf{s}(k), \mathbf{a}_{\mathrm{drl}}(k))$ in Equation (5)) and learn an action policy (i.e., $\mathbf{a}_{\mathrm{drl}}(k) = \pi(\mathbf{s}(k))$), respectively. We note from Equation (5) that the action-value function is a direct function of our defined reward but involves unknown future rewards. So, some invariant knowledge exists that the critic and/or actor networks shall strictly comply with, which moti-

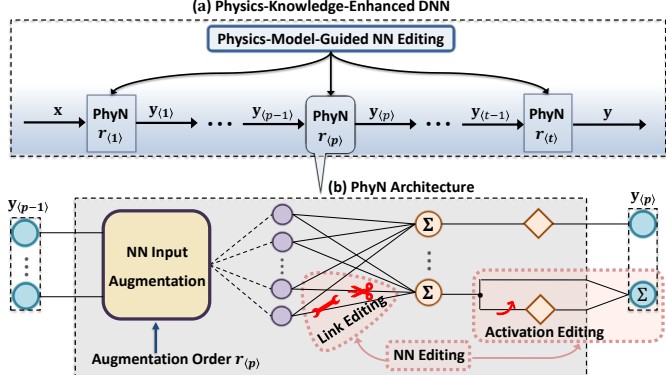

Figure 2: Physics-Knowledge-Enhanced DNN architecture.

vates Problem 1.4. To address this problem, as shown in Figure 1, we develop physics-enhanced critic and actor networks for Phy-DRL. The proposed networks are built on physics-knowledge-enhanced DNN, whose architecture is depicted in Figure 2. The DNN has two innovations in neural architecture:

i) Neural Network (NN) Input Augmentation described by Algorithm 1, and ii) Physics-Model-Guided NN Editing described by Algorithm 2. To understand how Algorithm 1 and Algorithm 2 address Problem 1.4, we describe the ground-truth models of action-value function and action policy as

$$\mathcal{Q}\left(\mathbf{s}, \mathbf{a}_{\mathrm{drl}}\right) = \underbrace{\mathbf{A}_{\mathcal{Q}}}_{\text{weight matrix}} \cdot \underbrace{\mathfrak{m}(\mathbf{s}, \mathbf{a}_{\mathrm{drl}}, r_{\mathcal{Q}})}_{\text{node-representation vector}} + \underbrace{\mathbf{p}(\mathbf{s}, \mathbf{a}_{\mathrm{drl}})}_{\text{unknown model mismatch}} \in \mathbb{R}, \tag{13}$$

$$\pi\left(\mathbf{s}\right) = \underbrace{\mathbf{A}_{\pi}}_{\text{weight matrix}} \cdot \underbrace{\mathfrak{m}(\mathbf{s}, r_{\pi})}_{\text{node-representation vector}} + \underbrace{\mathbf{p}(\mathbf{s})}_{\text{unknown model mismatch}} \in \mathbb{R}^{\mathrm{len}(\pi(\mathbf{s}))}, \tag{14}$$

where the vectors $\mathfrak{m}(\mathbf{s}, \mathbf{a}_{\mathrm{drl}}, r_{\mathcal{Q}})$ and $\mathfrak{m}(\mathbf{s}, r_{\pi})$ are, respectively, augmentations of input vectors $[\mathbf{s}; \mathbf{a}_{\mathrm{drl}}]$ and $\mathbf{s}$, which embraces all the non-missing and non-redundant monomials of a Taylor series. One motivation behind the augmentations is that according to Taylor's Theorem in Appendix G, the Taylor series can approximate arbitrary nonlinear functions with controllable accuracy via controlling series orders: $r_{\mathcal{Q}}$ and $r_{\pi}$. The second motivation is our proposed safety-embedded reward $r(\mathbf{s}(k), \mathbf{s}(k+1))$ in Equation (8) is a typical Taylor series and is pre-defined. If using the Taylor series to approximate the action-value function and an action policy, we can also discover hidden invariant knowledge.

Algorithm 2 needs inputs of knowledge sets $\mathbb{K}_{\mathcal{Q}}$ and $\mathbb{K}_{\pi}$. which include available knowledge about governing Equation (13) and Equation (14), respectively. The two knowledge sets are defined below.

$$\mathbb{K}_{\mathcal{Q}} \triangleq \{[\mathbf{A}_{\mathcal{Q}}]_i \,|\, \text{no } [\mathfrak{m}(\mathbf{s}, \mathbf{a}_{\mathrm{drl}}, r_{\mathcal{Q}})]_i \text{ in } \mathbf{p}(\mathbf{s}, \mathbf{a}_{\mathrm{drl}}), \, i \in \{1, \ldots, \mathrm{len}(\mathfrak{m}(\mathbf{s}, \mathbf{a}_{\mathrm{drl}}, r_{\mathcal{Q}}))\}\}, \tag{15}$$

$$\mathbb{K}_{\pi} \triangleq \left\{[\mathbf{A}_{\pi}]_{i,j} \,\middle|\, \text{no } [\mathfrak{m}(\mathbf{s}, r_{\pi})]_j \text{ in } [\mathbf{p}(\mathbf{s})]_i, \, i \in \{1, \ldots, \mathrm{len}(\mathbf{p}(\mathbf{s}))\}, j \in \{1, \ldots, \mathrm{len}(\mathfrak{m}(\mathbf{s}, r_{\pi}))\}\right\}. \tag{16}$$

*Remark* 6.1 (**Toy Examples: Obtaining Knowledge Sets**). Due to the page limit, an example for obtaining $\mathbb{K}_{\mathcal{Q}}$ via Taylor's theorem is presented in Appendix H. This example is about obtaining $\mathbb{K}_{\pi}$. According to the dynamics and control of vehicles, the throttle command of traction control for preventing sliding and slipping depends on longitudinal velocity (denoted by $v$) and angular velocity (denoted by $w$) only Rajamani (2011); Mao et al. (2023). For simplification, we let $\mathbf{s} = [v, w, \zeta]^{\top}$, where $\zeta$ denotes yaw. While action policy $\pi\left(\mathbf{s}\right) \in \mathbb{R}^2$, with $[\pi\left(\mathbf{s}\right)]_1$ and $[\pi\left(\mathbf{s}\right)]_2$ denote the throttle command and steering command, respectively. By Algorithm 1 with $r_{\langle t \rangle} = r_{\pi} = 2$ and $\mathbf{y}_{\langle t \rangle} = \mathbf{s}$, we have $\mathfrak{m}(\mathbf{s}, r_{\pi}) = [1, v, w, \zeta, v^2, vw, v\zeta, w^2, w\zeta, \zeta^2]^{\top}$. Considering Equation (14), the $\mathfrak{m}(\mathbf{s}, r_{\pi})$, in conjunction with the knowledge "$[\pi\left(\mathbf{s}\right)]_1$ depends on $w$ and $v$ only", leads to the information: 1) $[\mathbf{p}(\mathbf{s})]_1$ in Equation (14) in this example does not have monomials: $1, \zeta, v\zeta, w\zeta$, and $\zeta^2$, and 2)

$$\mathbf{A}_{\pi} = \begin{bmatrix} 0 & w_1 & w_2 & 0 & w_3 & w_4 & 0 & w_5 & 0 & 0 \\ w_6 & w_7 & w_8 & w_9 & w_{10} & w_{11} & w_{12} & w_{13} & w_{14} & w_{15} \end{bmatrix},$$

where $w_1, \ldots, w_{15}$ are learning weights. Referring to Equation (16), we then have $\mathbb{K}_{\pi} = \{[\mathbf{A}_{\pi}]_{1,1} = 0, [\mathbf{A}_{\pi}]_{1,2} = w_1, \ldots, [\mathbf{A}_{\pi}]_{1,10} = 0\}$.

With knowledge sets at hand, we can introduce two design aims for addressing Problem 1.4.

**Aim 6.2.** *Given $\mathbb{K}_{\mathcal{Q}}$ (15), consider the critic network built on physic-knowledge-enhanced DNN in Figure 2, where $\mathbf{x} = (\mathbf{s}, \mathbf{a}_{drl})$ and $\mathbf{y} = \widehat{Q}\left(\mathbf{s}, \mathbf{a}_{drl}\right)$ (i.e., $\mathbf{y}$ approximates $\mathcal{Q}\left(\mathbf{s}, \mathbf{a}_{drl}\right)$). The end-to-end input/output of the critic network strictly complies with available knowledge about the governing Equation (13), i.e., if $[\mathbf{A}_{\mathcal{Q}}]_i \in \mathbb{K}_{\mathcal{Q}}$, then $\mathbf{y}$ does not have monomials $[\mathfrak{m}(\mathbf{s}, \mathbf{a}_{drl}, r_{\mathcal{Q}})]_i$.*

**Aim 6.3.** *Given $\mathbb{K}_{\pi}$ (16), consider the actor network built on physic-knowledge-enhanced DNN in Figure 2, where $\mathbf{x} = \mathbf{s}$ and $\mathbf{y} = \widehat{\pi}\left(\mathbf{s}\right)$ (i.e., $\mathbf{y}$ approximates $\pi\left(\mathbf{s}\right)$). The end-to-end input/output of the actor network strictly complies with available knowledge about the governing Equation (14), i.e., if $[\mathbf{A}_{\pi}]_{i,j} \in \mathbb{K}_{\pi}$, then $[\mathbf{y}]_i$ does not have monomials $[\mathfrak{m}(\mathbf{s}, r_{\pi})]_j$.*

Algorithm 2, in conjunction with Algorithm 1, is able to deliver Aim 6.2 and Aim 6.3. It is formally stated in Theorem 6.4, whose proof appears in Appendix J.

**Theorem 6.4.** *If the critic and actor networks are built on physics-knowledge-enhanced DNN (described in Figure 2), whose NN input augmentation and NN editing are described by Algorithm 1 and Algorithm 2, respectively, then Aim 6.2 and Aim 6.3 are achieved.*

Finally, we refer to the toy example in Remark 6.1 for an overview of NN editing. For the end-to-end mapping $[\mathbf{y}]_1 = [\widehat{\pi}\left(\mathbf{s}\right)]_1$, given $\mathbb{K}_{\pi}$ and output of Algorithm 1, the link editing of Algorithm 2 removes all connections with node representations $1, \zeta, v\zeta, w\zeta, \zeta^2$ and maintain link connections with $v, w, v^2, vw$ and $w^2$. Meanwhile, the action editing of Algorithm 2 guarantees the usages of action functions in all PhyN layers do not introduce monomials of $\zeta$ to the mapping $[\mathbf{y}]_1 = [\widehat{\pi}\left(\mathbf{s}\right)]_1$.

**Algorithm 1** NN Input Augmentation      ▷ Aim at representation vectors to embrace all the non-missing and non-redundant monomials of the Taylor series.

1: **Input:** augmentation order $r_{\langle t \rangle}$, input $\mathbf{y}_{\langle t \rangle}$;      ▷ $t$ indicates layer number, e.g, $t = 2$ denotes the second NN layer.
2: Generate index vector of input: $\mathbf{i} \leftarrow [1; 2; \ldots; \text{len}(\mathbf{y}_{\langle t \rangle})]$;
3: Initialize augmentation vector: $\mathfrak{m}(\mathbf{y}_{\langle t \rangle}, r_{\langle t+1 \rangle}) \leftarrow \mathbf{y}_{\langle t \rangle}$;
4: **for** $\_ = 2$ to $r_{\langle t \rangle}$ **do**
5:   **for** $i = 1$ to $\text{len}(\mathbf{y}_{\langle t \rangle})$ **do**
6:     Compute temp: $\mathbf{t}_a \leftarrow [\mathbf{y}_{\langle t \rangle}]_i \cdot [\mathbf{y}_{\langle t \rangle}]_{[[\mathbf{i}]_i \,:\, \text{len}(\mathbf{y}_{\langle t \rangle})]}$;      ▷ Capture hard-to-learn nonlinear representations, in the form of monomials of the Taylor series, such as the 2nd-order monomials of sub-reward $r(\mathbf{s}(k), \mathbf{s}(k+1))$ in Equation (8).
7:     **if** $i == 1$ **then**
8:       Generate temp: $\widetilde{\mathbf{t}}_b \leftarrow \widetilde{\mathbf{t}}_a$;
9:     **else if** $i > 1$ **then**
10:       Generate temp: $\widetilde{\mathbf{t}}_b \leftarrow \left[\widetilde{\mathbf{t}}_b; \widetilde{\mathbf{t}}_a\right]$;      ▷ Avoid missing and redundant monomials.
11:     **end if**
12:     Update index entry: $[\mathbf{i}]_i \leftarrow \text{len}(\mathbf{y}_{\langle t \rangle})$;
13:     Augment: $\mathfrak{m}(\mathbf{y}_{\langle t \rangle}, r_{\langle t+1 \rangle}) \leftarrow \left[\mathfrak{m}(\mathbf{y}_{\langle t \rangle}, r_{\langle t \rangle}); \widetilde{\mathbf{t}}_b\right]$;
14:   **end for**
15: **end for**      ▷ Line 4–Line 15 generate node-representation vector that embraces all the non-missing and non-redundant monomials of the Taylor series. One illustration example is in Figure 8 in Appendix F.
16: **Output**: $\mathfrak{m}(\mathbf{y}_{\langle t \rangle}, r_{\langle t \rangle}) \leftarrow \left[\mathbf{1}; \mathfrak{m}(\mathbf{y}_{\langle t \rangle}, r_{\langle t+1 \rangle})\right]$.      ▷ Controllable Model Accuracy: The algorithm provides one option of approximating the ground-truth models (13) and (14) via Taylor series. According to Taylor's Theorem (Chapter 2.4 Königsberger (2013)), the networks have controllable model accuracy by controlling augmentation orders $r_{\langle t \rangle}$ (see Appendix G for further explanations).

**Algorithm 2** Physics-Model-Guided Neural Network Editing      ▷ Perform on deep PhyN, and each layer needs Algorithm 1 for generating node-representation vectors.   Detailed explanations of Algorithm 2 appear in Appendix I

1: **Input:** Network type set $\mathbb{T} = \{'\mathcal{Q}', '\pi'\}$, knowledge sets $\mathbb{K}_{\mathcal{Q}}$ (15) and $\mathbb{K}_{\pi}$ (16), number of PhyNs $p_{\mathcal{Q}}$ and $p_{\pi}$, origin input $\mathbf{x}$, augmentation orders $r_{\mathcal{Q}}$ and $r_{\pi}$, model matrices $\mathbf{A}_{\mathcal{Q}}$ and $\mathbf{A}_{\pi}$, terminal output dimension $\text{len}(\mathbf{y})$.
2: Choose network type $\varpi \in \mathbb{T}$;
3: Specify augmentation order of the first PhyN: $r_{\langle 1 \rangle} \leftarrow r_{\varpi}$;
4: **for** $t = 1$ to $p_{\varpi}$ **do**
5:   **if** $t == 1$ **then**
6:     Generate node-representation vector $\mathfrak{m}(\mathbf{x}, r_{\langle 1 \rangle})$ via Algorithm 1;      ▷ Corresponding to $\mathfrak{m}(\mathbf{s}, \mathbf{a}_{\text{drl}}, r_{\mathcal{Q}})$ and $\mathfrak{m}(\mathbf{s}, r_{\pi})$ in the ground-truth models (13) and (14), because $r_{\langle 1 \rangle} \leftarrow r_{\varpi}$.
7:     Generate raw weight matrix via gradient descent algorithm: $\mathbf{W}_{\langle 1 \rangle}$; ▷ Raw weight matrix usually responds to a fully-connected NN layer, which can violate physics knowledge.
8:     Generate knowledge matrix $\mathbf{K}_{\langle 1 \rangle}$: $[\mathbf{K}_{\langle 1 \rangle}]_{i,j} \leftarrow \begin{cases} [\mathbf{A}_{\varpi}]_{i,j}, & [\mathbf{A}_{\varpi}]_{i,j} \in \mathbb{K}_{\varpi} \\ 0, & \text{otherwise} \end{cases}$;      ▷ Include all elements in knowledge set.
9:     Generate weight-masking matrix $\mathbf{M}_{\langle 1 \rangle}$: $[\mathbf{M}_{\langle 1 \rangle}]_{i,j} \leftarrow \begin{cases} 0, & [\mathbf{A}_{\varpi}]_{i,j} \in \mathbb{K}_{\varpi} \\ 1, & \text{otherwise} \end{cases}$;
10:     Generate activation-masking vector $\mathbf{a}_{\langle 1 \rangle}$: $[\mathbf{a}_{\langle 1 \rangle}]_i \leftarrow \begin{cases} 0, & [\mathbf{M}_{\langle 1 \rangle}]_{i,j} = 0, \forall j \in \{1, \ldots, \text{len}(\mathfrak{m}(\mathbf{x}, r_{\langle 1 \rangle}))\} \\ 1, & \text{otherwise} \end{cases}$;
11:   **else if** $t > 1$ **then**
12:     Generate raw weight matrix via gradient descent algorithm: $\mathbf{W}_{\langle t \rangle}$;  ▷ Raw weight matrix usually responds to a fully-connected NN layer, which can violate physics knowledge.
13:     Generate node-representation vector $\mathfrak{m}(\mathbf{y}_{\langle t-1 \rangle}, r_{\langle t \rangle})$ via Algorithm 1;
14:     Generate knowledge matrix: $\mathbf{K}_{\langle t \rangle} \leftarrow \left[\begin{array}{cc:c} \mathbf{0}_{\text{len}(\mathbf{y})} & \mathbf{I}_{\text{len}(\mathbf{y})} & \mathbf{O}_{\text{len}(\mathbf{y}) \times (\text{len}(\mathfrak{m}(\mathbf{y}_{\langle t-1 \rangle}, r_{\langle t \rangle})) - \text{len}(\mathbf{y}) - 1)} \\ \hdashline & \mathbf{O}_{(\text{len}(\mathbf{y}_{\langle t \rangle}) - \text{len}(\mathbf{y})) \times \text{len}(\mathfrak{m}(\mathbf{y}_{\langle t-1 \rangle}, r_{\langle t \rangle}))} \end{array}\right]$;
15:     Generate weight-masking matrix $\mathbf{M}_{\langle t \rangle}$:
$$[\mathbf{M}_{\langle t \rangle}]_{i,j} \leftarrow \begin{cases} 0, & \frac{\partial [\mathfrak{m}(\mathbf{y}_{\langle t \rangle}, r_{\langle t \rangle})]_j}{\partial [\mathfrak{m}(\mathbf{x}, r_{\langle 1 \rangle})]_v} \neq 0 \text{ and } [\mathbf{M}_{\langle 1 \rangle}]_{i,v} = 0, \; v \in \{1, \ldots, \text{len}(\mathfrak{m}(\mathbf{x}, r_{\langle 1 \rangle}))\} \\ 1, & \text{otherwise} \end{cases};$$
16:     Generate activation-masking vector $\mathbf{a}_{\langle t \rangle} \leftarrow \left[\mathbf{a}_{\langle 1 \rangle}; \mathbf{1}_{\text{len}(\mathbf{y}_{\langle t \rangle}) - \text{len}(\mathbf{y})}\right]$;
17:   **end if**
18:   Generate uncertainty matrix $\mathbf{U}_{\langle t \rangle} \leftarrow \mathbf{M}_{\langle t \rangle} \odot \mathbf{W}_{\langle t \rangle}$;
19:   Compute output: $\mathbf{y}_{\langle t \rangle} \leftarrow \mathbf{K}_{\langle t \rangle} \cdot \mathfrak{m}(\mathbf{y}_{\langle t-1 \rangle}, r_{\langle t \rangle}) + \mathbf{a}_{\langle t \rangle} \odot \text{act}\left(\mathbf{U}_{\langle t \rangle} \cdot \mathfrak{m}\left(\mathbf{y}_{\langle t-1 \rangle}, r_{\langle t \rangle}\right)\right)$;
20: **end for**
21: **Output**: $\widehat{\mathbf{y}} \leftarrow \mathbf{y}_{\langle p \rangle}$.

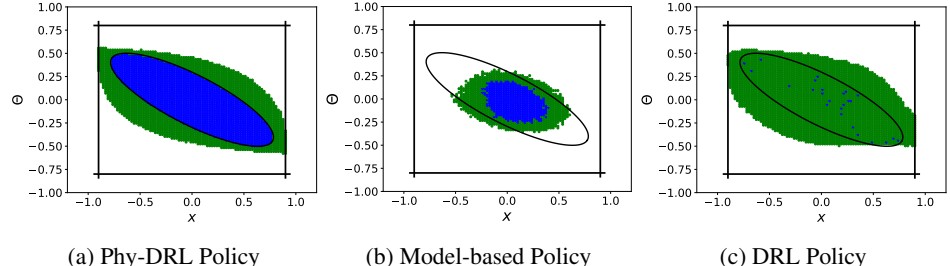

(a) Phy-DRL Policy      (b) Model-based Policy      (c) DRL Policy

Figure 3: Blue: area of IE samples defined in Equation (18). Green: area of EE samples defined in Equation (19). Rectangular area: safety set. Ellipse area: safety envelope.

## 7 EXPERIMENTS

### 7.1 CART-POLE SYSTEM

We take the cart-pole simulator provided in Open-AI Gym Brockman et al. (2016). Its mechanical analog is shown in Figure 11 in Appendix K, characterized by pendulum's angle $\theta$ and angular velocity $\omega \overset{\triangle}{=} \dot{\theta}$, and cart's position $x$ and velocity $v \overset{\triangle}{=} \dot{x}$. Phy-DRL's action policy is to stabilize the pendulum at equilibrium $\mathbf{s}^* = [x^*, v^*, \theta^*, \omega^*]^\top = [0, 0, 0, 0]^\top$, while constraining system state to

$$\textbf{Safety Set:} \quad \mathbb{X} = \left\{ \mathbf{s} \in \mathbb{R}^4 \,\middle|\, -0.9 \le x \le 0.9, \; -0.8 < \theta < 0.8 \right\}. \tag{17}$$

To demonstrate the robustness of Phy-DRL, we intentionally create a large model mismatch for obtaining a model-based action policy of Phy-DRL. Specifically, as explained in Appendix K.1, the physics-model knowledge represented by $(\mathbf{A}, \mathbf{B})$ is obtained through ignoring friction force, and letting $\cos\theta \approx 1$, $\sin\theta \approx \theta$ and $\omega^2 \sin\theta \approx 0$. The system trajectories in Figure 12 in Appendix K.4 show that the sole model-based action policy does not guarantee safety.

The computations of $\mathbf{F}$, $\mathbf{P}$ and $\overline{\mathbf{A}}$ are presented in Appendix K.1–Appendix K.3. Meanwhile, for the high-performance sub-reward in Equation (8), we let $w(\mathbf{s}(k), \mathbf{a}_{\mathrm{drl}}(k)) = -\mathbf{a}_{\mathrm{drl}}^2(k)$. Finally, to validate our theoretical results and present experimental comparisons, we define two safe samples:

$$\textbf{Safe Internal-Envelope (IE) Sample} \overset{\triangle}{=} \widetilde{\mathbf{s}}\textbf{:} \text{ if } \mathbf{s}(1) = \widetilde{\mathbf{s}} \in \Omega, \text{ then } \mathbf{s}(k) \in \Omega, \forall k \in \mathbb{N}. \tag{18}$$

$$\textbf{Safe External-Envelope (EE) Sample} \overset{\triangle}{=} \widetilde{\mathbf{s}}\textbf{:} \text{ if } \mathbf{s}(1) = \widetilde{\mathbf{s}} \in \mathbb{X}, \text{ then } \mathbf{s}(k) \in \mathbb{X} \setminus \Omega, \exists k \in \mathbb{N}. \tag{19}$$

We consider a CLF (control Lyapunov function) reward, proposed in Westenbroek et al. (2022), as $\mathcal{R}(\cdot) = \mathbf{s}^\top(k) \cdot \mathbf{P} \cdot \mathbf{s}(k) - \mathbf{s}^\top(k+1) \cdot \mathbf{P} \cdot \mathbf{s}(k+1) + w(\mathbf{s}(k), \mathbf{a}(k))$, where the $\mathbf{P}$ is the same as the one in the Phy-DRL's safety-embedded reward. We mainly compare our Phy-DRL with purely data-driven DRL having the CLF reward for testing. Both models are trained for 75000 steps and have the same configurations of critic and actor networks, presented in Appendix K.5. The performance metrics are the areas of IE and EE samples. Figure 3 shows that i) Phy-DRL successfully renders the safety envelope invariant, demonstrating Theorem 5.3 ($r(\mathbf{s}(k), \mathbf{s}(k+1)) \ge \alpha - 1$ holds in final training episode), and ii) the safety areas of the sole model-based policy and the purely data-driven DRL are much smaller. Additional comparisons with incorporating a model for model-based DRL (with the proposed CLF reward) and our Phy-DRL for state prediction are presented in Appendix K.7.

### 7.2 QUADRUPED ROBOT

In this experiment, action policies' missions are concurrent safe center-gravity management, safe lane tracking alone x-axis, and safe velocity regulation. We define safety constraints as:

$$\mathbb{X} = \{\widehat{\mathbf{s}} \,|\, \text{CoM z-height} - 0.24\,\text{m}| \le 0.13\,\text{m}, |\text{yaw}| \le 0.17\,\text{rad}, |\text{CoM x-velocity} - r_{\mathrm{x}}| \le |r_{\mathrm{x}}|\}, \tag{20}$$

$$\text{Targeted Equilibrium:} \quad \widehat{\mathbf{s}}^* = [0;\, 0;\, 0.24\,\text{m};\, 0;\, 0;\, 0;\, r_{\mathrm{x}};\, 0;\, 0;\, 0;\, 0;\, 0], \tag{21}$$

where the $\widehat{\mathbf{s}}$ denotes the robot's state vector (given in Equation (74) in Appendix L.2), the $r_{\mathrm{x}}$ denotes the desired CoM x-velocity. The system state of the model (1) is expressed as $\mathbf{s} = \widehat{\mathbf{s}} - \widehat{\mathbf{s}}^*$.

The designs of model-based policy and reward appear in Appendix L.3 and the training details are presented in Appendix L.6 and Appendix L.7. To demonstrate the performance of trained Phy-DRL,

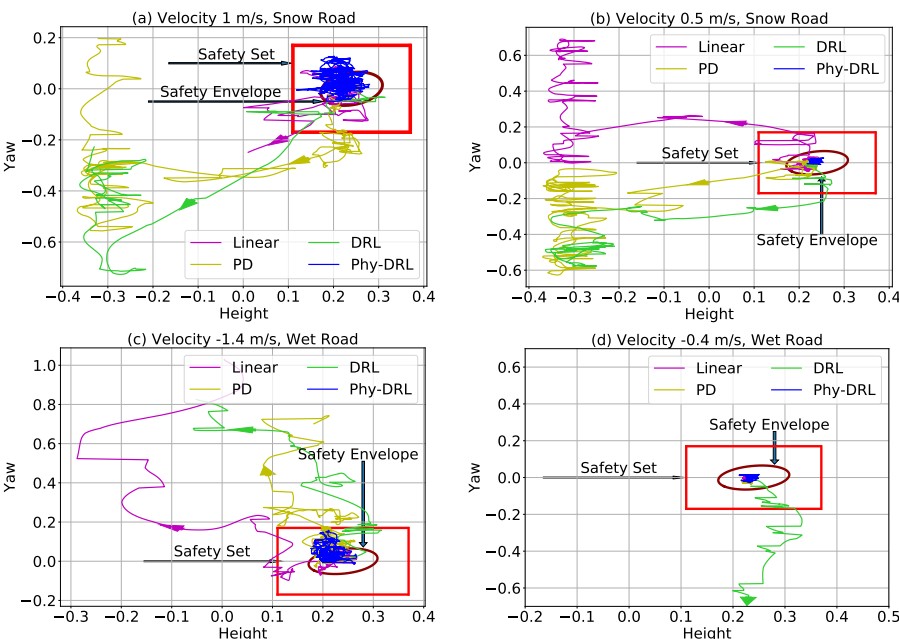

Figure 4: Phase plots of models running different environments, given different velocity commands.

we consider the comparisons of four policies:

– **Phy-DRL** policy, whose network configurations are summarized in the model **PKN-15** in Table 2 in Appendix L.6. Its total number of training steps is only $10^6$.

– **DRL** policy, denoting a purely data-driven action policy trained in standard DRL. Its network configurations are summarized in the model **FC MLP** in Table 2. Its reward for training is the CLF proposed in Westenbroek et al. (2022) and given in Equation (84), where the **P** is the same as the one in the Phy-DRL's safety-embedded reward. Its number of training steps is large as $10^7$.

– **PD** policy, denoting a default proportional-derivative controller developed in Da et al. (2021).

– **Linear** policy, which is the sole model-based action policy equation 4 used in Phy-DRL.

We compare the four policies in four testing environments: a) $r_x = 1$ m/s and snow road, b) $r_x = 0.5$ m/s and snow road, c) $r_x = -1.4$ m/s and wet road, and d) $r_x = -0.4$ m/s and wet road. The links to demonstration videos are available at Appendix L.9, with Figure 4 showing that Phy-DRL successfully constraints the robot's states to a safety set. Given more reasonable velocity commands in environments b) and d), Phy-DRL can also successfully constrain system states to the safety envelope. The Linear and PD policies can only constrain system states to a safety envelope in environment d). The DRL policy violates the safety requirements in all environments, which implies that purely data-driven DRL needs more training steps to search for a safe and robust policy. Meanwhile, Appendix L.4, Appendix L.6, and Appendix L.7 show that Phy-DRL features remarkably better velocity-regulation performance, fewer learning parameters, and fast and stable training.

## 8 CONCLUSION AND DISCUSSION

This paper proposes Phy-DRL: a physics-regulated deep reinforcement learning framework for safety-critical autonomous systems. Phy-DRL exhibits a mathematically provable safety guarantee. Compared with purely data-driven DRL and solely model-based design, Phy-DRL features fewer learning parameters and fast and stable training while offering enhanced safety assurance.

We recall the computation of matrix **P** (used in defining reward and safety envelope) depends on a linear model. However, if the linear model's mismatch is large, no safe policies may exist to render the safety envelope – defined by **P** – invariant. How to address safety concerns induced by faulty **P** constitutes our future research. We also note the derived condition of safety guarantee in Theorem 5.3 is not yet ready for a practical testing procedure due to the necessary full coverage testing within the domain $\Omega$. Transforming the theoretical safety conditions into practical and efficient ones will be another future research.

## 9 REPRODUCIBILITY STATEMENT

The code to reproduce our experimental results and supplementary materials are available at `https://github.com/HP-CAO/phy_rl`.

The experimental settings are described in Appendix K.5, Appendix K.6, and Appendix L.8.

## 10 ACKNOWLEDGEMENTS

We would like to first thank the anonymous reviewers for their helpful feedback, thoughtful reviews, and insightful comments. We appreciate Mirco Theile's helpful suggestions regarding the technical details, which inspire our future research directions. We also thank Yihao Cai for his help in deploying Phy-DRL on a physical quadruped robot.

This work was partly supported by the National Science Foundation under Grant CPS-2311084 and Grant CPS-2311085 and the Alexander von Humboldt Professorship Endowed by the German Federal Ministry of Education and Research.

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

# Appendices

# A    NOTATIONS THROUGHOUT PAPER

Table 1: Notation

| | |
|---|---|
| $\mathbb{R}^n$ | set of $n$-dimensional real vectors |
| $\mathbb{N}$ | set of natural numbers |
| $\text{len}(\mathbf{s})$ | length of vector $\mathbf{s}$ |
| $[\mathbf{x}]_i$ | $i$-th entry of vector $\mathbf{x}$ |
| $[\mathbf{x}]_{i:j}$ | a sub-vector formed by the $i$-th to $j$-th entries of vector $\mathbf{x}$ |
| $[\mathbf{W}]_{i,:}$ | $i$-th row of matrix $\mathbf{W}$ |
| $[\mathbf{W}]_{i,j}$ | element at row $i$ and column $j$ of matrix $\mathbf{W}$ |
| $\mathbf{P} \succ 0$ | matrix $\mathbf{P}$ is positive definite |
| $\mathbf{P} \prec 0$ | matrix $\mathbf{P}$ is negative definite |
| $\top$ | matrix or vector transposition |
| $\mathbf{I}_n$ | $n \times n$-dimensional identity matrix |
| $\mathbf{1}_n$ | $n$-dimensional vector of all ones |
| $\mathbf{0}_n$ | $n$-dimensional vector of all zeros |
| $\odot$ | Hadamard product |
| $[\mathbf{x} ; \mathbf{y}]$ | stacked (tall column) vector of vectors $\mathbf{x}$ and $\mathbf{y}$ |
| act | activation function |
| $\mathbf{O}_{m \times n}$ | $m \times n$-dimensional zero matrix |
| $\boxplus$ | an element in knowledge set |
| $*$ | a learning element |
| $\mathbb{X} \setminus \Omega$ | complement set of $\Omega$ with respect to $\mathbb{X}$ |

# B AUXILIARY LEMMAS

**Lemma B.1** (Schur Complement Zhang (2006)). *For any symmetric matrix* $\mathbf{M} = \begin{bmatrix} \mathbf{A} & \mathbf{B} \\ \mathbf{B}^\top & \mathbf{C} \end{bmatrix}$, *then* $\mathbf{M} \succ 0$ *if and only if* $\mathbf{C} \succ 0$ *and* $\mathbf{A} - \mathbf{B}\mathbf{C}^{-1}\mathbf{B}^\top \succ 0$.

**Lemma B.2.** *Corresponding to the set* $\mathbb{X}$ *defined in Equation* (2), *we define:*

$$\widehat{\mathbb{X}} \triangleq \left\{ \mathbf{s} \in \mathbb{R}^n \,|\, -\mathbf{1}_h \leq \mathbf{d} \leq \underline{\mathbf{D}} \cdot \mathbf{s}, \quad and \quad \overline{\mathbf{D}} \cdot \mathbf{s} \leq \mathbf{1}_h \right\}. \tag{22}$$

*The sets* $\mathbb{X} = \widehat{\mathbb{X}}$ *if and only if* $\overline{\mathbf{D}} = \frac{\mathbf{D}}{\overline{\Lambda}}$ *and* $\underline{\mathbf{D}} = \frac{\mathbf{D}}{\underline{\Lambda}}$, *where* $\mathbf{d}$, $\overline{\Lambda}$ *and* $\underline{\Lambda}$ *are defined in Lemma 5.1.*

*Proof.* The condition of set defined in Equation (2) is equivalent to

$$[\underline{\mathbf{v}} + \mathbf{v}]_i \leq [\mathbf{D}]_{i,:} \cdot \mathbf{s} \leq [\overline{\mathbf{v}} + \mathbf{v}]_i, \quad i \in \{1, 2, \ldots, h\}, \tag{23}$$

based on which, we consider three cases.

Case One: If $[\underline{\mathbf{v}} + \mathbf{v}]_i > 0$, we obtain from Equation (23) that $[\overline{\mathbf{v}} + \mathbf{v}]_i > 0$ as well, such that the Equation (23) can be rewritten equivalently as

$$\frac{[\mathbf{D}]_{i,:} \cdot \mathbf{s}}{[\overline{\mathbf{v}} + \mathbf{v}]_i} = \frac{[\mathbf{D}]_{i,:} \cdot \mathbf{s}}{[\overline{\Lambda}]_{i,i}} \leq 1, \text{ and } \frac{[\mathbf{D}]_{i,:} \cdot \mathbf{s}}{[\underline{\mathbf{v}} + \mathbf{v}]_i} = \frac{[\mathbf{D}]_{i,:} \cdot \mathbf{s}}{[\underline{\Lambda}]_{i,i}} \geq 1 = [\mathbf{d}]_i, \ i \in \{1, 2, \ldots, h\}, \tag{24}$$

which is obtained via considering the second items of $[\overline{\Lambda}]_{i,j}$ and $[\underline{\Lambda}]_{i,j}$ and the first item of $[\mathbf{d}]_i$, presented in Lemma 5.1.

Case Two: If $[\overline{\mathbf{v}} + \mathbf{v}]_i < 0$, we obtain from Equation (23) that $[\underline{\mathbf{v}} + \mathbf{v}]_i < 0$ as well, such that the Equation (23) can be rewritten equivalently as

$$\frac{[\mathbf{D}]_{i,:} \cdot \mathbf{s}}{[\underline{\mathbf{v}} + \mathbf{v}]_i} = \frac{[\mathbf{D}]_{i,:} \cdot \mathbf{s}}{[\overline{\Lambda}]_{i,i}} \leq 1, \text{ and } \frac{[\mathbf{D}]_{i,:} \cdot \mathbf{s}}{[\overline{\mathbf{v}} + \mathbf{v}]_i} = \frac{[\mathbf{D}]_{i,:} \mathbf{s}}{[\underline{\Lambda}]_{i,i}} \geq 1 = [\mathbf{d}]_i, \ i \in \{1, 2, \ldots, h\}, \tag{25}$$

which is obtained via considering the third items of $[\overline{\Lambda}]_{i,j}$ and $[\underline{\Lambda}]_{i,j}$ and the second item of $[\mathbf{d}]_i$, presented in Lemma 5.1.

Case Three: If $[\overline{\mathbf{v}} + \mathbf{v}]_i > 0$ and $[\underline{\mathbf{v}} + \mathbf{v}]_i < 0$, the Equation (23) can be rewritten equivalently as

$$\frac{[\mathbf{D}]_{i,:} \cdot \mathbf{s}}{[\overline{\mathbf{v}} + \mathbf{v}]_i} = \frac{[\mathbf{D}]_{i,:} \mathbf{s}}{[\overline{\Lambda}]_{i,i}} \leq 1, \text{ and } \frac{[\mathbf{D}]_{i,:} \cdot \mathbf{s}}{[-\underline{\mathbf{v}} - \mathbf{v}]_i} = \frac{[\mathbf{D}]_{i,:} \cdot \mathbf{s}}{[\underline{\Lambda}]_{i,i}} \geq -1 = [\mathbf{d}]_i, \ i \in \{1, 2, \ldots, h\}, \tag{26}$$

which is obtained via considering the fourth items of $[\overline{\Lambda}]_{i,j}$ and $[\underline{\Lambda}]_{i,j}$ and the third item of $[\mathbf{d}]_i$, presented in Lemma 5.1.

We note from the first items of $[\overline{\Lambda}]_{i,j}$ and $[\underline{\Lambda}]_{i,j}$ in Lemma 5.1 that the defined $\overline{\Lambda}$ and $\underline{\Lambda}$ are diagonal matrices. The conjunctive results (23)–(26) can thus be equivalent described by

$$\frac{\mathbf{D} \cdot \mathbf{s}}{\overline{\Lambda}} \leq \mathbf{1}_h \text{ and } \frac{\mathbf{D} \cdot \mathbf{s}}{\underline{\Lambda}} \geq \mathbf{d} \geq -\mathbf{1}_h,$$

substituting $\overline{\mathbf{D}} = \frac{\mathbf{D}}{\overline{\Lambda}}$ and $\underline{\mathbf{D}} = \frac{\mathbf{D}}{\underline{\Lambda}}$ into which, we obtain $-\mathbf{1}_h \leq \mathbf{d} \leq \underline{\mathbf{D}} \cdot \mathbf{s}$, and $\overline{\mathbf{D}} \cdot \mathbf{s} \leq \mathbf{1}_h$, which is the condition for defining the set $\widehat{\mathbb{X}}$ in Equation (22). We thus conclude the statement. $\qquad\square$

## C  PROOF OF LEMMA 5.1

In light of Lemma B.2 in Appendix B, we have $\mathbb{X} = \widehat{\mathbb{X}}$. Therefore, to prove $\Omega \subseteq \mathbb{X}$, we consider the proof of $\Omega \subseteq \widehat{\mathbb{X}}$, which is carried out below. The $\widehat{\mathbb{X}}$ defined in Equation (22) relies on two conjunctive conditions: $-\mathbf{1}_h \leq \mathbf{d} \leq \underline{\mathbf{D}} \cdot \mathbf{s}$ and $\overline{\mathbf{D}} \cdot \mathbf{s} \leq \mathbf{1}_h$, based on which the proof is separated into two cases.

**Case One:** $\overline{\mathbf{D}} \cdot \mathbf{s} \leq \mathbf{1}_h$, which can be rewritten as $[\overline{\mathbf{D}} \cdot \mathbf{s}]_i \leq 1$, $i \in \{1, 2, \ldots, h\}$.

We next prove that $\max_{\mathbf{s} \in \Omega} \left\{ [\overline{\mathbf{D}} \cdot \mathbf{s}]_i \right\} = \sqrt{\left[\overline{\mathbf{D}} \cdot \mathbf{P}^{-1} \cdot \overline{\mathbf{D}}^\top\right]_{i,i}}$, $i \in \{1, 2, \ldots, h\}$. To achieve this, let us consider the constrained optimization problem:

$$\max \left\{ [\overline{\mathbf{D}} \cdot \mathbf{s}]_i \right\}, \quad \text{subject to } \mathbf{s}^\top \cdot \mathbf{P} \cdot \mathbf{s} \leq 1, \quad i \in \{1, 2, \ldots, h\}.$$

Let $\mathbf{s}^*$ be the optimal solution. Then, according to the Kuhn-Tucker conditions, we have

$$\left[\overline{\mathbf{D}}\right]_{i,:}^\top - 2\lambda \cdot \mathbf{P} \cdot \mathbf{s}^* = 0, \quad \lambda \cdot (1 - (\mathbf{s}^*)^\top \cdot \mathbf{P} \cdot \mathbf{s}^*) = 0,$$

which, in conjunction with $\lambda > 0$, lead to

$$2\lambda \cdot \mathbf{P} \cdot \mathbf{s}^* = \left[\overline{\mathbf{D}}\right]_{i,:}^\top, \tag{27}$$

$$(\mathbf{s}^*)^\top \cdot \mathbf{P} \cdot \mathbf{s}^* = 1. \tag{28}$$

Multiplying both left-hand sides of Equation (27) by $(\mathbf{s}^*)^\top$ yields $2\lambda \cdot (\mathbf{s}^*)^\top \cdot \mathbf{P} \cdot \mathbf{s}^* = (\mathbf{s}^*)^\top \cdot \left[\overline{\mathbf{D}}\right]_{i,:}^\top$, which, in conjunction with Equation (28), results in

$$2\lambda = (\mathbf{s}^*)^\top \cdot \left[\overline{\mathbf{D}}\right]_{i,:}^\top > 0. \tag{29}$$

Multiplying both left-hand sides of Equation (27) by $\mathbf{P}^{-1}$ leads to $2\lambda \cdot \mathbf{s}^* = \mathbf{P}^{-1} \cdot \left[\overline{\mathbf{D}}\right]_{i,:}^\top$, multiplying both left-hand sides of which by $\left[\overline{\mathbf{D}}\right]_{i,:}$, we arrive in

$$2\lambda \cdot \left[\overline{\mathbf{D}}\right]_{i,:} \cdot \mathbf{s}^* = \left[\overline{\mathbf{D}}\right]_{i,:} \cdot \mathbf{P}^{-1} \cdot \left[\overline{\mathbf{D}}\right]_{i,:}^\top. \tag{30}$$

Substituting Equation (29) into Equation (30), we obtain $(\mathbf{s}^*)^\top \cdot \left[\overline{\mathbf{D}}\right]_{i,:}^\top \cdot \left[\overline{\mathbf{D}}\right]_{i,:} \cdot \mathbf{s}^* = \left[\overline{\mathbf{D}}\right]_{i,:} \cdot \mathbf{P}^{-1} \cdot \left[\overline{\mathbf{D}}\right]_{i,:}^\top$, from which we can have $(\mathbf{s}^*)^\top \cdot \left[\overline{\mathbf{D}}\right]_{i,:}^\top = \sqrt{\left[\overline{\mathbf{D}}\right]_{i,:} \cdot \mathbf{P}^{-1} \cdot \left[\overline{\mathbf{D}}\right]_{i,:}^\top} > 0$, which with Equation (29) indicate:

$$2\lambda = \sqrt{\left[\overline{\mathbf{D}}\right]_{i,:} \cdot \mathbf{P}^{-1} \cdot \left[\overline{\mathbf{D}}\right]_{i,:}^\top}. \tag{31}$$

We note that the Equation (27) is equivalent to $\mathbf{s}^* = \frac{1}{2\lambda} \cdot \mathbf{P}^{-1} \cdot \left[\overline{\mathbf{D}}\right]_{i,:}^\top$, substituting Equation (31) into which results in $\mathbf{s}^* = \frac{1}{\sqrt{\left[\overline{\mathbf{D}}\right]_{i,:} \cdot \mathbf{P}^{-1} \cdot \left[\overline{\mathbf{D}}\right]_{i,:}^\top}} \cdot \mathbf{P}^{-1} \cdot \left[\overline{\mathbf{D}}\right]_{i,:}^\top$, multiplying both sides of which by $\left[\overline{\mathbf{D}}\right]_{i,:}$ means

$$\max_{\mathbf{s} \in \Omega} \left\{ [\overline{\mathbf{D}} \cdot \mathbf{s}]_i \right\} = \max_{\mathbf{s} \in \Omega} \left\{ \left[\overline{\mathbf{D}}\right]_{i,:} \cdot \mathbf{s} \right\} = \left[\overline{\mathbf{D}}\right]_{i,:} \cdot \mathbf{s}^* = \frac{\left[\overline{\mathbf{D}}\right]_{i,:} \cdot \mathbf{P}^{-1} \cdot \left[\overline{\mathbf{D}}\right]_{i,:}^\top}{\sqrt{\left[\overline{\mathbf{D}}\right]_{i,:} \cdot \mathbf{P}^{-1} \cdot \left[\overline{\mathbf{D}}\right]_{i,:}^\top}}$$
$$= \sqrt{\left[\overline{\mathbf{D}}\right]_{i,:} \cdot \mathbf{P}^{-1} \cdot \left[\overline{\mathbf{D}}^\top\right]_{:,i}}, \quad i \in \{1, 2, \ldots, h\}$$

which means

$$\max_{\mathbf{s} \in \Omega} \left\{ [\overline{\mathbf{D}} \cdot \mathbf{s}]_i \right\} \leq 1 \text{ if and only if } \left[\overline{\mathbf{D}}\right]_{i,:} \cdot \mathbf{P}^{-1} \cdot \left[\overline{\mathbf{D}}^\top\right]_{:,i} \leq 1, \quad i \in \{1, 2, \ldots, h\}, \tag{32}$$

which further implies that

$$\overline{\mathbf{D}} \cdot \mathbf{s} \leq \mathbf{1}_h, \quad \text{if } \left[\overline{\mathbf{D}}\right]_{i,:} \cdot \mathbf{P}^{-1} \cdot \left[\overline{\mathbf{D}}^\top\right]_{:,i} \leq 1, \ i \in \{1, 2, \ldots, h\}. \tag{33}$$

**Case Two:** $-\mathbf{1}_h \leq \mathbf{d} \leq \underline{\mathbf{D}} \cdot \mathbf{s}$, which includes two scenarios: $[\mathbf{d}]_i = 1$ and $[\mathbf{d}]_i = -1$, referring to $\mathbf{d}$ in Lemma 5.1. We first consider $[\underline{\mathbf{D}} \cdot \mathbf{s}]_i \geq -1$, which can be rewritten as $[\widehat{\underline{\mathbf{D}}} \cdot \mathbf{s}]_i \leq -[\mathbf{d}]_i = 1, i \in \{1, 2, \ldots, h\}$, with $\widehat{\underline{\mathbf{D}}} = -\underline{\mathbf{D}}$. Following the same steps to derive Equation (32), we obtain

$$\max_{\mathbf{s} \in \Omega} \left\{ \left[ \widehat{\underline{\mathbf{D}}} \cdot \mathbf{s} \right]_i \right\} \leq 1 \text{ if and only if } \left[ \widehat{\underline{\mathbf{D}}} \right]_{i,:} \cdot \mathbf{P}^{-1} \cdot \left[ \widehat{\underline{\mathbf{D}}}^\top \right]_{:,i} < 1, \ \ i \in \{1, 2, \ldots, h\},$$

which with $\widehat{\underline{\mathbf{D}}} = -\underline{\mathbf{D}}$ indicate that

$$\min_{\mathbf{s} \in \Omega} \left\{ [\underline{\mathbf{D}} \cdot \mathbf{s}]_i \right\} \geq -1 \text{ iff } [\underline{\mathbf{D}}]_{i,:} \cdot \mathbf{P}^{-1} \cdot \left[ \underline{\mathbf{D}}^\top \right]_{:,i} < 1, \ \ [\mathbf{d}]_i = -1, \ \ i \in \{1, 2, \ldots, h\}. \tag{34}$$

We next consider $\underline{[\mathbf{d}]_i = 1}$. We prove in this scenario, $\min_{\mathbf{s} \in \Omega} \left\{ [\underline{\mathbf{D}} \cdot \mathbf{s}]_i \right\} = \sqrt{\left[ \underline{\mathbf{D}} \cdot \mathbf{P}^{-1} \cdot \underline{\mathbf{D}}^\top \right]_{i,i}}$. To achieve this, let us consider the constrained optimization problem:

$$\min \left\{ [\underline{\mathbf{D}} \cdot \mathbf{s}]_i \right\}, \ \ \text{subject to } \mathbf{s}^\top \cdot \mathbf{P} \cdot \mathbf{s} \leq 1.$$

Let $\hat{\mathbf{s}}^*$ be the optimal solution. Then, according to the Kuhn-Tucker conditions, we have

$$[\underline{\mathbf{D}}]_{i,:}^\top + 2\hat{\lambda} \cdot \mathbf{P} \cdot \hat{\mathbf{s}}^* = 0, \quad \hat{\lambda} \cdot (1 - (\hat{\mathbf{s}}^*)^\top \cdot \mathbf{P} \cdot \hat{\mathbf{s}}^*) = 0,$$

which, in conjunction with $\hat{\lambda} < 0$, leads to

$$2\hat{\lambda} \cdot \mathbf{P} \cdot \hat{\mathbf{s}}^* = - [\underline{\mathbf{D}}]_{i,:}^\top, \tag{35}$$

$$(\hat{\mathbf{s}}^*)^\top \cdot \mathbf{P} \cdot \hat{\mathbf{s}}^* = 1. \tag{36}$$

Multiplying both left-hand sides of Equation (35) by $(\hat{\mathbf{s}}^*)^\top$ yields $2\hat{\lambda} \cdot (\hat{\mathbf{s}}^*)^\top \cdot \mathbf{P} \cdot \hat{\mathbf{s}}^* = -(\hat{\mathbf{s}}^*)^\top \cdot [\underline{\mathbf{D}}]_{i,:}^\top$, which, in conjunction with Equation (36), results in

$$2\hat{\lambda} = -(\hat{\mathbf{s}}^*)^\top \cdot [\underline{\mathbf{D}}]_{i,:}^\top < 0. \tag{37}$$

Multiplying both left-hand sides of Equation (35) by $\mathbf{P}^{-1}$ leads to $2\hat{\lambda} \cdot \hat{\mathbf{s}}^* = -\mathbf{P}^{-1} \cdot [\underline{\mathbf{D}}]_{i,:}^\top$, multiplying both left-hand sides of which by $[\underline{\mathbf{D}}]_{i,:}$, we arrive in

$$2\hat{\lambda} \cdot [\underline{\mathbf{D}}]_{i,:} \cdot \hat{\mathbf{s}}^* = - [\underline{\mathbf{D}}]_{i,:} \cdot \mathbf{P}^{-1} \cdot [\underline{\mathbf{D}}]_{i,:}^\top. \tag{38}$$

Substituting Equation (37) into Equation (38), we obtain $-(\hat{\mathbf{s}}^*)^\top \cdot [\underline{\mathbf{D}}]_{i,:}^\top \cdot [\underline{\mathbf{D}}]_{i,:} \hat{\mathbf{s}}^* = - [\underline{\mathbf{D}}]_{i,:} \cdot \mathbf{P}^{-1} \cdot [\underline{\mathbf{D}}]_{i,:}^\top$, which together with Equation (37) indicate the solution:

$$2\hat{\lambda} = -(\hat{\mathbf{s}}^*)^\top \cdot [\underline{\mathbf{D}}]_{i,:}^\top = -\sqrt{[\underline{\mathbf{D}}]_{i,:} \cdot \mathbf{P}^{-1} \cdot [\underline{\mathbf{D}}]_{i,:}^\top}. \tag{39}$$

We note that the Equation (35) is equivalent to $\hat{\mathbf{s}}^* = -\frac{1}{2\hat{\lambda}} \cdot \mathbf{P}^{-1} \cdot [\underline{\mathbf{D}}]_{i,:}^\top$, substituting Equation (39) into which results in $\hat{\mathbf{s}}^* = \frac{1}{\sqrt{[\underline{\mathbf{D}}]_{i,:} \cdot \mathbf{P}^{-1} \cdot [\underline{\mathbf{D}}]_{i,:}^\top}} \mathbf{P}^{-1} \cdot [\underline{\mathbf{D}}]_{i,:}^\top$, multiplying both sides of which by $[\underline{\mathbf{D}}]_{i,:}$ means

$$\min_{\mathbf{s} \in \Omega} \left\{ [\underline{\mathbf{D}} \cdot \mathbf{s}]_i \right\} = \min_{\mathbf{s} \in \Omega} \left\{ [\underline{\mathbf{D}}]_{i,:} \cdot \mathbf{s} \right\} = [\underline{\mathbf{D}}]_{i,:} \cdot \hat{\mathbf{s}}^* = \frac{[\underline{\mathbf{D}}]_{i,:} \cdot \mathbf{P}^{-1} [\underline{\mathbf{D}}]_{i,:}^\top}{\sqrt{[\underline{\mathbf{D}}]_{i,:} \cdot \mathbf{P}^{-1} \cdot [\underline{\mathbf{D}}]_{i,:}^\top}}$$

$$= \sqrt{[\underline{\mathbf{D}}]_{i,:} \cdot \mathbf{P}^{-1} \cdot \left[ \underline{\mathbf{D}}^\top \right]_{:,i}}, \quad i \in \{1, 2, \ldots, h\},$$

which means

$$\min_{\mathbf{s} \in \Omega} \left\{ [\underline{\mathbf{D}} \cdot \mathbf{s}]_i \right\} \geq 1 \text{ iff } [\underline{\mathbf{D}}]_{i,:} \cdot \mathbf{P}^{-1} \cdot \left[ \underline{\mathbf{D}}^\top \right]_{:,i} \geq 1, \ \ [\mathbf{d}]_i = 1, \quad i \in \{1, 2, \ldots, h\}. \tag{40}$$

Summarizing Equation (34) and Equation (40), with the consideration of $\mathbf{d}$ presented in Lemma 5.1, we conclude that

$$\min_{\mathbf{s} \in \Omega} \left\{ [\underline{\mathbf{D}} \cdot \mathbf{s}]_i \right\} \geq [\mathbf{d}]_i \ \ \text{iff} \ \ [\underline{\mathbf{D}}]_{i,:} \cdot \mathbf{P}^{-1} \cdot \left[ \underline{\mathbf{D}}^\top \right]_{:,i} = \begin{cases} \geq 1, & \text{if } [\mathbf{d}]_i = 1 \\ \leq 1, & \text{if } [\mathbf{d}]_i = -1 \end{cases}, \ \ i \in \{1, 2, \ldots, h\},$$

which, in conjunction with the fact that $\underline{\mathbf{D}} \cdot \mathbf{s}$ and $\mathbf{d}$ are vectors, implies that

$$\underline{\mathbf{D}} \cdot \mathbf{s} \geq \mathbf{d} \geq -\mathbf{1}_h, \text{ if } [\underline{\mathbf{D}}]_{i,:} \cdot \mathbf{P}^{-1} \cdot \left[\underline{\mathbf{D}}^\top\right]_{:,i} = \begin{cases} \geq 1, & \text{if } [\mathbf{d}]_i = 1 \\ \leq 1, & \text{if } [\mathbf{d}]_i = -1 \end{cases}, \; i \in \{1, 2, \ldots, h\}. \quad (41)$$

We now conclude from Equation (33) and Equation (41) that $\overline{\mathbf{D}} \cdot \mathbf{s} \leq \mathbf{1}_h$ and $\underline{\mathbf{D}} \cdot \mathbf{s} \geq \mathbf{d} \geq -\mathbf{1}_h$ hold, if for any $i \in \{1, 2, \ldots, h\}$,

$$\left[\overline{\mathbf{D}}\right]_{i,:} \cdot \mathbf{P}^{-1} \cdot \left[\overline{\mathbf{D}}^\top\right]_{:,i} \leq 1 \; \text{ and } \; [\underline{\mathbf{D}}]_{i,:} \cdot \mathbf{P}^{-1} \cdot \left[\underline{\mathbf{D}}^\top\right]_{:,i} = \begin{cases} \geq 1, & \text{if } [\mathbf{d}]_i = 1 \\ \leq 1, & \text{if } [\mathbf{d}]_i = -1. \end{cases}$$

Meanwhile, noticing the $\overline{\mathbf{D}} \cdot \mathbf{s} \leq \mathbf{1}_h$ and $\underline{\mathbf{D}} \cdot \mathbf{s} \geq \mathbf{d} \geq -\mathbf{1}_h$ is the condition of forming the set $\widehat{\mathbb{X}}$ in Equation (22), we can finally obtain Equation (7).

# D   THEOREM 5.3

## D.1   PROOF OF THEOREM 5.3

We note that $\mathbf{Q} = \mathbf{Q}^\top$ and the $\mathbf{F} = \mathbf{R} \cdot \mathbf{Q}^{-1}$ is equivalent to the $\mathbf{F} \cdot \mathbf{Q} = \mathbf{R}$, substituting which into Equation (11) yields

$$\begin{bmatrix} \alpha \cdot \mathbf{Q} & \mathbf{Q} \cdot (\mathbf{A} + \mathbf{B} \cdot \mathbf{F})^\top \\ (\mathbf{A} + \mathbf{B} \cdot \mathbf{F}) \cdot \mathbf{Q} & \mathbf{Q} \end{bmatrix} \succ 0. \tag{42}$$

We note Equation (42) implies $\alpha > 0$ and $\mathbf{Q} \succ 0$. Then, according to the auxiliary Lemma B.1 in Appendix B, we have

$$\alpha \cdot \mathbf{Q} - \mathbf{Q} \cdot (\mathbf{A} + \mathbf{B} \cdot \mathbf{F})^\top \cdot \mathbf{Q}^{-1} \cdot (\mathbf{A} + \mathbf{B} \cdot \mathbf{F}) \cdot \mathbf{Q} \succ 0. \tag{43}$$

Since $\mathbf{P} = \mathbf{Q}^{-1}$, multiplying both left-hand and right-hand sides of Equation (43) by $\mathbf{P}$ we obtain

$$\alpha \cdot \mathbf{P} - (\mathbf{A} + \mathbf{B} \cdot \mathbf{F})^\top \cdot \mathbf{P} \cdot (\mathbf{A} + \mathbf{B} \cdot \mathbf{F}) \succ 0,$$

which, in conjunction with $\overline{\mathbf{A}}$ defined in Equation (9), leads to

$$\alpha \mathbf{P} - \overline{\mathbf{A}}^\top \cdot \mathbf{P} \cdot \overline{\mathbf{A}} \succ 0. \tag{44}$$

We now define a function:

$$V(\mathbf{s}(k)) = \mathbf{s}^\top(k) \cdot \mathbf{P} \cdot \mathbf{s}(k). \tag{45}$$

With the consideration of function in Equation (45), along the real plant (1) with Equation (3) and Equation (4), we have

$$V(\mathbf{s}(k+1))$$
$$= (\mathbf{B} \cdot \mathbf{a}_{\mathrm{drl}}(k) + \mathbf{f}(\mathbf{s}(k), \mathbf{a}(k)))^\top \cdot \mathbf{P} \cdot (\mathbf{B} \cdot \mathbf{a}_{\mathrm{drl}}(k) + \mathbf{f}(\mathbf{s}(k), \mathbf{a}(k)))$$
$$+ 2\mathbf{s}^\top(k) \cdot \overline{\mathbf{A}}^\top \cdot \mathbf{P} \cdot (\mathbf{B} \cdot \mathbf{a}_{\mathrm{drl}}(k) + \mathbf{f}(\mathbf{s}(k), \mathbf{a}(k))) + \mathbf{s}^\top(k) \cdot (\overline{\mathbf{A}}^\top \cdot \mathbf{P} \cdot \overline{\mathbf{A}}) \cdot \mathbf{s}(k) \tag{46}$$
$$< (\mathbf{B} \cdot \mathbf{a}_{\mathrm{drl}}(k) + \mathbf{f}(\mathbf{s}(k), \mathbf{a}(k)))^\top \cdot \mathbf{P} \cdot (\mathbf{B} \cdot \mathbf{a}_{\mathrm{drl}}(k) + \mathbf{f}(\mathbf{s}(k), \mathbf{a}(k)))$$
$$+ 2\mathbf{s}^\top(k) \cdot \overline{\mathbf{A}}^\top \cdot \mathbf{P} \cdot (\mathbf{B} \cdot \mathbf{a}_{\mathrm{drl}}(k) + \mathbf{f}(\mathbf{s}(k), \mathbf{a}(k))) + \alpha \cdot V(\mathbf{s}(k)), \tag{47}$$

where the Equation (47) is obtained from its previous step via considering Equation (45) and Equation (44). Observing Equation (46), we obtain

$$-r(\mathbf{s}(k), \mathbf{s}(k+1)) = V(\mathbf{s}(k+1)) - \mathbf{s}^\top(k) \cdot (\overline{\mathbf{A}}^\top \cdot \mathbf{P} \cdot \overline{\mathbf{A}}) \cdot \mathbf{s}(k)$$
$$= (\mathbf{B} \cdot \mathbf{a}_{\mathrm{drl}}(k) + \mathbf{f}(\mathbf{s}(k), \mathbf{a}(k)))^\top \cdot \mathbf{P} \cdot (\mathbf{B} \cdot \mathbf{a}_{\mathrm{drl}}(k) + \mathbf{f}(\mathbf{s}(k), \mathbf{a}(k)))$$
$$+ 2\mathbf{s}^\top(k) \cdot \overline{\mathbf{A}}^\top \cdot \mathbf{P} \cdot (\mathbf{B} \cdot \mathbf{a}_{\mathrm{drl}}(k) + \mathbf{f}(\mathbf{s}(k), \mathbf{a}(k))) \tag{48}$$

where $r(\mathbf{s}(k), \mathbf{s}(k+1))$ is defined in Equation (8). Substituting Equation (48) into Equation (47) yields

$$V(\mathbf{s}(k+1)) < \alpha \cdot V(\mathbf{s}(k)) - r(\mathbf{s}(k), \mathbf{s}(k+1)),$$

which further implies that

$$V(\mathbf{s}(k+1)) - V(\mathbf{s}(k)) < (\alpha - 1) \cdot V(\mathbf{s}(k)) - r(\mathbf{s}(k), \mathbf{s}(k+1)). \tag{49}$$

Since $0 < \alpha < 1$, we have $\alpha - 1 < 0$. So, the $(\alpha - 1) \cdot V(\mathbf{s}(k)) - r(\mathbf{s}(k), \mathbf{s}(k+1)) \geq 0$ means $V(\mathbf{s}(k)) \leq \frac{r(\mathbf{s}(k), \mathbf{s}(k+1))}{\alpha - 1}$. Therefore, if $\frac{r(\mathbf{s}(k), \mathbf{s}(k+1))}{\alpha - 1} \leq 1$, we have

$$V(\mathbf{s}(k)) \leq 1, \quad \text{and} \quad (\alpha - 1) \cdot V(\mathbf{s}(k)) - r(\mathbf{s}(k), \mathbf{s}(k+1)) \geq 0, \tag{50}$$

where the second inequality, in conjunction with Equation (49), implies that there exists a scalar $\theta$ such that

$$V(\mathbf{s}(k+1)) - V(\mathbf{s}(k)) < \theta, \quad \text{with} \quad \theta > 0. \tag{51}$$

The result in Equation (51) can guarantee the safety of real plants. To prove this, let's consider the worst-case scenario that the $V(\mathbf{s}(k))$ is strictly increasing with respect to time $k \in \mathbb{N}$. So, starting from system state $\mathbf{s}(k)$ satisfying $V(\mathbf{s}(k)) \leq \frac{r(\mathbf{s}(k),\mathbf{s}(k+1))}{\alpha-1} \leq 1$, $V(\mathbf{s}(k))$ will increase to $V(\mathbf{s}(q)) = \frac{r(\mathbf{s}(q),\mathbf{s}(q+1))}{\alpha-1} \leq 1$, where $q > k \in \mathbb{N}$. Meanwhile, we note that the $(\alpha-1) \cdot V(\mathbf{s}(k)) - r(\mathbf{s}(k),\mathbf{s}(k+1)) \leq 0$ is equivalent to $V(\mathbf{s}(k)) \geq \frac{r(\mathbf{s}(k),\mathbf{s}(k+1))}{\alpha-1}$, and also in conjunction with Equation (49) implies that $V(\mathbf{s}(k+1)) - V(\mathbf{s}(k)) < 0$. These mean that if $V(\mathbf{s}(q)) = \frac{r(\mathbf{s}(q),\mathbf{s}(q+1))}{\alpha-1} \leq 1$ is achieved, the $V(\mathbf{s}(k))$ will start decreasing [1]. We thus conclude here that in the worst-case scenario, if starting from a point not larger than $\frac{r(\mathbf{s}(k),\mathbf{s}(k+1))}{\alpha-1}$, i.e., $V(\mathbf{s}(k)) \leq \frac{r(\mathbf{s}(k),\mathbf{s}(k+1))}{\alpha-1} \leq 1$, we have

$$V(\mathbf{s}(k)) \leq 1, \quad \forall k \in \mathbb{N}. \tag{52}$$

We now consider the other case, i.e., $1 \geq V(\mathbf{s}(k)) > \frac{r(\mathbf{s}(k),\mathbf{s}(k+1))}{\alpha-1}$. Recalling that in this case $V(\mathbf{s}(k+1)) - V(\mathbf{s}(k)) < 0$, which means $V(\mathbf{s}(k))$ is strictly decreasing with respect to time $k$, until $V(\mathbf{s}(q)) \leq \frac{r(\mathbf{s}(q),\mathbf{s}(q+1))}{\alpha-1} < 1$, $q > k \in \mathbb{N}$. Then, following the same analysis path of the worst case, we can conclude Equation (52) consequently. In other words,

$$V(\mathbf{s}(k)) \leq 1, \forall k \in \mathbb{N}, \quad \text{if } V(\mathbf{s}(1)) \leq 1,$$

which, in conjunction with Equation (6) and Equation (45), lead to

$$\mathbf{s}(k) \in \Omega, \forall k \in \mathbb{N}, \quad \text{if } \mathbf{s}(1) \in \Omega. \tag{53}$$

This proof path is illustrated in Figure 5.

Finally, we note from Lemma 5.1 that the condition in Equation (7) is to guarantee that $\Omega \subseteq \mathbb{X}$, which with Equation (53) result in $\mathbf{s}(k) \in \Omega \subseteq \mathbb{X}, \forall k \in \mathbb{N}$, if $\mathbf{s}(1) \in \Omega$, which completes the proof.

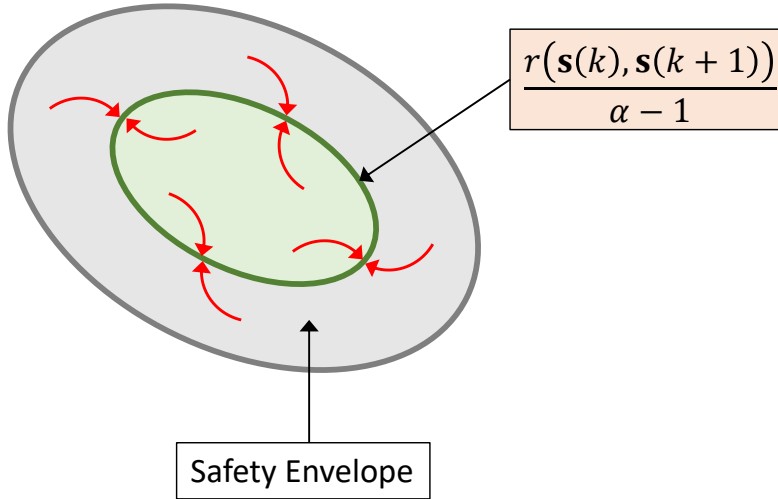

Figure 5: Illustration of the proof path of Theorem 5.3.

## D.2 Extension: Explanation of Fast Training Toward Safety Guarantee

The experimental results demonstrate that compared with purely data-driven DRL, our proposed Phy-DRL has much faster training toward safety guarantee, which can be explained by Equation (44),

---

[1] In practice, the control command at one control period should not drive the system to escape the safety envelop $\Omega$ from $V(\mathbf{s}(q)) = \frac{r(\mathbf{s}(q),\mathbf{s}(q+1))}{\alpha-1}$.

Equation (45), and Equation (46). Specifically, because of them and $0 < \alpha < 1$, we have

$$
\begin{aligned}
&V(\mathbf{s}(k+1)) - V(\mathbf{s}(k)) \\
&\leq V(\mathbf{s}(k+1)) - \alpha \cdot V(\mathbf{s}(k)) \\
&= (\mathbf{B} \cdot \mathbf{a}_{\mathrm{drl}}(k) + \mathbf{f}(\mathbf{s}(k), \mathbf{a}(k)))^\top \cdot \mathbf{P} \cdot (\mathbf{B} \cdot \mathbf{a}_{\mathrm{drl}}(k) + \mathbf{f}(\mathbf{s}(k), \mathbf{a}(k))) \\
&\quad + 2\mathbf{s}^\top(k) \cdot \overline{\mathbf{A}}^\top \cdot \mathbf{P} \cdot (\mathbf{B} \cdot \mathbf{a}_{\mathrm{drl}}(k) + \mathbf{f}(\mathbf{s}(k), \mathbf{a}(k))) + \mathbf{s}^\top(k) \cdot (\overline{\mathbf{A}}^\top \cdot \mathbf{P} \cdot \overline{\mathbf{A}} - \mathbf{P}) \cdot \mathbf{s}(k).
\end{aligned} \tag{54}
$$

The (off-line designed) model-based property in Equation (44) implies that Equation (54) has a constant negative term, i.e., $\mathbf{s}^\top(k) \cdot (\overline{\mathbf{A}}^\top \cdot \mathbf{P} \cdot \overline{\mathbf{A}} - \alpha \cdot \mathbf{P}) \cdot \mathbf{s}(k) < 0$. As depicted by Figure 6, the $\mathbf{s}^\top(k) \cdot (\overline{\mathbf{A}}^\top \cdot \mathbf{P} \cdot \overline{\mathbf{A}} - \alpha \cdot \mathbf{P}) \cdot \mathbf{s}(k) < 0$ can be understood that it puts a global attractor insides safety envelope, which generate attracting force toward safety envelope. Obviously, because of the always-existing attracting force, system states under the control of Phy-DRL are more likely and quickly to stay inside the safety envelope, compared with purely data-driven DRL frameworks.

In summary, the driving factor of Phy-DRL's fast training toward safety guarantee is the concurrent safety-embedded reward and residual action policy.

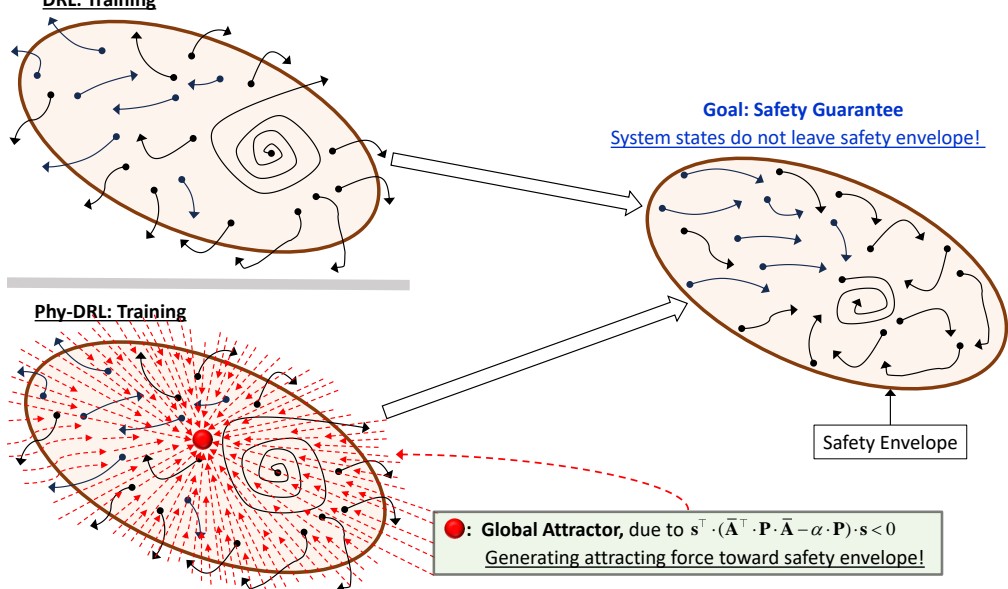

Figure 6: Explanation of Phy-DRL's fast training: the root reason is the (off-line designed) model-based property in Equation (44), which generates always-existing attracting force toward the safety envelope.

# E EXTENSION: MATHEMATICALLY-PROVABLE SAFETY AND STABILITY GUARANTEES

We first present the definition of a stability guarantee.

**Definition E.1.** The real plant (1) is said to be stability guaranteed, if given any $\mathbf{s}(1) \in \mathbb{R}^n$, then $\lim_{k \to \infty} \mathbf{s}(k) = \mathbf{0}_n$.

The relation between safety guarantee and stability guarantee is presented in Appendix E.1.

## E.1 SAFETY VERSUS STABILITY

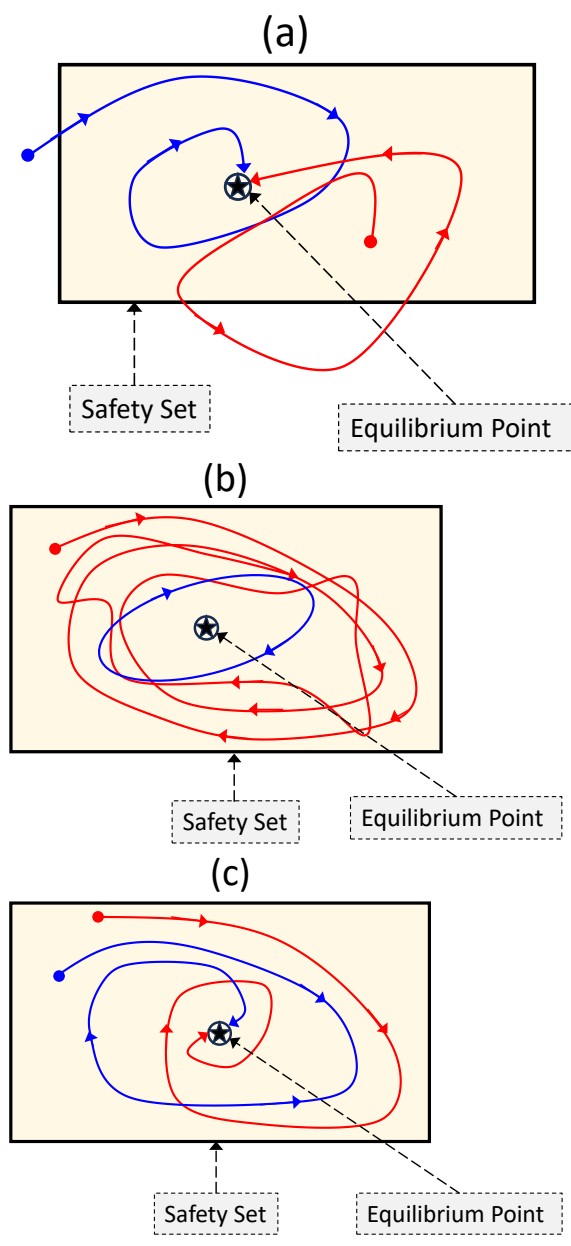

Figure 7: Explanations of safety and stability via phase plots: (a) only stability is guaranteed, (b) only safety is guaranteed, (c) both safety and stability are guaranteed.

According to Definition 2.1 and Definition E.1, the phase plots in Figure 7 well depict the relation between safety and stability:

- Figure 7 (a): **Only Stability** is guaranteed. Operating from any initial condition (inside or outside the safety set), the system state will converge to the equilibrium (zero). But safety is not guaranteed, since operating from an initial condition inside the safety set, the system will leave the safety set later.
- Figure 7 (b): **Only Safety** is guaranteed. Operating from any initial condition inside the safety set, the system state never leaves the safety set but does not converge to the equilibrium point.
- Figure 7 (c): **Both Safety and Stability** are guaranteed. Operating from any initial condition inside the safety set, the system state never leaves the safety set and converges to the equilibrium.

### E.2 PROVABLE SAFETY AND STABILITY GUARANTEES

Thanks to the residual action policy and safety-embedded reward, the proposed Phy-DRL exhibits mathematically-provable safety and stability guarantees, which is formally stated in the following theorem.

**Theorem E.2** (**Mathematically-Provable Safety and Stability Guarantees**). *Consider the safety set $\mathbb{X}$ (2), the safety envelope $\Omega$ (6), and the system (1) under control of Phy-DRL. The matrices $\mathbf{F}$ and $\mathbf{P}$ involved in the model-based action policy (4) and the safety-embedded reward (8) are computed according to Equation* (11)*, where $\mathbf{R}$ and $\mathbf{Q}^{-1}$ satisfies the inequalities (7) and (11). Both the safety and stability of the system (1) are guaranteed, if the sub-reward $r(\mathbf{s}(k), \mathbf{s}(k+1))$ in Equation (8) satisfies $r(\mathbf{s}(k), \mathbf{s}(k+1)) > (\alpha - 1) \cdot \mathbf{s}^\top(k) \cdot \mathbf{P} \cdot \mathbf{s}(k)$.*

*Proof.* This proof is straightforward and is based on the Proof of Theorem 5.3 in Appendix D.1. If $(\alpha - 1) \cdot V(\mathbf{s}(k)) - r(\mathbf{s}(k), \mathbf{s}(k+1)) < 0$, we obtain from Equation (49) that

$$V(\mathbf{s}(k+1)) < V(\mathbf{s}(k)), \quad \forall k \in \mathbb{N},$$

which implies that $V(\mathbf{s}(k))$ is strictly decreasing with respect to time $k \in \mathbb{N}$. The Phy-DRL in this condition thus stabilizes the real plant (1). Additionally, because of $V(\mathbf{s}(k))$'s strict decreasing, we obtain Equation (53) via considering the Equation (6). In light of Lemma 5.1, the condition Equation (7) is to guarantee that $\Omega \subseteq \mathbb{X}$, which with Equation (53) result in $\mathbf{s}(k) \in \Omega \subseteq \mathbb{X}, \forall k \in \mathbb{N}$, if $\mathbf{s}(1) \in \Omega$. We thus conclude that in this condition, both safety and stability are guaranteed, which completes the proof. □

## F  NN INPUT AUGMENTATION: EXPLANATIONS AND EXAMPLE

Line 16 shows that Algorithm 1 finally stacks vector with one. This operation means a PhyN node will be assigned to be one, and the bias will be thus treated as link weights associated with the nodes of ones.

As an example shown in Figure 8, the NN input augmentation empowers PhyN to capture well core nonlinearities of physical quantities such as kinetic energy ($\triangleq \frac{1}{2}mv^2$) and aerodynamic drag force ($\triangleq \frac{1}{2}\rho v^2 C_D A$), that drive the state dynamics of physical systems and then represent or approximate physical knowledge in form of the polynomial function.

Line 6–Line 13 of Algorithm 1 guarantee that the generated node-representation vectors embrace all the non-missing and non-redundant monomials of a polynomial function. One such example is shown in Figure 8. In this example, the $[\mathbf{x}]_1^2 \cdot [\mathbf{x}]_2^2$ is generated only by $[\mathbf{x}]_1 \cdot ([\mathbf{x}]_1 \cdot [\mathbf{x}]_2^2)$, not including others (see e.g., $[\mathbf{x}]_2 \cdot ([\mathbf{x}]_2 \cdot [\mathbf{x}]_1^2)$). Meanwhile, it can be straightforward to verify from the compact example in Figure 8 that all the generated monomials are non-missing and non-redundant.

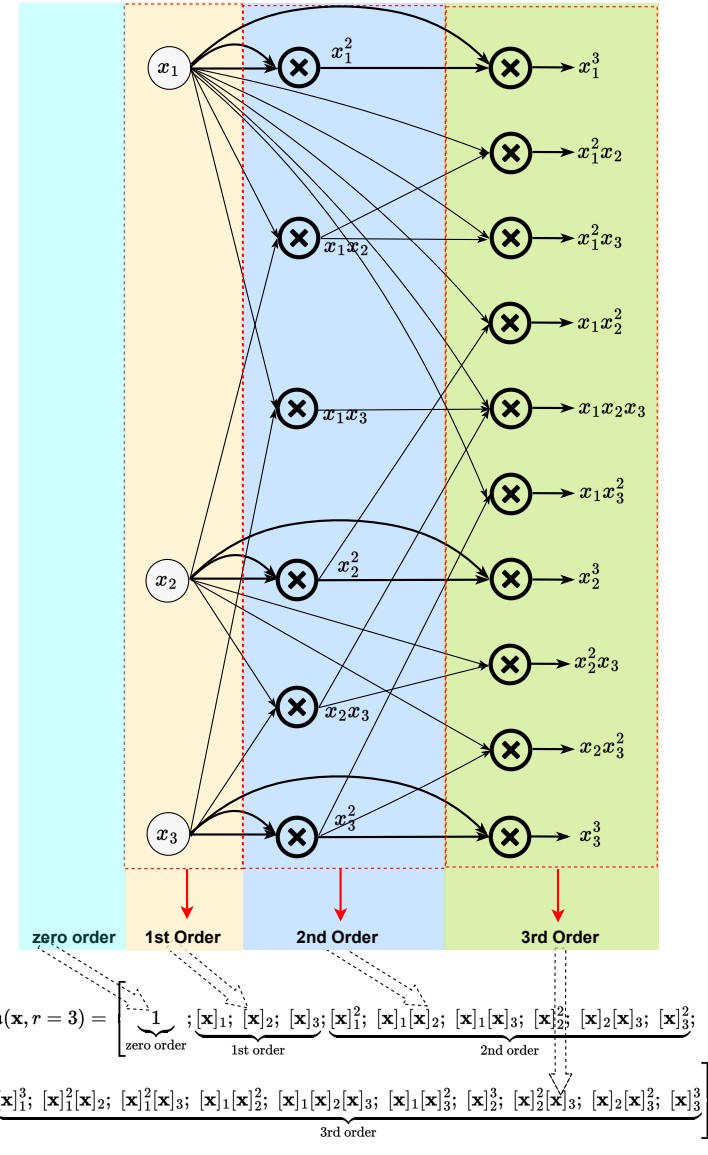

Figure 8: An example of Algorithm 1 in TensorFlow framework, where the input $\mathbf{x} \in \mathbb{R}^3$ and the augmentation order $r = 3$.

## G  CONTROLLABLE MODEL ACCURACY

The Taylor's Theorem offers a series expansion of arbitrary nonlinear functions, as shown below.

---

**Taylor's Theorem (Chapter 2.4 Königsberger (2013)):** Let $\mathbf{g} : \mathbb{R}^n \to \mathbb{R}$ be a $r$-times continuously differentiable function at the point $\mathbf{o} \in \mathbb{R}^n$. Then there exists $\mathbf{h}_\alpha : \mathbb{R}^n \to \mathbb{R}$, where $|\alpha| = r$, such that

$$\mathbf{g}(\mathbf{x}) = \sum_{|\alpha| \leq r} \frac{\partial^\alpha \mathbf{g}(\mathbf{o})}{\alpha!} (\mathbf{x} - \mathbf{o})^\alpha + \sum_{|\alpha| = r} \mathbf{h}_\alpha(\mathbf{x})(\mathbf{x} - \mathbf{o})^\alpha, \text{and} \lim_{\mathbf{x} \to \mathbf{o}} \mathbf{h}_\alpha(\mathbf{x}) = \mathbf{0}, \quad (55)$$

where $\alpha = [\alpha_1; \alpha_2; \ldots; \alpha_n]$, $|\alpha| = \sum_{i=1}^{n} \alpha_i$, $\alpha! = \prod_{i=1}^{n} \alpha_i!$, $\mathbf{x}^\alpha = \prod_{i=1}^{n} \mathbf{x}_i^{\alpha_i}$, and $\partial^\alpha \mathbf{g} = \frac{\partial^{|\alpha|} \mathbf{g}}{\partial \mathbf{x}_1^{\alpha_1} \cdot \ldots \cdot \partial \mathbf{x}_n^{\alpha_n}}$.

---

Given $\mathbf{h}_\alpha(\mathbf{x})$ is finite and $\|\mathbf{x} - \mathbf{o}\| < 1$, the error $\sum_{|\alpha| = r} \mathbf{h}_\alpha(\mathbf{x})(\mathbf{x} - \mathbf{o})^\alpha$ for approximating the ground truth $\mathbf{g}(\mathbf{x})$ will drop significantly as the order $r = |\alpha|$ increases and $\lim_{|\alpha| = r \to \infty} \mathbf{h}_\alpha(\mathbf{x})(\mathbf{x} - \mathbf{o})^\alpha = \mathbf{0}$. This allows for controllable model accuracy via controlling the order $r$.

## H  EXAMPLE: OBTAINING KNOWLEDGE SET $\mathbb{K}_{\mathcal{Q}}$

We use a simple example to explain how $\mathbb{K}_{\mathcal{Q}}$ is derived from Equation (5), according to Taylor's theorem Königsberger (2013). For simplification, we let $\mathbf{s}(k) \in \mathbb{R}, \mathbf{a}_{\text{drl}}(k) \in \mathbb{R}$, and $r_{\mathcal{Q}} = 2$. By Algorithm 1 with $r_{\langle t \rangle} = r_{\mathcal{Q}}$ and $\mathbf{y}_{\langle t \rangle} = [\mathbf{s}(k), \ \mathbf{s}(k+1), \ \mathbf{a}_{\text{drl}}(k)]^{\top}$, we have

$$\mathfrak{m}(\mathbf{s}(k), \mathbf{a}_{\text{drl}}(k), r_{\mathcal{Q}}) = [1, \mathbf{s}(k), \ \mathbf{s}(k+1), \ \mathbf{a}_{\text{drl}}(k), \ \mathbf{s}^2(k), \ \mathbf{s}(k) \cdot \mathbf{s}(k+1), \ \mathbf{s}(k) \cdot \mathbf{a}_{\text{drl}}(k), \ \mathbf{s}^2(k+1),$$
$$\mathbf{s}(k+1) \cdot \mathbf{a}_{\text{drl}}(k), \ \mathbf{a}_{\text{drl}}^2(k)]^{\top}. \quad (56)$$

Observing Equation (5), we can also denote the action-value function as $\mathcal{Q}^{\pi}(\mathcal{R}(\mathbf{s}(k), \mathbf{a}_{\text{drl}}(k))) \triangleq \mathcal{Q}^{\pi}(\mathbf{s}(k), \mathbf{a}_{\text{drl}}(k))$. For our reward, we let

$$\mathcal{R}(\mathbf{s}(k), \mathbf{a}_{\text{drl}}(k)) = \mathbf{s}^{\top}(k) \cdot \overline{\mathbf{A}}^{\top} \cdot \mathbf{P} \cdot \overline{\mathbf{A}} \cdot \mathbf{s}(k) - \mathbf{s}^{\top}(k+1) \cdot \mathbf{P} \cdot \mathbf{s}(k+1). \quad (57)$$

Right now, according to Taylor's theorem in Appendix G, expanding the action-value function $\mathcal{Q}^{\pi}(\mathcal{R}(\mathbf{s}(k), \mathbf{a}_{\text{drl}}(k)))$ around the $\mathcal{R}(\mathbf{s}(k), \mathbf{a}_{\text{drl}}(k))$, we have

$$\mathcal{Q}^{\pi}(\mathcal{R}(\mathbf{s}(k), \mathbf{a}_{\text{drl}}(k))) = b + w_1 \cdot \mathcal{R}(\mathbf{s}(k), \mathbf{a}_{\text{drl}}(k)) + \underbrace{w_2 \cdot \mathcal{R}^2(\mathbf{s}(k), \mathbf{a}_{\text{drl}}(k)) + \ldots}_{\triangleq \mathbf{p}(\mathbf{s}(k), \mathbf{a}_{\text{drl}}(k))}$$
$$= \mathbf{A}_{\mathcal{Q}} \cdot \mathfrak{m}(\mathbf{s}(k), \mathbf{a}_{\text{drl}}(k), r_{\mathcal{Q}}) + \mathbf{p}(\mathbf{s}(k), \mathbf{a}_{\text{drl}}(k)). \quad (58)$$

Recalling Equation (56), Equation (57), and Taylor's theorem in Appendix G, we then conclude from Equation (58) that

- Weight matrix $\mathbf{A}_{\mathcal{Q}} = \begin{bmatrix} b & 0 & 0 & 0 & w_1 \cdot [P]_{1,1} & 2w_1 \cdot [P]_{1,2} & 0 & w_1 \cdot [P]_{2,2} & 0 & 0 \end{bmatrix}$, where $b$ and $w_1$ are learning parameters.

- The unknown $\mathbf{p}(\mathbf{s}(k), \mathbf{a}_{\text{drl}}(k))$ does not include any monomial in Equation (56). So, the elements of $\mathbb{K}_{\mathcal{Q}}$ in this example are all entries of $\mathbf{A}_{\mathcal{Q}}$, i.e., $\mathbb{K}_{\mathcal{Q}} = \{[\mathbf{A}_{\mathcal{Q}}]_1, \cdots, [\mathbf{A}_{\mathcal{Q}}]_{10}\}$.

# I    PHYSICS-MODEL-GUIDED NEURAL NETWORK EDITING: EXPLANATIONS

## I.1    ACTIVATION EDITING

For the edited weight matrix $\mathbf{W}_{\langle t \rangle}$, if its entries in the same row are all in the knowledge set, the associated activation should be inactivated. Otherwise, the end-to-end input/output of DNN may not strictly preserve the available physics knowledge due to the additional nonlinear mappings induced by the activation functions. This thus motivates the physics-knowledge preserving computing, i.e., Line 19 of Algorithm 2. Figure 9 summarizes the flowchart of NN editing in a single PhyN layer:

- Given the node-representation vector from Algorithm 1, the original (fully-connected) weight matrix is edited via link editing to embed assigned physics knowledge, resulting in $\mathbf{W}_{\langle t \rangle}$.
- The edited weight matrix $\mathbf{W}_{\langle t \rangle}$ is separated into knowledge matrix $\mathbf{K}_{\langle t \rangle}$ and uncertainty matrix $\mathbf{U}_{\langle t \rangle}$, such that $\mathbf{W}_{\langle t \rangle} = \mathbf{K}_{\langle t \rangle} + \mathbf{U}_{\langle t \rangle}$. Specifically, the $\mathbf{K}_{\langle t \rangle}$, generated in Line 8 and Line 14 of Algorithm 2, includes all the parameters in the knowledge set. While the $\mathbf{M}_{\langle t \rangle}$, generated in Line 9 and Line 15, is used to generate uncertainty matrix $\mathbf{U}_{\langle t \rangle}$ (see Line 18) to include all the parameters excluded from knowledge set. This is achieved by freezing the parameters of $\mathbf{W}_{\langle t \rangle}$ that are included in the knowledge set to zeros.
- The $\mathbf{K}_{\langle t \rangle}$, $\mathbf{M}_{\langle t \rangle}$ and activation-masking vector $\mathbf{a}_{\langle t \rangle}$ (generated in Line 10 and Line 16) are used by activation editing for the physical-knowledge-preserving computing of output in each PhyN layer. The function of $\mathbf{a}_{\langle t \rangle}$ is to avoid the extra mapping (induced by activation) that prior physics knowledge does not include.

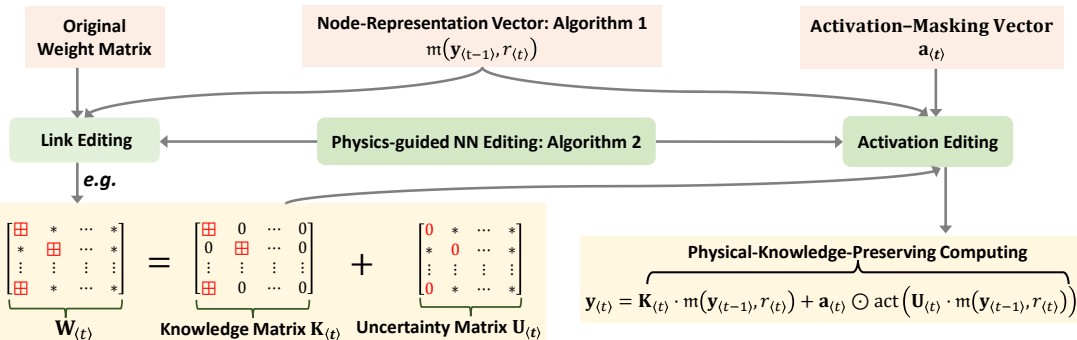

Figure 9: Flowchart of NN editing in single PhyN layer, where $\boxplus$ and $*$ denote a parameter included in and excluded from knowledge set, respectively.

## I.2    KNOWLEDGE PRESERVING AND PASSING

The flowchart of NN editing operating in cascade PhyNs is depicted in Figure 10. Lines 5-9 of Algorithm 2 means that $\mathbf{A}_\varpi = \mathbf{K}_{\langle 1 \rangle} + \mathbf{M}_{\langle 1 \rangle} \odot \mathbf{A}_\varpi$, leveraging which and the setting $r_{\langle 1 \rangle} = r_\varpi$, the ground-truth model (13) and (14) can be rewritten as

$$
\begin{aligned}
\mathbf{y} &= (\mathbf{K}_{\langle 1 \rangle} + \mathbf{M}_{\langle 1 \rangle} \odot \mathbf{A}_\varpi) \cdot \mathfrak{m}(\mathbf{x}, r) + \mathbf{p}(\mathbf{x}) \\
&= \mathbf{K}_{\langle 1 \rangle} \cdot \mathfrak{m}(\mathbf{x}, r_{\langle 1 \rangle}) + (\mathbf{M}_{\langle 1 \rangle} \odot \mathbf{A}_\varpi) \cdot \mathfrak{m}(\mathbf{x}, r_{\langle 1 \rangle}) + \mathbf{p}(\mathbf{x}).
\end{aligned} \tag{59}
$$

where we define:

$$
\mathbf{y} \triangleq \begin{cases} \mathcal{Q}(\mathbf{s}, \mathbf{a}_{\mathrm{drl}}), & \varpi = `\mathcal{Q}' \\ \pi(\mathbf{s}), & \varpi = `\pi' \end{cases}, \quad \mathbf{x} \triangleq \begin{cases} [\mathbf{s}; \mathbf{a}_{\mathrm{drl}}], & \varpi = `\mathcal{Q}' \\ \mathbf{s}, & \varpi = `\pi' \end{cases}, \quad r \triangleq \begin{cases} r_\mathcal{Q}, & \varpi = `\mathcal{Q}' \\ r_\pi, & \varpi = `\pi' \end{cases}.
$$

We obtain from Line 19 of Algorithm 2 that the output of the first PhyN layer is

$$
\mathbf{y}_{\langle 1 \rangle} = \mathbf{K}_{\langle 1 \rangle} \cdot \mathfrak{m}(\mathbf{x}, r_{\langle 1 \rangle}) + \mathbf{a}_{\langle 1 \rangle} \odot \mathrm{act}\left(\mathbf{U}_{\langle 1 \rangle} \cdot \mathfrak{m}\left(\mathbf{x}, r_{\langle 1 \rangle}\right)\right). \tag{60}
$$

Recalling that $\mathbf{K}_{\langle 1 \rangle}$ includes all the entries of $\mathbf{A}_\varpi$ while the $\mathbf{U}_{\langle 1 \rangle}$ includes remainders, we conclude from Equation (59) and Equation (60) that the available physics knowledge pertaining to the ground-truth model has been embedded to the first PhyN layer. As Figure 10 shows the knowledge embedded

in the first layer shall be passed down to the remaining cascade PhyNs and preserved therein, such that the end-to-end critic and actor network can strictly comply with the physics knowledge. This knowledge passing is achieved by the block matrix $\mathbf{K}_{\langle p \rangle}$ generated in Line 14, thanks to which, the output of $t$-th PhyN layer satisfies

$$[\mathbf{y}_{\langle t \rangle}]_{1:\mathrm{len}(\mathbf{y})} = \underbrace{\mathbf{K}_{\langle 1 \rangle} \cdot \mathfrak{m}(\mathbf{x}, r_{\langle 1 \rangle})}_{\text{knowledge passing}} + \underbrace{[\mathbf{a}_{\langle t \rangle} \odot \mathrm{act}\big(\mathbf{U}_{\langle t \rangle} \cdot \mathfrak{m}(\mathbf{y}_{\langle t-1 \rangle}, r_{\langle t \rangle})\big)]_{1:\mathrm{len}(\mathbf{y})}}_{\text{knowledge preserving}}, \ \forall t \in \{2, \dots, p\}. \quad (61)$$

Meanwhile, the $\mathbf{U}_{\langle t \rangle} = \mathbf{M}_{\langle t \rangle} \odot \mathbf{W}_{\langle t \rangle}$ means the masking matrix $\mathbf{M}_{\langle t \rangle}$ generated in Line 15 is to remove the spurious correlations in the cascade PhyNs, which is depicted by the cutting link operation in Figure 10.

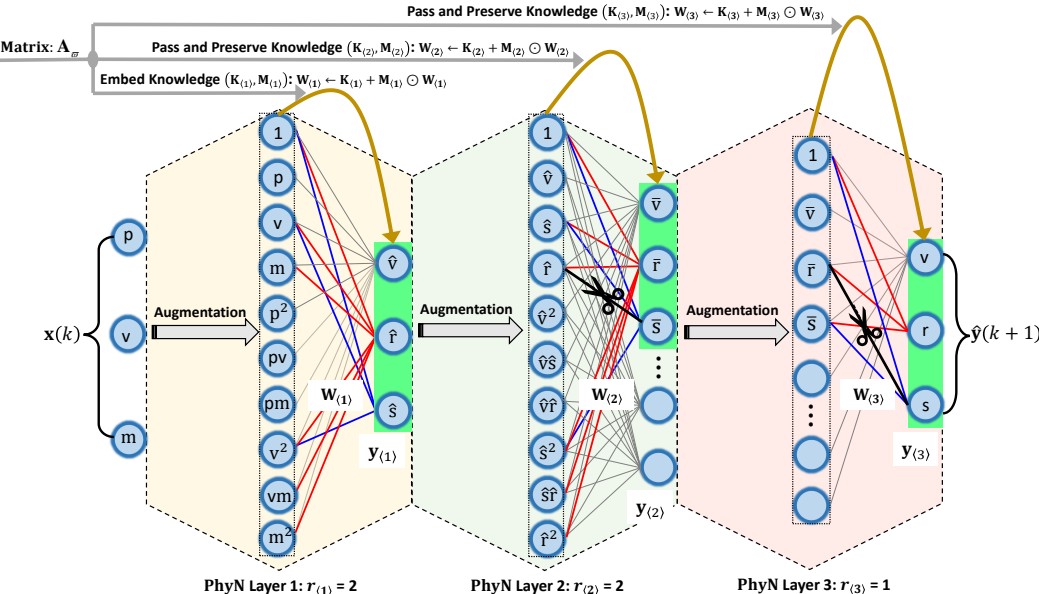

Figure 10: Example of NN editing, i.e., Algorithm 2. (i) Parameters excluded from the knowledge set are formed by the grey links, while the parameters included in the knowledge set are formed by the red and blue links. (ii) Cutting black links is to avoid spurious correlations. Otherwise, the links can lead to violation of physics knowledge about the governing Equation (13) and Equation (14).

## J    PROOF OF THEOREM 6.4

Let us first consider the first PhyN layer, i.e., the case $t = 1$. Line 8 of Algorithm 2 means that the knowledge matrix $\mathbf{K}_{\langle 1 \rangle}$ includes parameters included in knowledge sets, whose corresponding entries in the masking matrix $\mathbf{M}_{\langle 1 \rangle}$ (generated in Line 9 of Algorithm 2) are frozen to be zeros. Consequently, both $\mathbf{M}_{\langle 1 \rangle} \odot \mathbf{A}_\varpi$ and $\mathbf{U}_{\langle 1 \rangle} = \mathbf{M}_{\langle 1 \rangle} \odot \mathbf{W}_{\langle 1 \rangle}$ excludes all the parameters of knowledge matrix $\mathbf{K}_{\langle 1 \rangle}$. We thus conclude that $\mathbf{M}_{\langle 1 \rangle} \odot \mathbf{A}_\varpi \cdot \mathrm{m}(\mathbf{x}, r_{\langle 1 \rangle}) + \mathbf{p}(\mathbf{x})$ in the ground-truth model (59) and $\mathbf{a}_{\langle 1 \rangle} \odot \mathrm{act}\big(\mathbf{U}_{\langle 1 \rangle} \cdot \mathrm{m}\big(\mathbf{x}, r_{\langle 1 \rangle}\big)\big)$ in the output computation in Line 21 are independent of the term $\mathbf{K}_{\langle 1 \rangle} \cdot \mathrm{m}(\mathbf{x}, r_{\langle 1 \rangle})$.

Moreover, the activation-masking vector (generated in Line 10 of Algorithm 2) indicates that if all the entries in the $i$-th row of masking matrix are zeros (implying all the entries in the $i$-th row of weight matrix are included in the knowledge set $\mathbb{K}_\varpi$), the activation function corresponding to the output's $i$-th entry is inactive. Finally, we arrive in the conclusion that the input/output of the first PhyN layer strictly complies with the available physics knowledge pertaining to the ground truth (59), i.e., if the $[\mathbf{A}_\varpi]_{i,j} \in \mathbb{K}_\varpi$, the $[\mathbf{y}_{\langle 1 \rangle}]_i$ does not have monomials $[\mathrm{m}(\mathbf{x}, r_\varpi)]_j$.

We next consider the remaining PhyN layers. Considering Line 19 of Algorithm 2, we have

$$
\begin{aligned}
& [\mathbf{y}_{\langle p \rangle}]_{1:\mathrm{len}(\mathbf{y})} \\
&= [\mathbf{K}_{\langle p \rangle} \cdot \mathrm{m}(\mathbf{y}_{\langle p-1 \rangle}, r_{\langle p \rangle})]_{1:\mathrm{len}(\mathbf{y})} + [\mathbf{a}_{\langle p \rangle} \odot \mathrm{act}\big(\mathbf{U}_{\langle p \rangle} \cdot \mathrm{m}(\mathbf{y}_{\langle p-1 \rangle}, r_{\langle p \rangle})\big)]_{1:\mathrm{len}(\mathbf{y})} \\
&= \mathbf{I}_{\mathrm{len}(\mathbf{y})} \cdot [\mathrm{m}(\mathbf{y}_{\langle p-1 \rangle}, r_{\langle p \rangle})]_{2:(\mathrm{len}(\mathbf{y})+1)} + [\mathbf{a}_{\langle p \rangle} \odot \mathrm{act}\big(\mathbf{U}_{\langle p \rangle} \cdot \mathrm{m}(\mathbf{y}_{\langle p-1 \rangle}, r_{\langle p \rangle})\big)]_{1:\mathrm{len}(\mathbf{y})} && (62) \\
&= \mathbf{I}_{\mathrm{len}(\mathbf{y})} \cdot [\mathbf{y}_{\langle p-1 \rangle}]_{1:\mathrm{len}(\mathbf{y})} + [\mathbf{a}_{\langle p \rangle} \odot \mathrm{act}\big(\mathbf{U}_{\langle p \rangle} \cdot \mathrm{m}(\mathbf{y}_{\langle p-1 \rangle}, r_{\langle p \rangle})\big)]_{1:\mathrm{len}(\mathbf{y})} && (63) \\
&= [\mathbf{y}_{\langle p-1 \rangle}]_{1:\mathrm{len}(\mathbf{y})} + [\mathbf{a}_{\langle p \rangle} \odot \mathrm{act}\big(\mathbf{U}_{\langle p \rangle} \cdot \mathrm{m}(\mathbf{y}_{\langle p-1 \rangle}, r_{\langle p \rangle})\big)]_{1:\mathrm{len}(\mathbf{y})} \\
&= [\mathbf{K}_{\langle p-1 \rangle} \cdot \mathrm{m}(\mathbf{y}_{\langle p-2 \rangle}, r_{\langle p-1 \rangle})]_{1:\mathrm{len}(\mathbf{y})} + [\mathbf{a}_{\langle p \rangle} \odot \mathrm{act}\big(\mathbf{U}_{\langle p \rangle} \cdot \mathrm{m}(\mathbf{y}_{\langle p-1 \rangle}, r_{\langle p \rangle})\big)]_{1:\mathrm{len}(\mathbf{y})} \\
&= \mathbf{I}_{\mathrm{len}(\mathbf{y})} \cdot [\mathrm{m}(\mathbf{y}_{\langle p-2 \rangle}, r_{\langle p-1 \rangle})]_{2:(\mathrm{len}(\mathbf{y})+1)} + [\mathbf{a}_{\langle p \rangle} \odot \mathrm{act}\big(\mathbf{U}_{\langle p \rangle} \cdot \mathrm{m}(\mathbf{y}_{\langle p-1 \rangle}, r_{\langle p \rangle})\big)]_{1:\mathrm{len}(\mathbf{y})} \\
&= \mathbf{I}_{\mathrm{len}(\mathbf{y})} \cdot [\mathbf{y}_{\langle p-2 \rangle}]_{1:\mathrm{len}(\mathbf{y})} + [\mathbf{a}_{\langle p \rangle} \odot \mathrm{act}\big(\mathbf{U}_{\langle p \rangle} \cdot \mathrm{m}(\mathbf{y}_{\langle p-1 \rangle}, r_{\langle p \rangle})\big)]_{1:\mathrm{len}(\mathbf{y})} \\
&= [\mathbf{y}_{\langle p-2 \rangle}]_{1:\mathrm{len}(\mathbf{y})} + [\mathbf{a}_{\langle p \rangle} \odot \mathrm{act}\big(\mathbf{U}_{\langle p \rangle} \cdot \mathrm{m}(\mathbf{y}_{\langle p-1 \rangle}, r_{\langle p \rangle})\big)]_{1:\mathrm{len}(\mathbf{y})} \\
&= \ldots \\
&= [\mathbf{y}_{\langle 1 \rangle}]_{1:\mathrm{len}(\mathbf{y})} + [\mathbf{a}_{\langle p \rangle} \odot \mathrm{act}\big(\mathbf{U}_{\langle p \rangle} \cdot \mathrm{m}(\mathbf{y}_{\langle p-1 \rangle}, r_{\langle p \rangle})\big)]_{1:\mathrm{len}(\mathbf{y})} \\
&= [\mathbf{K}_{\langle 1 \rangle} \cdot \mathrm{m}(\mathbf{x}, r_{\langle 1 \rangle})]_{1:\mathrm{len}(\mathbf{y})} + [\mathbf{a}_{\langle p \rangle} \odot \mathrm{act}\big(\mathbf{U}_{\langle p \rangle} \cdot \mathrm{m}(\mathbf{y}_{\langle p-1 \rangle}, r_{\langle p \rangle})\big)]_{1:\mathrm{len}(\mathbf{y})} \\
&= \mathbf{K}_{\langle 1 \rangle} \cdot \mathrm{m}(\mathbf{x}, r_{\langle 1 \rangle}) + [\mathbf{a}_{\langle p \rangle} \odot \mathrm{act}\big(\mathbf{U}_{\langle p \rangle} \cdot \mathrm{m}(\mathbf{y}_{\langle p-1 \rangle}, r_{\langle p \rangle})\big)]_{1:\mathrm{len}(\mathbf{y})}, && (64)
\end{aligned}
$$

where Equation (62) and Equation (63) are obtained from their previous steps via considering the structure of block matrix $\mathbf{K}_{\langle t \rangle}$ (generated in Line 14 of Algorithm 2) and the formula of augmented monomials: $\mathrm{m}(\mathbf{y}, r) = \big[1;\ \mathbf{y};\ [\mathrm{m}(\mathbf{y}, r)]_{(\mathrm{len}(\mathbf{y})+2):\mathrm{len}(\mathrm{m}(\mathbf{y},r))}\big]$ (generated via Algorithm 2). The remaining iterative steps follow the same path.

The training loss function is to push the terminal output of Algorithm 2 to approximate the real output $\mathbf{y}$, which in light of Equation (64) yields

$$
\begin{aligned}
\widehat{\mathbf{y}} &= \mathbf{K}_{\langle 1 \rangle} \cdot \mathrm{m}(\mathbf{x}, r_{\langle 1 \rangle}) + [\mathbf{a}_{\langle p \rangle} \odot \mathrm{act}\big(\mathbf{U}_{\langle p \rangle} \cdot \mathrm{m}(\mathbf{y}_{\langle p-1 \rangle}, r_{\langle p \rangle})\big)]_{1:\mathrm{len}(\mathbf{y})} \\
&= \mathbf{K}_{\langle 1 \rangle} \cdot \mathrm{m}(\mathbf{x}, r_{\langle 1 \rangle}) + \mathbf{a}_{\langle p \rangle} \odot \mathrm{act}\big(\mathbf{U}_{\langle p \rangle} \cdot \mathrm{m}(\mathbf{y}_{\langle p-1 \rangle}, r_{\langle p \rangle})\big), && (65)
\end{aligned}
$$

where Equation (65) from its previous step is obtained via considering the fact $\mathrm{len}(\widehat{\mathbf{y}}) = \mathrm{len}(\mathbf{y}) = \mathrm{len}(\mathbf{y}_{\langle p \rangle})$. Meanwhile, the condition of generating a weight-masking matrix in Line 15 of Algorithm 2 removes all the node-representations' connections with the parameters of knowledge set included in $\mathbf{K}_{\langle 1 \rangle}$. Therefore, we can conclude that in the terminal output computation in Equation (65), the term $\mathbf{a}_{\langle p \rangle} \odot \mathrm{act}\big(\mathbf{U}_{\langle p \rangle} \cdot \mathrm{m}(\mathbf{y}_{\langle p-1 \rangle}, r_{\langle p \rangle})\big)$ does not have influence on the computing of knowledge term $\mathbf{K}_{\langle 1 \rangle} \cdot \mathrm{m}(\mathbf{x}, r_{\langle 1 \rangle})$. Thus, the Algorithm 2 strictly embeds and preserves the available knowledge pertaining to the physics model of ground truth in Equation (59).

## K  EXPERIMENT: CART-POLE SYSTEM

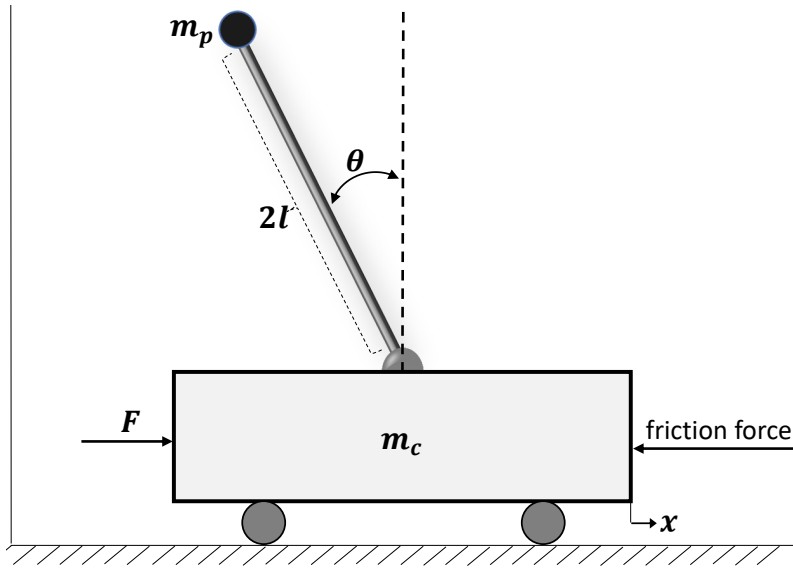

Figure 11: Mechanical analog of inverted pendulums.

### K.1  PHYSICS-MODEL KNOWLEDGE

To have the model-based action policy, the first step is to obtain system matrix $\mathbf{A}$ and control structure matrix $\mathbf{B}$ in a real plant (1). In other words, the available model knowledge about the dynamics of the cart-pole system is a linear model:

$$\bar{\mathbf{s}}(k+1) = \mathbf{A} \cdot \bar{\mathbf{s}}(k) + \mathbf{B} \cdot \bar{\mathbf{a}}(k), \quad k \in \mathbb{N}. \tag{66}$$

We refer to the dynamics model of cart-pole system described in Florian (2007) and consider the approximations $\cos\theta \approx 1$, $\sin\theta \approx \theta$ and $\omega^2 \sin\theta \approx 0$ for obtaining $(\mathbf{A}, \mathbf{B})$ as

$$\mathbf{A} = \begin{bmatrix} 1 & 0.0333 & 0 & 0 \\ 0 & 1 & -0.0565 & 0 \\ 0 & 0 & 1 & 0.0333 \\ 0 & 0 & 0.8980 & 1 \end{bmatrix}, \quad \mathbf{B} = [0 \; 0.0334 \; 0 \; -0.0783]^\top. \tag{67}$$

### K.2  SAFETY KNOWLEDGE

Considering the safety conditions in Equation (17) and the formula of safety set in Equation (2), we have

$$\mathbf{s} = \begin{bmatrix} x \\ v \\ \theta \\ \omega \end{bmatrix}, \quad \mathbf{D} = \begin{bmatrix} 1 & 0 & 0 & 0 \\ 0 & 0 & 1 & 0 \end{bmatrix}, \quad \mathbf{v} = \begin{bmatrix} 0 \\ 0 \end{bmatrix}, \quad \overline{\mathbf{v}} = \begin{bmatrix} 0.9 \\ 0.8 \end{bmatrix}, \quad \underline{\mathbf{v}} = \begin{bmatrix} -0.9 \\ -0.8 \end{bmatrix}, \tag{68}$$

based on which, then according to the $\overline{\Lambda}$, $\underline{\Lambda}$ and $\mathbf{d}$ defined in Lemma 5.1, we have

$$\mathbf{d} = \begin{bmatrix} -1 \\ -1 \end{bmatrix}, \quad \overline{\Lambda} = \underline{\Lambda} = \begin{bmatrix} 0.9 & 0 \\ 0 & 0.8 \end{bmatrix} \tag{69}$$

from which and $\mathbf{D}$ given in Equation (68), we then have

$$\overline{\mathbf{D}} = \frac{\mathbf{D}}{\overline{\Lambda}} = \underline{\mathbf{D}} = \frac{\mathbf{D}}{\underline{\Lambda}} = \begin{bmatrix} \frac{10}{9} & 0 & 0 & 0 \\ 0 & 0 & \frac{5}{4} & 0 \end{bmatrix}. \tag{70}$$

### K.3 Model-Based Action Policy and DRL Reward

With the knowledge given in Equation (69) and Equation (70), the matrices $\mathbf{F}$ and $\mathbf{P}$ are ready to be obtained through solving the centering problem in Equation (12). We let $\alpha = 0.98$. By the CVXPY toolbox Diamond & Boyd (2016) in Python, we obtain

$$\mathbf{Q} = \begin{bmatrix} 0.66951866 & -0.69181711 & -0.27609583 & 0.55776279 \\ -0.69181711 & 9.86247186 & 0.1240829 & -12.4011146 \\ -0.27609583 & 0.1240829 & 0.66034399 & -2.76789607 \\ 0.55776279 & -12.4011146 & -2.76789607 & 32.32280039 \end{bmatrix},$$

$$\mathbf{R} = \begin{bmatrix} -6.40770185 & -18.97723676 & 6.10235911 & 31.03838284 \end{bmatrix},$$

based on this, we then have

$$\mathbf{P} = \mathbf{Q}^{-1} = \begin{bmatrix} 4.6074554 & 1.49740096 & 5.80266046 & 0.99189224 \\ 1.49740096 & 0.81703147 & 2.61779592 & 0.51179642 \\ 5.80266046 & 2.61779592 & 11.29182733 & 1.87117709 \\ 0.99189224 & 0.51179642 & 1.87117709 & 0.37041435 \end{bmatrix}, \tag{71}$$

$$\mathbf{F} = \mathbf{R} \cdot \mathbf{P} = \begin{bmatrix} 8.25691599 & 6.76016534 & 40.12484514 & 6.84742553 \end{bmatrix}, \tag{72}$$

$$\overline{\mathbf{A}} = \mathbf{A} + \mathbf{B} \cdot \mathbf{F} = \begin{bmatrix} 1 & 0.03333333 & 0 & 0 \\ 0.27592037 & 1.22590363 & 1.2843559 & 0.2288196 \\ 0 & 0 & 1 & 0.03333333 \\ -0.64668827 & -0.52946156 & -2.24458365 & 0.46370415 \end{bmatrix}. \tag{73}$$

With these solutions and letting $w(\mathbf{s}(k), \mathbf{a}(k)) = -\mathbf{a}_{\mathrm{drl}}^2(k)$, the model-based action policy (4) and the safety-embedded reward (8) are then ready for the Phy-DRL.

### K.4 Sole Model-Based Action Policy: Failure Due to Large Model Mismatch

The trajectories of the cart-pole system under the control of sole model-based action policy, i.e., $\mathbf{a}(k) = \mathbf{a}_{\mathrm{phy}}(k) = \mathbf{F} \cdot \mathbf{s}(k)$, are shown in Figure 12. The system's initial condition lies in the safety envelope, i.e., $\mathbf{s}(1) \in \Omega$. Figure 12 shows the sole model-based action policy cannot stabilize the system and cannot guarantee its safety. This failure is due to a large model mismatch between the simplified linear model (66) and the real system having nonlinear dynamics. While Figure 3 shows the Phy-DRL successfully overcomes the large model mismatch and renders the system safe and stable, tested with many initial conditions $\mathbf{s}(1) \in \mathbb{X}$.

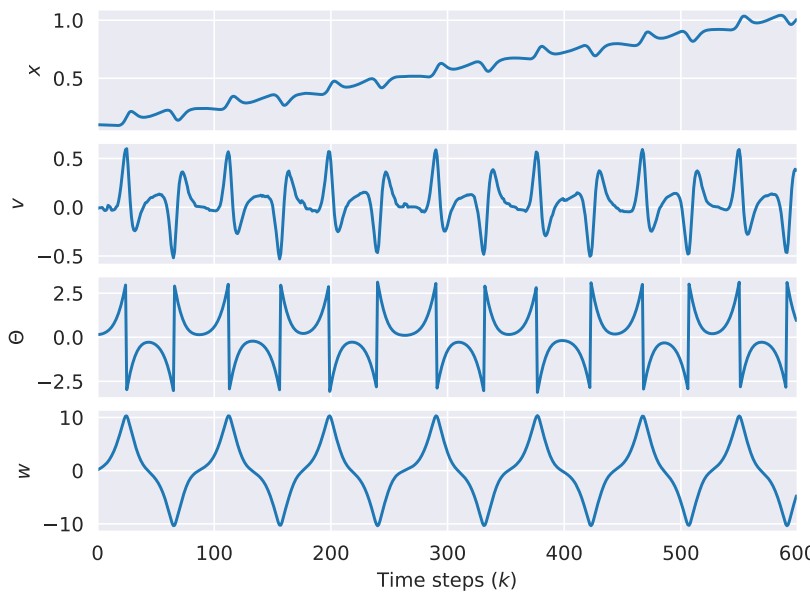

Figure 12: System trajectories of the cart-pole system: unstable and unsafe.

### K.5 Configurations: Networks and Training Conditions

In this case study, the goal of action policy is to stabilize the pendulum at the equilibrium $\mathbf{s}^* = [x^*, v^*, \theta^*, \omega^*]^\top = [0, 0, 0, 0]^\top$, while constraining system state to the safety set in Equation (17). We convert the measured angle $\theta$ into $\sin(\theta)$ and $\cos(\theta)$ to simplify the learning process. Therefore, the observation space can be expressed as

$$\mathbf{s} = [x, \ v, \ \sin(\theta), \ \sin(\theta), \ \omega]^\top.$$

We also added a terminal condition to the training episode that stops the running of the cart-pole system when a violation of safety occurs for both DRL and Phy-DRL in training, depicted as follows:

$$\beta(\mathbf{s}(k)) = \begin{cases} 1, & \text{if } |x(k)| \geq 0.9 \text{ or } |\theta(k)| \geq 0.8 \\ 0, & \text{otherwise.} \end{cases}$$

Specifically, the running of the cart-pole system (starting from an initial condition) during training is terminated if either its cart position or pendulum angle exceeds the safety bounds or the pendulum falls. During training, we reset episodes of the system running from random initial conditions inside the safety set if the maximum step of system running is reached, or $\beta(\mathbf{s}(k)) = 1$.

The development of Phy-DRL is based on the DDPG algorithm. The actor and critic networks in the DDPG algorithm are implemented as a Multi-Layer Perceptron (MLP) with four fully connected layers. The output dimensions of critic and actor networks are 256, 128, 64, and 1, respectively. The activation functions of the first three neural layers are ReLU, while the output of the last layer is the Tanh function for the actor-network and Linear for the critic network. The input of the critic network is $[\mathbf{s}; \mathbf{a}]$, while the input of the actor-network is $\mathbf{s}$.

### K.6 Training

For the code, we use the Python API for the TensorFlow framework Kingma & Ba and the Adam optimizer Abadi et al. for training. This project is using the settings: 1) Ubuntu 20.04, 2) Python 3.7, 3) TensorFlow 2.5.0, 4) Numpy 1.19.5, and 5) Gym 0.20.

For Phy-DRL, we let discount factor $\gamma = 0.4$, and the learning rates of critic and actor networks are the same as 0.0003. We set the batch size to 200. The total training steps are $10^6$, and the maximum step number of one episode is 1000. Each weight matrix is initialized randomly from a (truncated) normal distribution with zero mean and standard deviation, discarding and re-drawing any samples more than two standard deviations from the mean. We initialize each bias according to the normal distribution with zero mean and standard deviation.

### K.7 Testing Comparisons: Model for State Prediction?

We perform testing comparisons of two Phy-DRLs (with and without a model inside for state prediction) and two DRLs (with and without a model inside for state prediction). The two DRLs use the same CLF reward as $\mathcal{R}(\cdot) = \mathbf{s}^\top(k) \cdot \mathbf{P} \cdot \mathbf{s}(k) - \mathbf{s}^\top(k+1) \cdot \mathbf{P} \cdot \mathbf{s}(k+1) + w(\mathbf{s}(k), \mathbf{a}(k))$ (proposed in Westenbroek et al. (2022)), where the $\mathbf{P}$ is the same as the one in the Phy-DRL's safety-embedded reward. The model used for state prediction is the one in Equation (66). The testing results are presented in Figure 13. We note in Figure 13 that

- **mf-Phy-DRL**, denotes a policy trained via our Phy-DRL that does not adopt the model in Equation (66) for state prediction.
- **mb-Phy-DRL**, denotes a policy trained via our Phy-DRL that adopts the model in Equation (66) for state prediction.
- **mf-DRL**, denotes a policy trained via DRL that does not adopt the model in Equation (66) for state prediction.
- **mb-DRL**, denotes a policy trained via DRL that adopts the model in Equation (66) for state prediction.

Besides, all the training models of mf-Phy-DRL, mb-Phy-DRL, mf-DRL and mb-DRL have the same configurations of critic and actor networks, presented in Appendix K.5. The performance metrics are the areas of IE and EE samples, defined in Equation (18) and Equation (19), respectively.

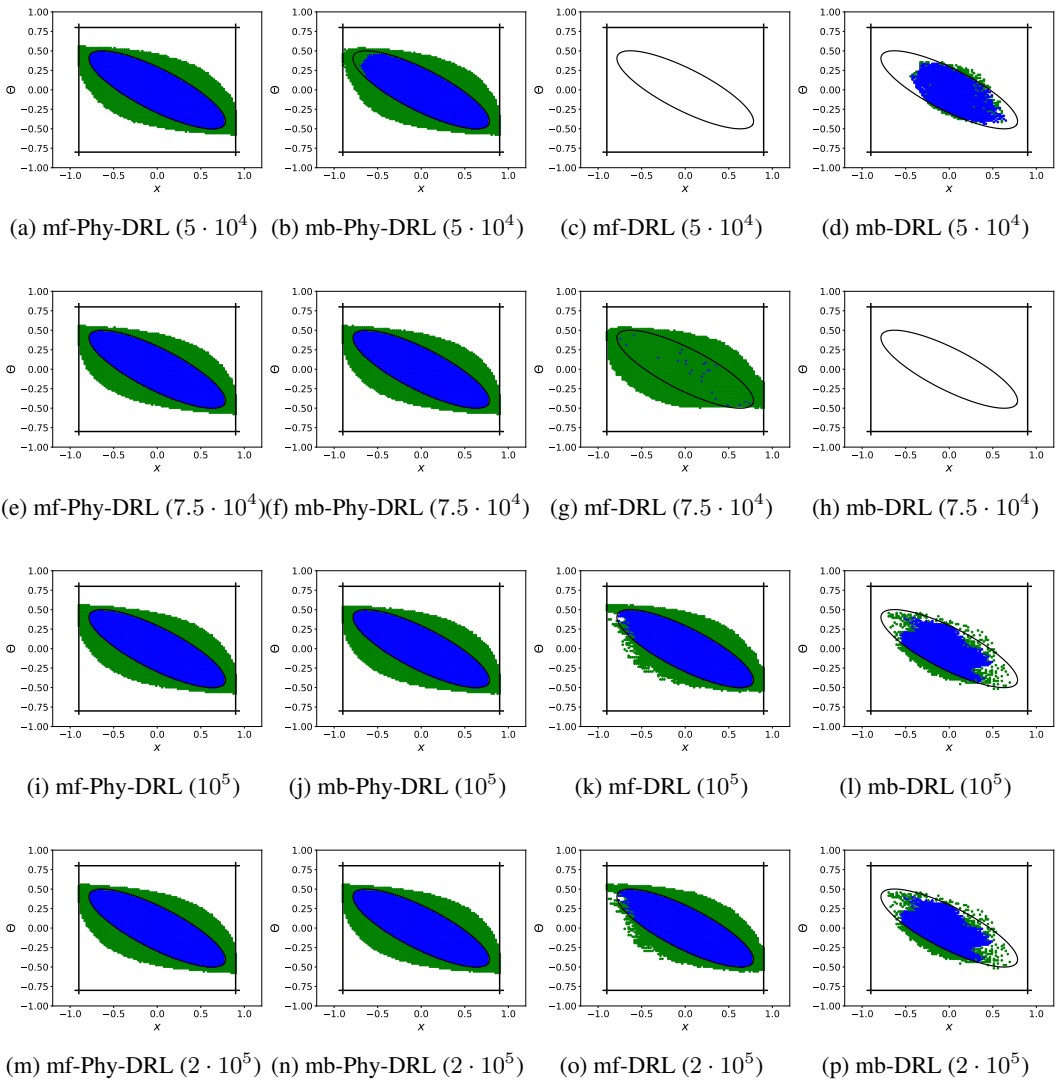

Figure 13: Blue: area of IE samples defined in Equation (18). Green: area of EE samples defined in Equation (19). Rectangular area: safety set. Ellipse area: safety envelope. (a)-(d): all the policies are obtained via training for $5 \cdot 10^4$ steps. (e)-(h): all the policies are obtained via training for $7.5 \cdot 10^4$ steps. (i)-(l): all the policies are obtained via training for $10^5$ steps. (m)-(p): all the policies are obtained via training for $2 \cdot 10^5$ steps.

Observing Figure 13, we conclude that

- Our Phy-DRL that does not adopt a model for state prediction can quickly complete training (only $5 \cdot 10^4$ steps) for rendering safety envelope invariant. While our Phy-DRL that adopts a model for state prediction is slightly slow, $7.5 \cdot 10^4$ training steps can complete the safety task.

- Safety areas of DRL policies are much smaller than our Phy-DRL policies. Even increasing training to $2 \cdot 10^5$ steps, mf-DRL and mb-DRL policies cannot render the safety envelope invariant, i.e., provide a safety guarantee.

- Incorporating our linear model into DRL for state prediction does not improve system performance, a root reason was revealed in Janner et al. (2019) that the performance of model-based RL is constrained by modeling errors or model mismatch. Specifically, if a model has a large model mismatch with the nonlinear model of real system dynamics, relying on the model for state prediction (in model-based RL) may lead to potential sub-optimal performance and safety violations.

## L  EXPERIMENT: QUADRUPED ROBOT

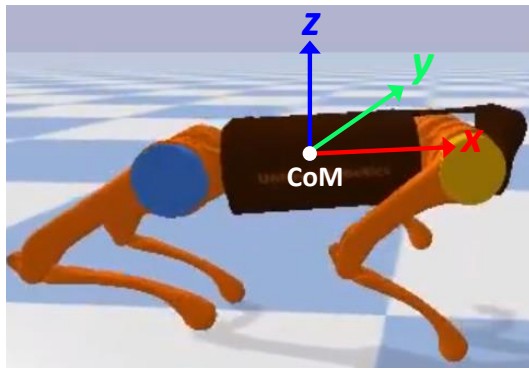

Figure 14: Quadruped robot: 3D single rigid-body model.

The developed package for training the robot via DRL is built on a Python-based framework for the A1 robot from Unitree, released in GitHub. The original framework includes a simulation based on Pybullet, an interface for direct sim-to-real transfer, and an implementation of the Convex MPC Controller for basic motion control. For the quadruped robot, the outputs of the designed action policies are the desired positional acceleration and rotational accelerations. The computed accelerations are then converted to the low-level motors' torque control.

### L.1  OVERVIEW: BEST-TRADE-OFF BETWEEN MODEL-BASED DESIGN AND DATA-DRIVEN DESIGN

In the following sections, we demonstrate that the residual action policy of Phy-DRL is a best-trade-off between the model-based policy and the data-driven DRL policy for safety-critical control. Specifically, Appendix L.2 explicitly shows the dynamics of quadruped robot is highly nonlinear, directly leveraging which for designing an analyzable and verifiable model-based action policy is extremely hard. While Appendix L.3 shows the residual action policy of Phy-DRL allows the model-based design to be simplified to be an analyzable and verifiable linear one, while offering fast and stable training (see Appendix L.6 and Appendix L.7).

### L.2  REAL SYSTEM DYNAMICS: HIGHLY NONLINEAR!

The dynamics model of the robot is based on a single rigid body subject to forces at the contact patches Di Carlo et al. (2018). Referring to Figure 14, the considered robot dynamics is characterized by the position of the body's center of mass (CoM) $\mathbf{p} = [p_x; p_y; p_z] \in \mathbb{R}^3$, the CoM velocity $\mathbf{v} \triangleq \dot{\mathbf{p}} \in \mathbb{R}^3$, the Euler angles $\widetilde{\mathbf{e}} = [\phi; \theta; \psi] \in \mathbb{R}^3$ with $\phi$, $\theta$ and $\psi$ being roll, pitch and yaw angles, respectively, and the angular velocity in world coordinates $\mathbf{w} \in \mathbb{R}^3$. The robot's state vector is

$$\widehat{\mathbf{s}} = [\text{CoM x-position; CoM y-position; CoM z-height; roll; pitch; yaw; CoM x-velocity;}$$
$$\text{CoM y-velocity; CoM z-velocity; angular velocity } \mathbf{w} \in \mathbb{R}^3]. \quad (74)$$

Before presenting the body dynamics model, we introduce the set of foot states {Left Front (LF), Right Front (RF), Left Rear (LR), Right Rear (RR)}, based on which we define the following two footsteps for describing the trotting behavior of the quadruped robot:

$$\text{Step}_1 \triangleq \{\text{LF}=1, \text{RF}=0, \text{LR}=0, \text{RR}=1\}, \quad \text{Step}_2 \triangleq \{\text{LF}=0, \text{RF}=1, \text{LR}=1, \text{RR}=0\}, \quad (75)$$

where '0' indicates that the corresponding foot is a stance, and '1' denotes otherwise. The considered walking controller lifts two feet at a time by switching between the two stepping primitives in the following order:

$$\text{Step}_1 \rightarrow \text{Step}_2 \rightarrow \text{Step}_1 \rightarrow \text{Step}_2 \rightarrow \text{Step}_1 \rightarrow \text{Step}_2 \rightarrow \text{Step}_1 \rightarrow \text{Step}_2 \rightarrow \ldots \rightarrow \text{repeating} \rightarrow \ldots$$

According to the literature Di Carlo et al. (2018), the body dynamics of quadruped robots can be described by

$$
\frac{d}{dt}
\underbrace{
\begin{bmatrix}
\mathbf{p} \\
\widetilde{\mathbf{e}} \\
\mathbf{v} \\
\mathbf{w}
\end{bmatrix}
}_{\triangleq \widehat{\mathbf{s}}}
=
\begin{bmatrix}
\mathbf{O}_3 & \mathbf{O}_3 & \mathbf{I}_3 & \mathbf{O}_3 \\
\mathbf{O}_3 & \mathbf{O}_3 & \mathbf{O}_3 & \mathbf{R}(\phi, \theta, \psi) \\
\mathbf{O}_3 & \mathbf{O}_3 & \mathbf{O}_3 & \mathbf{O}_3 \\
\mathbf{O}_3 & \mathbf{O}_3 & \mathbf{O}_3 & \mathbf{O}_3
\end{bmatrix}
\cdot
\begin{bmatrix}
\mathbf{p} \\
\widetilde{\mathbf{e}} \\
\mathbf{v} \\
\mathbf{w}
\end{bmatrix}
+ \widehat{\mathbf{B}}_{\sigma(t)} \cdot \mathbf{a}_{\sigma(t)} +
\underbrace{
\begin{bmatrix}
\mathbf{0}_3 \\
\mathbf{0}_3 \\
\mathbf{0}_3 \\
\widetilde{\mathbf{g}}
\end{bmatrix}
}_{\triangleq \mathbf{c}}
+ \mathbf{f}(\widehat{\mathbf{s}}), \quad (76)
$$

where $\widetilde{\mathbf{g}} = [0; 0; -g] \in \mathbb{R}^3$ with $g$ being the gravitational acceleration, $\mathbf{f}(\widehat{\mathbf{s}})$ denotes model mismatch, the $\mathbf{R}(\phi, \theta, \psi) = \mathbf{R}_z(\psi) \cdot \mathbf{R}_y(\theta) \cdot \mathbf{R}_x(\phi) \in \mathbb{R}^{3 \times 3}$ with $\mathbf{R}_i(\alpha) \in \mathbb{R}^{3 \times 3}$ being the rotation of angle $\alpha$ about axis $i$. The $\mathbf{a}_{\sigma(t)} \in \mathbb{R}^9$, $\sigma(t) \in \mathbb{S} \triangleq \{\text{Step}_1, \text{Step}_2\}$, in the dynamics Equation (76) are the switching action commands, i.e.,

$$
\mathbf{a}_{\text{Step}_1} = [\mathbf{f}_{\text{RF}}; \ \mathbf{f}_{\text{LR}}] \in \mathbb{R}^6, \quad \mathbf{a}_{\text{Step}_2} = [\mathbf{f}_{\text{LF}}; \ \mathbf{f}_{\text{RR}}] \in \mathbb{R}^6, \quad (77)
$$

where the $\mathbf{f}_{\text{LF}}, \mathbf{f}_{\text{RF}}, \mathbf{f}_{\text{RR}}, \mathbf{f}_{\text{LR}} \in \mathbb{R}^3$ are the ground reaction forces. While the $\widehat{\mathbf{B}}_{\sigma(t)} \in \mathbb{R}^{12 \times 6}$ denote the corresponding switching control structure matrices:

$$
\widehat{\mathbf{B}}_{\text{Step}_1} =
\begin{bmatrix}
\mathbf{O}_3 & \mathbf{O}_3 \\
\mathbf{O}_3 & \mathbf{O}_3 \\
\frac{1}{m} \cdot \mathbf{I}_3 & \frac{1}{m} \cdot \mathbf{I}_3 \\
\mathbf{I}^{-1}(\phi, \theta, \psi) \cdot [\mathbf{r}_{\text{RF}}]_\times & \mathbf{I}^{-1}(\phi, \theta, \psi) \cdot [\mathbf{r}_{\text{LR}}]_\times
\end{bmatrix},
\quad (78a)
$$

$$
\widehat{\mathbf{B}}_{\text{Step}_2} =
\begin{bmatrix}
\mathbf{O}_3 & \mathbf{O}_3 \\
\mathbf{O}_3 & \mathbf{O}_3 \\
\frac{1}{m} \cdot \mathbf{I}_3 & \frac{1}{m} \cdot \mathbf{I}_3 \\
\mathbf{I}^{-1}(\phi, \theta, \psi) \cdot [\mathbf{r}_{\text{LF}}]_\times & \mathbf{I}^{-1}(\phi, \theta, \psi) \cdot [\mathbf{r}_{\text{RR}}]_\times
\end{bmatrix},
\quad (78b)
$$

where $\mathbf{I}(\phi, \theta, \psi) \in \mathbb{R}^3$ is the robot's inertia tensor, $\mathbf{r}_{\text{LF}}, \mathbf{r}_{\text{RF}}, \mathbf{r}_{\text{LR}}, \mathbf{r}_{\text{RR}} \in \mathbb{R}^3$ denote the four foots' positions relative to CoM position, and $[\mathbf{r}_\sigma]_\times$ is defined as the skew-symmetric matrix:

$$
[\mathbf{r}_o]_\times =
\begin{bmatrix}
0 & -[\mathbf{r}_o]_z & [\mathbf{r}_o]_y \\
[\mathbf{r}_o]_z & 0 & -[\mathbf{r}_o]_x \\
-[\mathbf{r}_o]_y & [\mathbf{r}_o]_x & 0
\end{bmatrix}, \quad o \in \{\text{LF}, \text{RF}, \text{LR}, \text{RR}\}.
$$

### L.3 SIMPLIFYING MODEL-BASED DESIGNS

To have the model knowledge represented by $(\mathbf{A}, \mathbf{B})$ pertaining to robot dynamics (76), we make the following simplifications.

$$
\mathbf{R}(\phi, \theta, \psi) = \mathbf{I}_3, \quad \widetilde{\mathbf{B}} \triangleq
\begin{bmatrix}
\mathbf{O}_3 & \mathbf{O}_3 \\
\mathbf{O}_3 & \mathbf{O}_3 \\
\mathbf{I}_3 & \mathbf{O}_3 \\
\mathbf{O}_3 & \mathbf{I}_3
\end{bmatrix},
\quad (79)
$$

where the $\mathbf{R}(\phi, \theta, \psi) = \mathbf{I}_3$ is obtained through setting the zero angles of roll, pitch and yaw, i.e., $\phi = \theta = \psi = 0$.

Referring to the matrices in Equation (78), with the simplifications in Equation (79) at hand and the ignoring of unknown model mismatch of the ground-truth model, we can obtain a simplified linear model pertaining to robot dynamics (76):

$$
\frac{d}{dt}
\underbrace{
\begin{bmatrix}
\widetilde{\mathbf{p}} \\
\widetilde{\widetilde{\mathbf{e}}} \\
\widetilde{\mathbf{v}} \\
\widetilde{\mathbf{w}}
\end{bmatrix}
}_{\triangleq \widetilde{\mathbf{s}}}
=
\underbrace{
\begin{bmatrix}
\mathbf{O}_3 & \mathbf{O}_3 & \mathbf{I}_3 & \mathbf{O}_3 \\
\mathbf{O}_3 & \mathbf{O}_3 & \mathbf{O}_3 & \mathbf{I}_3 \\
\mathbf{O}_3 & \mathbf{O}_3 & \mathbf{O}_3 & \mathbf{O}_3 \\
\mathbf{O}_3 & \mathbf{O}_3 & \mathbf{O}_3 & \mathbf{O}_3
\end{bmatrix}
}_{\triangleq \widetilde{\mathbf{A}}}
\cdot
\begin{bmatrix}
\widetilde{\mathbf{p}} \\
\widetilde{\widetilde{\mathbf{e}}} \\
\widetilde{\mathbf{v}} \\
\widetilde{\mathbf{w}}
\end{bmatrix}
+ \widetilde{\mathbf{B}} \cdot \widetilde{\mathbf{u}}_{\sigma(t)},
\quad (80)
$$

where $\widetilde{\mathbf{u}}_{\text{Step}_1} \triangleq [\underline{\mathbf{f}}_{\text{RF}}; \ \underline{\mathbf{f}}_{\text{LR}}] \in \mathbb{R}^6$ and $\widetilde{\mathbf{u}}_{\text{Step}_2} \triangleq [\underline{\mathbf{f}}_{\text{LF}}; \ \underline{\mathbf{f}}_{\text{RR}}] \in \mathbb{R}^6$.

In light of the equilibrium point $\mathbf{s}^*$ in Equation (21) and $\widetilde{\mathbf{s}}$ given in Equation (80), we define $\mathbf{s} \triangleq \widetilde{\mathbf{s}} - \mathbf{s}^*$. It is then straightforward to obtain a dynamics from Equation (80) as $\dot{\mathbf{s}} = \widetilde{\mathbf{A}} \cdot \mathbf{s} + \widetilde{\mathbf{B}} \cdot \mathbf{u}_{\sigma(t)}$, which transforms to a discrete-time model via sampling technique:

$$\mathbf{s}(k+1) = \mathbf{A} \cdot \mathbf{s}(k) + \mathbf{B} \cdot \underbrace{\widetilde{\mathbf{u}}_{\sigma(k)}(k)}_{\triangleq\, \mathbf{a}_{\mathrm{phy}}(k)}, \text{ with } \mathbf{A} = \mathbf{I}_{12} + T \cdot \widetilde{\mathbf{A}} \text{ and } \mathbf{B} = T \cdot \widetilde{\mathbf{B}}, \qquad (81)$$

where $T = 0.001$ sec is the sampling period.

Considering the safety conditions in Equation (20), we obtain the safety set defined in Equation (2), where

$$\mathbf{D} = \begin{bmatrix} 0 & 0 & 0 & 0 & 0 & 1 & 0 & 0 & 0 & \mathbf{0}_3^\top \\ 0 & 0 & 1 & 0 & 0 & 0 & 0 & 0 & 0 & \mathbf{0}_3^\top \\ 0 & 0 & 0 & 0 & 0 & 0 & 1 & 0 & 0 & \mathbf{0}_3^\top \end{bmatrix}, \mathbf{v} = \begin{bmatrix} 0 \\ 0.24 \\ r_{\mathrm{x}} \end{bmatrix}, \overline{\mathbf{v}} = \begin{bmatrix} 0.17 \\ 0.13 \\ |r_{\mathrm{x}}| \end{bmatrix}, \underline{\mathbf{v}} = \begin{bmatrix} -0.17 \\ -0.13 \\ -|r_{\mathrm{x}}| \end{bmatrix}, \quad (82)$$

We now obtain the following model-based solutions, which satisfy the LMIs in Equation (7) and Equation (11).

$$\mathbf{P} = \begin{bmatrix} 0.016 & 0 & 0.023 & 0 & 0 & 0.102 & 0.003 & 0 \\ 0 & 0.015 & 0.001 & 0 & 0 & 0.002 & 0 & 0.001 \\ 0.023 & 0.001 & 226.355 & 0.078 & 0.082 & 55.206 & 0.017 & 0.003 \\ 0 & 0 & 0.078 & 0.641 & -0.001 & 0.118 & 0 & 0 \\ 0 & 0 & 0.082 & -0.001 & 0.638 & 0.125 & 0 & 0 \\ 0.102 & 0.002 & 55.206 & 0.118 & 0.125 & 247.803 & 0.045 & 0.003 \\ 0.003 & 0 & 0.017 & 0 & 0 & 0.045 & 1.065 & 0 \\ 0 & 0.001 & 0.003 & 0 & 0 & 0.003 & 0 & 0.03 \\ -0.211 & -0.008 & -314.163 & -0.73 & -0.789 & -483.681 & -0.168 & -0.021 \\ 0 & 0 & 0.006 & 0.042 & 0 & 0.004 & 0 & 0 \\ 0 & 0 & 0.003 & 0 & 0.042 & 0.003 & 0 & 0 \\ 0.002 & 0 & 1.249 & 0.003 & 0.003 & 5.547 & 0 & 0 \end{bmatrix}$$
$$\begin{bmatrix} -0.211 & 0 & 0 & 0.002 \\ -0.008 & 0 & 0 & 0 \\ -314.163 & 0.006 & 0.003 & 1.249 \\ -0.73 & 0.042 & 0 & 0.003 \\ -0.789 & 0 & 0.042 & 0.003 \\ -483.681 & 0.004 & 0.003 & 5.547 \\ 0 & 0 & 0 & 0 \\ -0.168 & 0 & 0 & 0 \\ -0.021 & 0 & 0 & 0 \\ 3229.212 & -0.049 & -0.018 & -10.901 \\ -0.049 & 0.017 & 0 & 0 \\ -0.018 & 0 & 0.017 & 0 \\ -10.901 & 0 & 0 & 0.169 \end{bmatrix},$$

$$\mathbf{F} = \begin{bmatrix} -0.1 & 0 & 0 & 0 & 0 & 0 & -40 & 0 & 0 & 0 & 0 & 0 \\ 0 & -0.1 & 0 & 0 & 0 & 0 & 0 & -30 & 0 & 0 & 0 & 0 \\ 0 & 0 & -100 & 0 & 0 & 0 & 0 & 0 & -10 & 0 & 0 & 0 \\ 0 & 0 & 0 & -100 & 0 & 0 & 0 & 0 & 0 & -10 & 0 & 0 \\ 0 & 0 & 0 & 0 & -100 & 0 & 0 & 0 & 0 & 0 & -10 & 0 \\ 0 & 0 & 0 & 0 & 0 & -100 & 0 & 0 & 0 & 0 & 0 & -30 \end{bmatrix},$$

with which and matrices $\mathbf{A}$ and $\mathbf{B}$ in Equation (81), we are able to deliver the model-based policy (4) and safety-embedded reward (8).

## L.4 TESTING EXPERIMENT: VELOCITY TRACKING PERFORMANCE

The velocity trajectories of the four models (defined in Section 7.2) running in Environments 1-4 are shown in Figure 15. Observing the Figure 15, we straightforwardly discover that the trained Phy-DRL can lead to much better performance of velocity regulation or tracking, compared with solely model-based action policies and purely data-driven DRL's action policy.

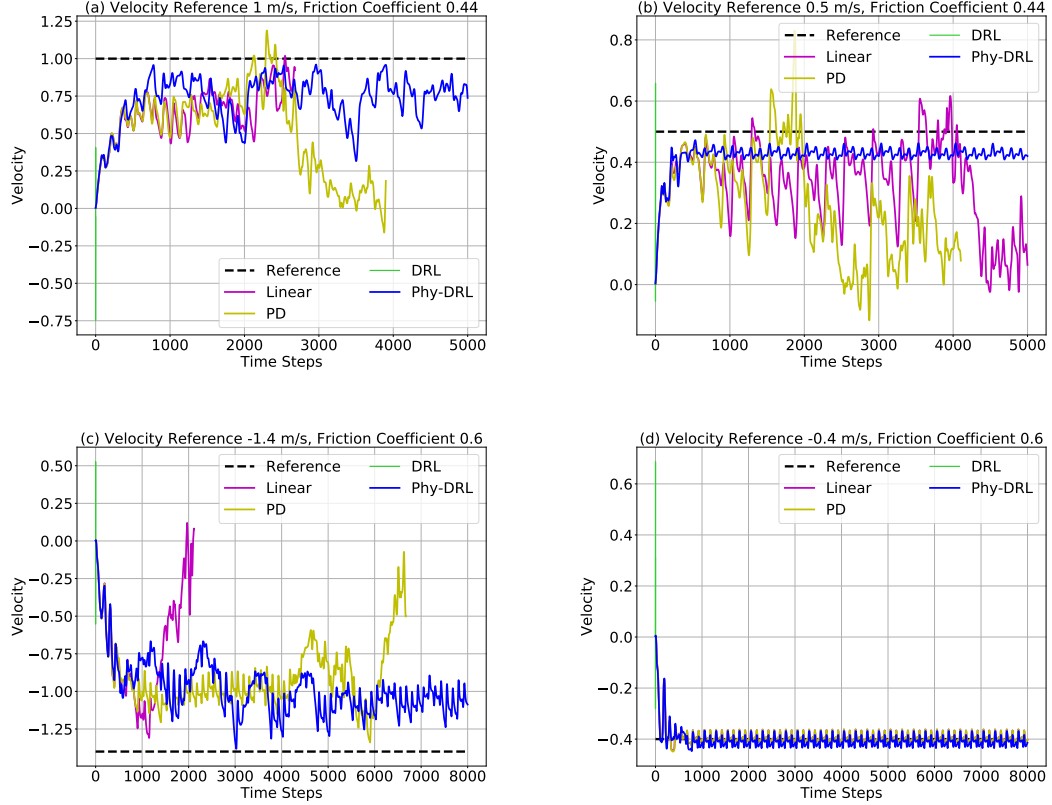

Figure 15: Velocity trajectories of quadruped robot in four different environments defined in Section 7.2.

## L.5 SAFETY-EMBEDDED REWARDS

For the aim of a mathematically-provable safety guarantee, the reward that is most similar to ours is the CLF reward proposed in Westenbroek et al. (2022). We also perform the comparisons. To simplify the comparisons, we do not consider the high-performance sub-reward, which means both rewards degrade to

$$\text{Ours}: \mathcal{R}(\mathbf{s}(k), \mathbf{a}_{\text{drl}}(k)) = \mathbf{s}^\top(k) \cdot (\overline{\mathbf{A}}^\top \cdot \mathbf{P} \cdot \overline{\mathbf{A}}) \cdot \mathbf{s}(k) - \mathbf{s}^\top(k+1) \cdot \mathbf{P} \cdot \mathbf{s}(k+1), \quad (83)$$

$$\text{CLF Reward}: \mathcal{R}(\mathbf{s}(k), \mathbf{a}_{\text{drl}}(k)) = \mathbf{s}^\top(k) \cdot \mathbf{P} \cdot \mathbf{s}(k) - \mathbf{s}^\top(k+1) \cdot \mathbf{P} \cdot \mathbf{s}(k+1). \quad (84)$$

## L.6 PHYSICS-KNOWLEDGE-ENHANCED CRITIC NETWORK

To apply the NN editing, we first obtain the available knowledge about the actor-value function and action policy. Referring to Equation (5), the action-value function can be re-denoted by

$$\mathcal{Q}^\pi(\mathbf{s}(k), \mathbf{a}_{\text{drl}}(k)) = \mathcal{Q}^\pi(\mathcal{R}(\mathbf{s}(k), \mathbf{a}_{\text{drl}}(k))). \quad (85)$$

According to Taylor's theorem in Appendix G, expanding the action-value function in Equation (85) around the (one-dimensional real value) $\mathcal{R}(\mathbf{s}(k), \mathbf{a}_{\text{drl}}(k))$, and recalling Equation (83), we conclude the action-value function does not include any odd-order monomials of $[\mathbf{s}(k)]_i, i = 1, \ldots, 12$, and is independent of $\mathbf{a}_{\text{drl}}(k)$. The critic network shall strictly comply with the knowledge. This knowledge compliance can be achieved via our proposed NN editing. We do not obtain the invariant knowledge about action policy in this example. In other words, according to our analysis, the action policy depends on all the elements of the system state $\mathbf{s}(k)$. So, in this example, we only need to design a physics-knowledge-enhanced critic network.

The architecture of the considered physics-knowledge-enhanced critic network in this example is shown in Figure 16, where the PhyN architecture is given in Figure 2 (b). We compare the performance of physics-knowledge-enhanced critic networks with fully-connected multi-layer perceptron (FC MLP). Figure 16 shows that different critic networks can be obtained by only changing the output dimensions $n$. We here consider three models: physics-knowledge-enhanced critic network with $n = 10$ (PKN-10), physics-knowledge-enhanced critic network with $n = 15$ (PKN-15), and physics-knowledge-enhanced critic network with $n = 20$ (PKN-20). The parameter numbers of all network models are summarized in Table 2.

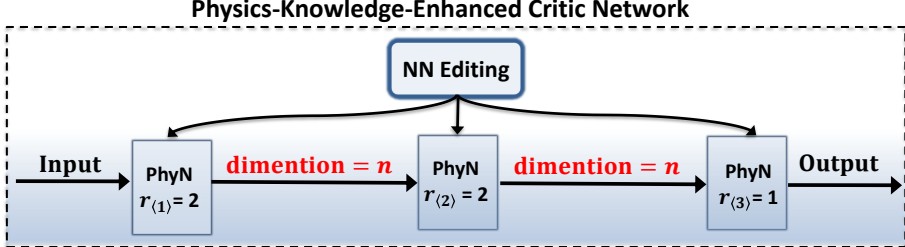

Figure 16: Considered physics-knowledge-enhanced critic network.

The trajectories of episode reward of the four models in Table 2 are shown in Figure 17, which, in conjunction with Table 2, show that except for the smallest knowledge-enhanced critic network (i.e., model PKN-10), other networks (i.e., models PKN-15 and PKN-20) outperform the very large FC MLP model, viewed from perspectives of parameter numbers, episode reward and stability of training. This, on the other hand, implies that the physics-knowledge-enhanced critic network can avoid significant spurious correlations via NN editing.

Table 2: Model Parameters

| Model ID | Layer 1 #weights | Layer 1 #bias | Layer 2 #Weights | Layer 2 #bias | Layer 3 #weights | Layer 3 #bias | Layer 4 #weights | Layer 4 #bias | #sum |
|---|---|---|---|---|---|---|---|---|---|
| PKN-10 | 1710 | 10 | 650 | 10 | 10 | 1 | — | — | 2391 |
| PKN-15 | 2565 | 15 | 2025 | 15 | 15 | 1 | — | — | 4636 |
| PKN-20 | 3420 | 20 | 4600 | 20 | 20 | 1 | — | — | 8081 |
| FC MLP | 2304 | 128 | 16384 | 128 | 16384 | 128 | 128 | 1 | 35585 |

### L.7  REWARD COMPARISONS

We next compare the two rewards in Equation (83) and Equation (84) from the perspectives of design differences and experiments. To have a fair experimental comparison, we compare the two rewards in the same Phy-DRL package. In other words, we use two Phy-DRL models to train the robot; the only difference is their reward: one is our proposed reward in Equation (83) while the other one is the CLF reward in Equation (84).

### L.7.1  COMPARISON: DESIGN DIFFERENCES

Along the ground-truth model of real plant (1) with the consideration of Equation (3), Equation (4) and Equation (9), we have

$$
\begin{aligned}
&\mathbf{s}^\top(k+1) \cdot \mathbf{P} \cdot \mathbf{s}(k+1) \\
&= (\mathbf{B} \cdot \mathbf{a}_{\mathrm{drl}}(k) + \mathbf{f}(\mathbf{s}(k), \mathbf{a}(k)))^\top \cdot \mathbf{P} \cdot (\mathbf{B} \cdot \mathbf{a}_{\mathrm{drl}}(k) + \mathbf{f}(\mathbf{s}(k), \mathbf{a}(k))) \\
&\quad + 2\mathbf{s}^\top(k) \cdot \overline{\mathbf{A}}^\top \cdot \mathbf{P} \cdot (\mathbf{B} \cdot \mathbf{a}_{\mathrm{drl}}(k) + \mathbf{f}(\mathbf{s}(k), \mathbf{a}(k))) + \mathbf{s}^\top(k) \cdot (\overline{\mathbf{A}}^\top \cdot \mathbf{P} \cdot \overline{\mathbf{A}}) \cdot \mathbf{s}(k).
\end{aligned} \tag{86}
$$

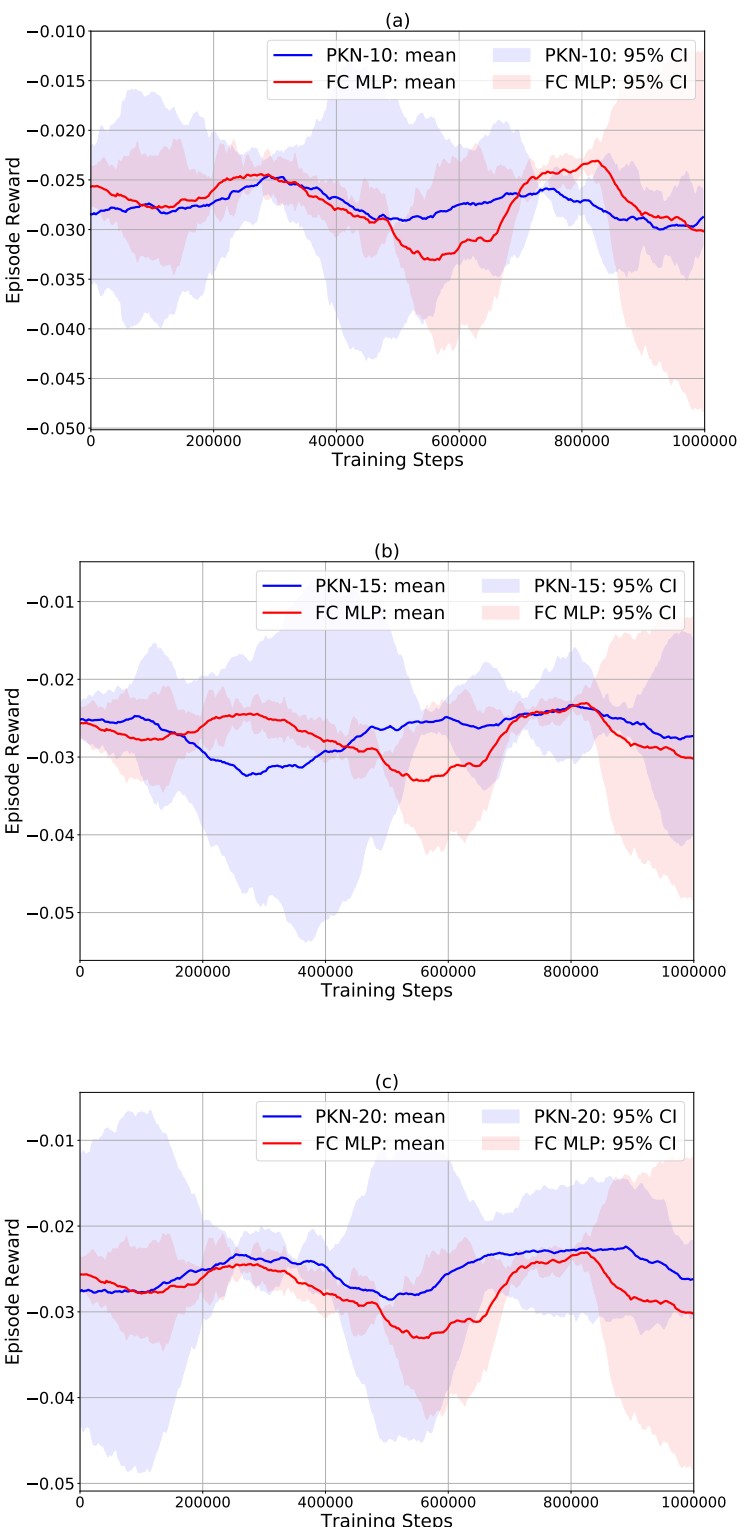

Figure 17: Trajectories of episode reward in training: smoothing rate 0.15 and 3 random seeds.

Table 3: Episode Numbers

|  | Ours | | CLF Reward | |
| --- | --- | --- | --- | --- |
| Model ID | PKN-15 | PKN-20 | PKN-15 | PKN-20 |
| Episode Number: average | 351 | 348 | 376 | 409 |

We next define two invariants and one unknown:

$$\text{invariant-1} \triangleq \mathbf{s}^\top(k) \cdot \overline{\mathbf{A}}^\top \cdot \mathbf{P} \cdot \overline{\mathbf{A}} \cdot \mathbf{s}(k), \tag{87}$$

$$\text{invariant-2} \triangleq \mathbf{s}^\top(k) \cdot \left(\mathbf{P} - \overline{\mathbf{A}}^\top \cdot \mathbf{P} \cdot \overline{\mathbf{A}}\right) \cdot \mathbf{s}(k). \tag{88}$$

$$\text{unknown} \triangleq \left(\mathbf{B} \cdot \mathbf{a}_{\text{drl}}(k) + \mathbf{f}(\mathbf{s}(k), \mathbf{a}(k))\right)^\top \cdot \mathbf{P} \cdot \left(\mathbf{B} \cdot \mathbf{a}_{\text{drl}}(k) + \mathbf{f}(\mathbf{s}(k), \mathbf{a}(k))\right)$$
$$+ 2\mathbf{s}^\top(k) \cdot \overline{\mathbf{A}}^\top \cdot \mathbf{P} \cdot \left(\mathbf{B} \cdot \mathbf{a}_{\text{drl}}(k) + \mathbf{f}(\mathbf{s}(k), \mathbf{a}(k))\right). \tag{89}$$

We note the formulas in Equation (87) and Equation (88) are named as '*invariants*', because all the terms in their right-hand sides (i.e., designed matrices $\overline{\mathbf{A}}$ and $\mathbf{P}$) are known to us and their properties are not influenced by training. While the formula in Equation (89) is defined as '*unknown*' since the terms in its right-hand side are unknown to us due to the unknown model mismatch $\mathbf{f}(\mathbf{s}(\mathbf{k}), \mathbf{a}(\mathbf{k}))$ and unknown data-driven action policy $\mathbf{a}_{\text{drl}}(k)$ during training.

Using the definitions in Equation (87)-Equation (89), the formula in Equation (86) is rewritten as

$$\mathbf{s}^\top(k+1) \cdot \mathbf{P} \cdot \mathbf{s}(k+1) = \text{unknown} + \text{invariant-1},$$

by which and recalling Equation (87)-Equation (89), the two rewards in Equation (83) and Equation (84) are equivalently rewritten as

$$\text{Ours: } \mathcal{R}(\mathbf{s}(k), \mathbf{a}_{\text{drl}}(k)) = \text{invariant-1} - \mathbf{s}^\top(k+1) \cdot \mathbf{P} \cdot \mathbf{s}(k+1) = -\text{unknown}, \tag{90}$$

$$\text{CLF Reward: } \mathcal{R}(\mathbf{s}(k), \mathbf{a}_{\text{drl}}(k)) = \mathbf{s}^\top(k) \cdot \mathbf{P} \cdot \mathbf{s}(k) - \mathbf{s}^\top(k+1) \cdot \mathbf{P} \cdot \mathbf{s}(k+1)$$
$$= \text{invariant-2} - \text{unknown}. \tag{91}$$

Observing the formulas in Equation (90) and Equation (91), we discover a critical difference between our proposed reward in Equation (83) and the CLF reward in Equation (84): our reward decouples invariant and unknown for learning (i.e., data-driven DRL only learn the unknown), while the CLF reward mixes invariant and unknown (i.e., data-driven DRL learn both unknown and invariant).

### L.7.2 Comparison: Training

We next present the training behavior. We note our reward in Equation (90) and the CLF reward in Equation (91) have different scales. To present fair comparisons, we process the raw episode reward via unionization. Specifically, the proceeded episode reward called the united episode reward, is defined as

$$\overline{\mathcal{R}}(m) \triangleq \frac{\mathcal{R}(m)}{\min_{r=1,2,\cdots} \left\{|\underline{\mathcal{R}}(r)|\right\}}, \tag{92}$$

where $\underline{\mathcal{R}}(m)$ denotes the raw episode reward at the episode index $m$.

We consider two models, PKN-15 and PKN-20, whose network configurations are summarized in Table 2. Each model has three seeds. For all the training of DRL and Phy-DRL, we set the maximum step number of an episode as 10200, while an episode will terminate if the |CoM height| $\leq 0.12$ m (robot falls). The two models' averages of episode number over the $10^6$ training steps and the three random seeds are presented in Table 3. The smaller average value therein means the more successful running time of the robot or the fewer times the robot falls. Meanwhile, the trajectories of the processed episode rewards are shown in Figure 18, observing which and Table 3, we can discover that our proposed reward leads to much more stable and safe training. The root reason can be that our reward decouples invariant and unknown and only lets data-driven DRL learn the unknown defined in Equation (89).

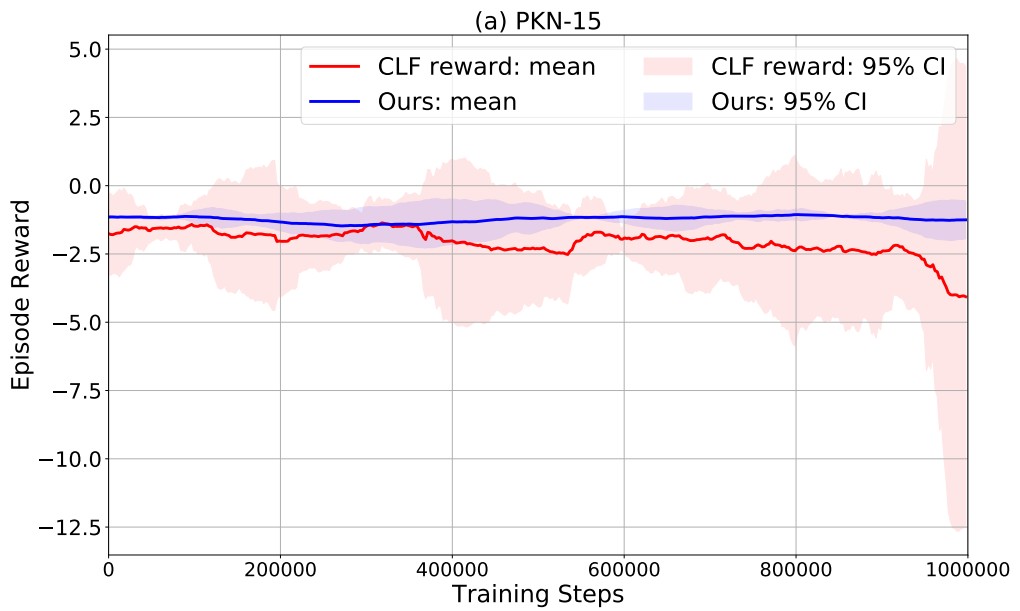

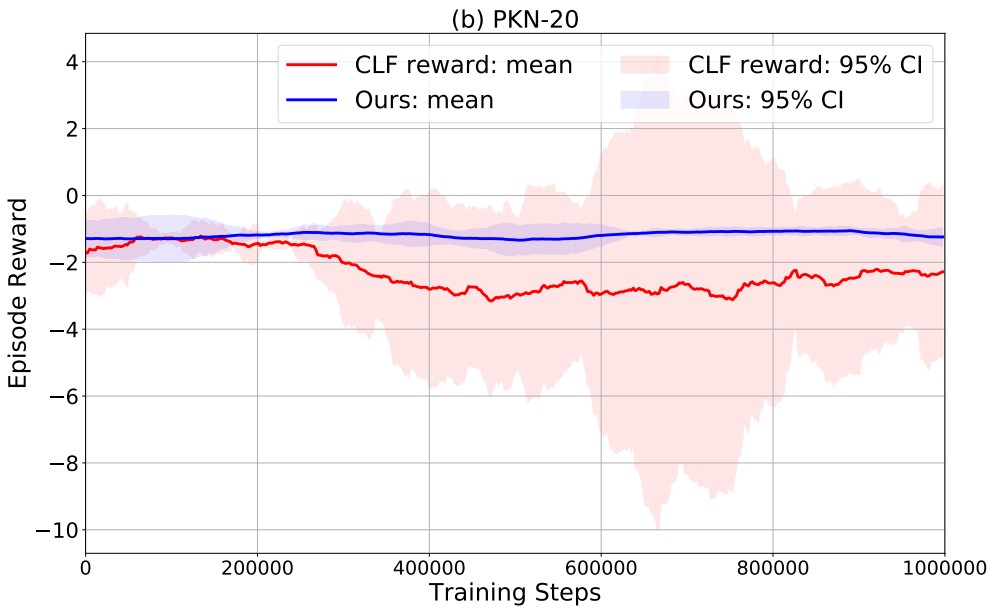

Figure 18: Trajectories of united episode reward: smoothing rate 0.15 and 3 random seeds.

### L.8 TRAINING

For the code, we use the Python API for the TensorFlow framework Kingma & Ba and the Adam optimizer Abadi et al. for training. This project is using the settings: 1) Ubuntu 22.04, 2) Python 3.7, 3) TensorFlow 2.5.0, 4) Numpy 1.19.5, and 5) Pybullet.

For Phy-DRL, the observation of the policy is a 12-dimensional tracking error vector between the robot's state vector and the mission vector. The agent's actions offset the desired positional and lateral accelerations generated from the model-based policy. The computed accelerations are then converted to the low-level motors' torque control.

The policy is trained using DDPG algorithm Lillicrap et al. (2016). The actor and critic networks are implemented as a Multi-Layer Perceptron (MLP) with four fully connected layers. The output dimensions of the critic network are 256, 128, 64, and 1. The output dimensions of actor networks are 256, 128, 64, and 6. The input of the critic network is the tracking error vector and the action vector. The input of the actor network is the tracking error vector. The activation functions of the first three neural layers are ReLU, while the output of the last layer is the Tanh function for the actor network and Linear for the critic network.

We let discount factor $\gamma = 0.2$, and the learning rates of critic and actor networks are the same as 0.0003. We set the batch size to 300. The maximum step number of one episode is 10200. Each weight matrix is initialized randomly from a (truncated) normal distribution with zero mean and standard deviation, discarding and re-drawing any samples more than two standard deviations from the mean. We initialize each bias according to the normal distribution with zero mean and standard deviation.

### L.9 LINKS: DEMONSTRATION VIDEOS

**Environment 1)**: velocity command: $r_x = 1$ m/s and snow road. A demonstration video is available at `https://www.youtube.com/watch?v=tspPMbZwfig&t=1s`.

**Environment ii)**: velocity command: $r_x = 0.5$ m/s and snow road. A demonstration video is available at `https://www.youtube.com/watch?v=BK8k92jahfI&t=21s`.

**Environment iii)**: velocity command: $r_x = -1.4$ m/s and wet road. A demonstration video is available at `https://www.youtube.com/shorts/gbC-CwqGj78`.

**Environment iv)**:velocity command: $r_x = -0.4$ m/s and wet road. A demonstration video is available at `https://www.youtube.com/shorts/UwQYRveLJUs`.

