# OpenReview forum: "Physics-Regulated Deep Reinforcement Learning: Invariant Embeddings"
_ICLR.cc/2024/Conference — ICLR 2024 spotlight_

### Official Review · Reviewer_bsXP · 2023-10-31

**Soundness:** 3 good
**Presentation:** 3 good
**Contribution:** 3 good
**Rating:** 8
**Confidence:** 2

**Summary:**

This work proposes three invariant-embedding principles to achieve a physics-regulated deep RL framework for safety-critical systems. The three principles include residual action policy, safety-embedded reward, and physics-model-guided neural network editing & augmentation. The main claim on safety-embedded reward and the editing are supported by theoratical proofs. Experimental results show that the proposed Phy-DRL achieve better performance in simulated cartpole and quadruped robot environments

**Strengths:**

- This work combines the safety requirement with neural network editing, which is novel to my understanding

- This work provides proofs to support the designed safety-embedded reward.

- The experimental results show that the proposed method achieves some good policies that follow safety requirements.

**Weaknesses:**

1. It's unclear how critical the residual physics policy is needed.

2. The proposed method is only compared to methods without consideration on safety requirements.

3. The method relies on known state definitions. If the state can not be obtained from the environment or definition, how this framework will be affected?

**Questions:**

Please see weaknesses part.

---

> ### Author Response · Authors · 2023-11-13
> **Response to Reviewer bsXP: Q1 (Part I)**
>
> We thank Reviewer bsXP for the positive feedback and insightful comments. Our answers to the three questions will address the concerns and clarify the paper's contributions. We have also updated our paper to clarify the points that the reviewer raised. Let us know if our response clarifies the unclear points.
>
> **Q1: It's unclear how critical the residual physics policy is needed.**
>
> $\textbf{A1:}$ The proposed residual action policy contributes to three distinguished traits of Phy-DRL: 1)  automatic construction of safety-embedded reward, 2) fast training, and 3) mathematically-provable safety guarantee. In the updated paper, how the residual policy is leveraged to derive the condition of mathematically-provable safety guarantee is formally presented in Appendix D.1. For this reason, in the discussions below, we explain the critical roles or functions of residual policy in delivering the traits 1) and 2) only.
>
> $$------------------------------------------------------------------$$
>
> **Trait 1):  Automatic Construction of Safety-embedded Reward.**
> Given many safety conditions, our proposed Phy-DRL can automatically and simultaneously construct the safety-embedded reward and compute the model-based action policy. To review this, we first recall our proposed safety-embedded sub-reward:
> $$r({s}(k), {a}(k)) = s^\top(k) \cdot {\overline{{A}}^\top \cdot {P} \cdot \overline{{A}} } \cdot {s}(k) - {{s}^\top(k+1)} \cdot {P} \cdot {s}(k+1), \text{with } \overline{{A}} \buildrel \Delta \over = {A} + {{B} } \cdot {{F}}.$$
> Meanwhile, the model-based action policy in the residual policy diagram is ${a}_{\text{phy}}(k) = {F} \cdot {s}(k).$
> We here conclude that model-based action policy in the residual policy diagram, i.e., $F$, is directly used in the safety-embedded reward formula. Besides, we note in the safety-embedded reward, $A$ and $B$ are known and available model knowledge, while $F$ and $P$ are to be designed. In our Phy-DRL framework, they are automatically and simultaneously obtained via Python’s CVXPY toolbox for solving an analytic centering problem. Specifically, the CVXPY toolbox first solves:
>
> ${\text{argmin}_{Q,R}} ~ ( {\log \det \left( {{Q}^{ - 1}} \right)}) \text{ subject to (7) and (11),}$
> where Equations (7) and (11) of the paper include the transformed safety conditions. With the obtained $Q$ and $R$ at hand, $F$ and $P$ are then obtained as  $F = R \cdot Q^{-1}, ~~~~ P = Q^{-1}.$ We here can further conclude that model-based action policy and safety-embedded reward are not separated. Their computations depend on each other under safety conditions formulated in Equations (7) and (11) of the paper.
>
> $$------------------------------------------------------------------$$
>
> **Trait 2): Fast Training Towards Safety Guarantee.**  To understand how our residual action policy leads to fast training, we review two system dynamics.
>
>  $\text{System dynamics with purely data-driven DRL policy:} ~\mathbf{s}(k+1) = A \cdot
>                                              s(k) + B \cdot a_{\text{drl}}(k)  + f(s(k)), ~k \in \mathbb{N}. ~~~(\text{e}1)$
>
> $\text{System dynamics with residual policy:} ~\mathbf{s}(k+1) = A \cdot s(k) + B \cdot (a_{\text{drl}}(k) + a_{\text{phy}}(k)) + f(s(k)), ~k \in \mathbb{N}~~~(\text{e}2)$
>
> where $a_{\text{drl}}(k)$ and $a_{\text{phy}}(k)$  denote data-driven and model-based actions, respectively.
>
> To review the training process from the perspective of safety guarantee, we construct a positive or energy function:
> $ {V}\left(\mathbf{s}(k) \right) = {\mathbf{s}^\top(k)}\cdot {\mathbf{P}} \cdot \mathbf{s}(k)， ~~~{\mathbf{P}} \succ 0.~~~(\text{e}3)$
>
> Along with system dynamics (e1), we have:
> $V( {{s(k+1)}}) - V( {{s(k)}}) = (B \cdot a_{\text{drl}}(k)  + f(s(k)))^{\top} \cdot P \cdot  (B \cdot a_{\text{drl}}(k)  + f(s(k))) + 2(A \cdot s(k))^{\top} \cdot P \cdot  (B \cdot a_{\text{drl}}(k)  + f(s(k))) + ~s^{\top} (k) \cdot (A^{\top} \cdot  P \cdot A) \cdot  s(k). ~~~(\text{e}4)$
>
> We note in our proposed residual policy diagram, the physics-model-based action policy is $a_{\text{phy}}(k) = F \cdot s(k)$, where the  $F$ is computed automatically (see Remark 5.5 in the paper), such that it renders the following inequality holds:
>
> $\overline{{A}}^\top \cdot {P} \cdot \overline{{A}} - {P} \prec 0, ~~\text{with }~\overline{{A}} \buildrel \Delta \over = {A} + {{B} } \cdot {{F}}. ~~~(\text{e}5)$
>
> With this designed property (e5) at hand, along with system dynamics (e2), we have
> $V( {{s(k+1)}}) - V( {{s(k)}}) = (B \cdot a_{\text{drl}}(k)  + f(s(k)))^{\top} \cdot P \cdot  (B \cdot a_{\text{drl}}(k)  + f(s(k))) + 2( \overline{A} \cdot s(k))^{\top} \cdot P \cdot  (B \cdot a_{\text{drl}}(k)  + f(s(k))) + s^{\top} (k) \cdot ( \overline{{A}}^{\top} \cdot  P \cdot  \overline{{A}}) \cdot  s(k). ~~~(\text{e}6)$

---

> ### Author Response · Authors · 2023-11-14
> **Response to Reviewer bsXP: Q1 (Part II)**
>
> With the consideration of designed model-based property (e5), through comparing (e4) and (e6),  we can discover that $V( {{{s}(k+1)}}) - V( {{{s}(k)}})$ under the residual policy, i.e. the (e6), has a constant negative term:  ${{{s}}^\top}(k) \cdot ( {{{\overline A}}^\top \cdot {{P}} \cdot {{{{\overline A}}}}} - {P}) \cdot {{s}}(k) < 0$. While  ${{{s}}^\top}(k) \cdot ( {{{A}}^\top \cdot {{P}} \cdot {{{{A}}}}} - {P}) \cdot {{s}}(k) < 0$ is not guaranteed in the  $V( {{\mathbf{s}(k+1)}}) - V( {{\mathbf{s}(k)}})$ under purely data-driven DRL policy (i.e.,  the (e4)). In addition, $V( {{\mathbf{s}(k+1)}}) - V( {{\mathbf{s}(k)}}) \le 0$  holds for $\forall k \in \mathbb{N}$ is desired for safety guarantee. Because it, in conjunction with the safety envelope (defined as state space that satisfies $s^\top \cdot P \cdot s < 1$), means the system states never leave the safety envelope (starting inside the safety envelope). Hereto, we can conclude that the residual policy can lead to fast training because it introduces a global attractor inside the safety envelope (through the model-based design property (e5)), such that  $V( {{{s}(k+1)}}) - V( {{{s}(k)}}) \le 0 $ is much more quickly to occur. This phenomenon can be depicted by the figure available at the [link (anonymous hosting and browsing)](https://www.dropbox.com/scl/fi/qp1hgpmdiw68mg04ypute/reubexp.pdf?rlkey=fbv7d8cr6hfd51kmozrhz5vzk&dl=0) (and also Figure 6 in Appendix D.2 of the paper).  Both figures illustrate that the model-based property of residual policy (e5) can be understood as putting a global attractor inside the safety envelope, which generates an attracting force toward the safety envelope. Obviously, in any training stage, because of the attracting force that always exists, system states under the control of residual policy are much more likely to stay inside the safety envelope.
>
> Motivated by the insightful question from the reviewer, we have added a subsection entitled Appendix D.2 Extension: Explanation of Fast Training to the paper, right after the proof of mathematically-provable safety guarantee.

---

> ### Author Response · Authors · 2023-11-14
> **Response to Reviewer bsXP: Q2**
>
> **Q2: The proposed method is only compared to methods without consideration on safety requirements.**
>
> $\textbf{A2:}$ Insufficient information in experiments leads to the confusion. All the comparison models, excluding the default proportional-derivative controller, considered the safety requirements. Our Phy-DRL and purely data-driven DRL share the same safety-embedded reward, and our Phy-DRL and sole model-based design share the same model-based action policy: $a_{\text{phy}}(k) = F \cdot s(k),$ which is obtained under safety conditions. Specifically, $F$ is computed according to:
>
> Step 1: $\text{argmin}_{Q,R} ~(\text{logdet}(Q^{-1})), \text{ subject to (7) and (11)}$,
>
> Step 2: $F = R \cdot Q^{-1}$,
>
> where Equations (7) and (11) in the paper include the transformed safety conditions. In addition, during training, the DRL and our Phy-DRL share the same termination condition of an episode.
>
> To remove the confusion, we have added the information to the updated paper.

---

> ### Author Response · Authors · 2023-11-14
> **Response to Reviewer bsXP: Q3**
>
> **Q3: The method relies on known state definitions. If the state can not be obtained from the environment or definition, how will this framework be affected?**
>
> $\textbf{A3:}$ In our proposed Phy-DRL framework, the data-driven DRL action policy and model-based action policy share the same real-time system state data for action computations. For our testbeds of autonomous systems, including autonomous cars, quadruped robots and drones, the available system states that the sensors directly output typically include *CoM x-position*, *CoM y-position*, *CoM z-height*, *roll*, *pitch*, *yaw*, *CoM x-velocity*, *CoM y-velocity*, *CoM z-velocity*, *velocities of roll, yaw and pitch*, car's longitudinal velocity, and car's wheel (angular) velocities. All these sensor data cover the needed system state of the general dynamics models of autonomous cars [1], quadruped robots [2], and drones [3].
>
> For our application systems, we currently do not face the problem of some unavailable state from the environment or definition. If this problem happens, intuitively, the physical model for our model-based design in Phy-DRL may not be available or have a large model mismatch that leads to an infeasible model-based action policy and safety-embedded reward. To address this problem, a practical approach is to use currently available real-time state data from the environment to obtain or learn a corresponding model represented by $(A, B)$, via system identification (e.g., non-asymptotic identification using a single and finite-time trajectory, proposed in [4]). The newly identified $(A, B)$ are then used for constructing the safety-embedded reward and designing the model-based action policy for Phy-DRL.
>
> $\left[ 1 \right]$ Rajamani, R. (2011). Vehicle dynamics and control. Springer Science & Business Media.
>
> $\left[ 2 \right]$ Di Carlo, J., Wensing, P. M., Katz, B., Bledt, G., & Kim, S. (2018, October). Dynamic locomotion in the MIT cheetah 3 through convex model-predictive control. In 2018 IEEE/RSJ international conference on intelligent robots and systems  (pp. 1-9).
>
> $\left[ 3 \right]$ Six, D., Briot, S., Chriette, A., & Martinet, P. (2017). The kinematics, dynamics and control of a flying parallel robot with three quadrotors. IEEE Robotics and Automation Letters, 3(1), 559-566.
>
> $\left[ 4 \right]$ Oymak, S., & Ozay, N. (2019, July). Non-asymptotic identification of LTI systems from a single trajectory. In 2019 American control conference (pp. 5655-5661).

---

> > ### Comment · Reviewer_bsXP · 2023-11-16
> > **Thanks for the clarifications**
> >
> > Dear authors,
> >
> > Thanks for those clarifications. Conceptually they addressed Q2 and Q3 well. I need to further check Q1 and questions raised by other reviewers.
> >
> > For fast training property mentioned in the discussion on Q1, it would be great to provide experimental results about improvements on either sample efficiency or wall-clock time.

---

> > > ### Author Response · Authors · 2023-11-21
> > > **Nearing end of discussion: any remaining concerns?**
> > >
> > > Dear Reviewer bsXP,
> > >
> > > We thank you again for your time and effort in reviewing our work! As the discussion period draws close (less than two days), we would like to know if we have addressed your concerns. If you have any more questions, please do not hesitate to let us know.
> > >
> > > Sincerely,
> > >
> > > Authors

---

> > > > ### Comment · Reviewer_bsXP · 2023-11-21
> > > >
> > > > Thanks for the experiments. I would like to raise the score to accept while I'm not very certain about the derivations.

---

> > > > > ### Author Response · Authors · 2023-11-22
> > > > >
> > > > > Dear Reviewer,
> > > > >
> > > > > We appreciate your recognition. Our paper has many mathematical works, but they are very critical in developing our framework. If you are interested in some parts, we will be very happy to continue our discussion and share very detailed explanations, even after the discussion stage.
> > > > >
> > > > > Sincerely,
> > > > >
> > > > > Authors

---

> ### Author Response · Authors · 2023-11-16
> **Reply to Reviewer bsXP**
>
> Dear Reviewer bsXP,
>
>
> We thank you for the very critical suggestion. We will add the wall-clock time to demonstrate the fast training towards safety guarantee further. We will let you know when the experimental results are added to the paper.
>
>
> Sincerely,
>
> Authors of paper.

---

> ### Author Response · Authors · 2023-11-18
> **Reply to Reviewer bsXP**
>
> **For fast training property mentioned in the discussion on Q1, it would be great to provide experimental results about improvements on either sample efficiency or wall-clock time.**
>
> $\textbf{Answer:}$ Appendix K.7 in the updated paper includes experimental results with information of sample efficiency and wall-clock time. For convenience, the experimental results are also available at [link (anonymous hosting and browsing)](https://www.dropbox.com/scl/fi/yi1u5ui3jlpayf9f230ys/commbrl.pdf?rlkey=fq9ttlu0fzasy4d46kqq2jhd5&dl=0). We saved the learned agent at 50K, 75K, 100K, and 200K steps for safety evaluation.
>
> As depicted in the plots at [link (anonymous hosting and browsing)](https://www.dropbox.com/scl/fi/yi1u5ui3jlpayf9f230ys/commbrl.pdf?rlkey=fq9ttlu0fzasy4d46kqq2jhd5&dl=0), our proposed phy-DRL framework in the model-free setting (Figure. a) demonstrates the capability to render the safety envelope invariant after just 50K training steps (note: we adopt step-wise optimization, where 50K training steps also mean 50K interaction steps in model-free setting). Phy-DRL trained in the model-based setting (Figure. b) can not render the safety envelope invariant after 50k training steps. This is attributed to the inherent inaccuracy of the model used in the model-based setting, leading to suboptimal agent performance in the true test MDP.
>
> In contrast, the purely data-driven DRL requires more training steps to learn in both model-free (Figure. c) and model-based settings (Figure. d). The evaluation results at 50K and 75K steps indicate unstable training progress without convergence. However, after training for 100K/200K steps, performance improves, though not as effectively as with Phy-DRL.
>
> To answer the question of sampling efficiency (interaction steps), Phy-DRL achieves convergence within 50K interaction steps (Figure. a), while traditional DRL requires considerably more steps (Figure. c). However, comparing model-free to model-based settings, the latter requires zero interaction with the true MDP, as the model is predetermined rather than learned. In real-world applications, additional interaction steps are typically needed to gather useful data for model learning/updating. This process may need to be repeated multiple times until the agent converges. Moreover, in some scenarios, data collection isn't straightforward, making it challenging to design an effective controller initially for data collection purposes. How many interaction steps are needed to improve the model is also dependent on the algorithm of updating the model and the policy, which makes the comparison of interaction steps challenging.
>
> From a wall-clock time perspective (training steps), the plots illustrate that Phy-DRL achieves convergence after 50K training steps, making it the most time-efficient approach.
>
> $$$$
>
> **I need to further check Q1 and questions raised by other reviewers.**
>
> $\textbf{Answer:}$ We have completed the first-round responses to all reviewers' comments. They are ready for your review. We look forward to your new feedback.
>
> Sincerely,
>
> Authors of paper.

---

### Official Review · Reviewer_BXs3 · 2023-10-31

**Soundness:** 3 good
**Presentation:** 3 good
**Contribution:** 3 good
**Rating:** 8
**Confidence:** 3

**Summary:**

This paper proposes a physics-regulated DRL framework for safety-critical tasks. The framework contains 1) residual action policy fusing model-based and model-free policies 2) safety-embedded reward by incorporating the knowledge from the approximated linear model dynamics 3) physics-enforced actor-critic algorithm including editing and activation editing, and achieves mathematically-provable safety guarantee.

**Strengths:**

1. The paper is well-motivated and well-organized. It is in general easy to follow.
2. The approach, as the reviewer can tell, is sound and solid with remarkable provable safety guarantees.
3. This framework is novel, specifically for the part of RL plus linearised model with a systematical reward construction approach to achieve provable safety guarantees.
4. The experiments are significant compared to pure DRL and pure model-based approach.

**Weaknesses:**

1. In general, the paper is easy to follow. However,  Section 6 can be a bit confusing to the reviewer and needs back-and-forth checking while reading. This section can be improved by adding more overview, intuition, and connecting intro between and among subsections. It is somewhat difficult to accept a bunch of symbols such as in Equation (13), (14), (15), (16).
2. It is somewhat hard to understand Algorithms (1) and (2). There is a lot of white space in Algorithm (1) and (2), why don't use it to add an overview, explanations, and comments for better understanding?
3. In the experiments, the authors compared their approach to model-free RL, model-based control. For the reviewer, It is a bit unfair as the authors did not compare their approach to model-based RL such as RL + linearised model.

**Questions:**

The reviewer is a bit confused about the Section 6,
Question 1: How does NN input augmentation help embed what physics knowledge into AC?
2: how does NN editing help embed what knowledge into AC?
3. How do algorithms (1)(2) achieve NN input aug and NN editing?

---

> ### Author Response · Authors · 2023-11-16
> **Response to Reviewer BXs3: Part I**
>
> We appreciate the reviewer's valuable comments and suggestions, which helped improve the clarity and readability of the paper.
> We hope we have answered the reviewer's questions clearly and satisfactorily, and are eager for further discussion.
> $$$$
>  **W1: In general, the paper is easy to follow. However, Section 6 can be a bit confusing to the reviewer and needs back-and-forth checking while reading. This section can be improved by adding more overview, intuition, and connecting intro between and among subsections. It is somewhat difficult to accept a bunch of symbols such as in Equation (13), (14), (15), (16).**
>
> $\textbf{Answer:}$ In the updated paper, we have reorganized Section 6 to include more detailed intuition/motivations, connecting paragraphs, and an overview. To improve the readability of Equations (13), (14), (15), (16), we added corresponding examples. Due to the page limit, the examples related to Equations (13) and (15) are moved to Appendix H.
> $$$$
>  **W2: It is somewhat hard to understand Algorithms (1) and (2). There is a lot of white space in Algorithm (1) and (2), why don't use it to add an overview, explanations, and comments for better understanding?**
>
> $\textbf{Answer:}$ In the updated paper, more detailed explanations and comments have been added to Algorithm 1. However, Algorithm 2 needs a large space for a detailed overview, explanations, and comments. Due to the paper limit, they are included in Appendix I.
> $$$$
> **W3: In the experiments, the authors compared their approach to model-free RL, model-based control. For the reviewer, It is a bit unfair as the authors did not compare their approach to model-based RL such as RL + linearized model.**
>
> $\textbf{Answer:}$ We would like to recall that model-based RL involves using a  model (whether learned or known) to predict future states for optimizing policies. Within the model-based RL framework, the model is used primarily to generate states, allowing the agent to query the Markov Decision Process at any desired state-action pair to train the action policy [1].
>
> In our Phy-DRL, we introduce an alternative approach for using a model. A linear model is used to deliver Phy-DRL's three distinguished traits (rather than state prediction): 1) mathematically-provable safety guarantee, 2) automatic construction of safety-embedded reward, and 3) fast training towards safety guarantee. From a learning perspective, both model-based DRL and our proposed Phy-DRL can expedite the learning process with fewer interaction steps. However, we note the limitation of model-based RL, as discussed in [2]: the performance is constrained by modeling errors or model mismatch. Meanwhile, safety-critical autonomous systems, such as autonomous vehicles, usually have system-environment interaction dynamics. The environmental factors, such as road friction force and latency, are often hard to model or measure, especially in dynamic and unforeseen driving environments. Therefore, relying on a linear model for state prediction (in model-based RL) may lead to sub-optimal performance and safety violations potentially. Motivation by the concerns, our proposed Phy-DRL adopts a linear model only for delivering the residual action policy, automatic construction of safety-embedded reward, and mathematically-provable safety guarantee.
>
> In addition. we note the residual action policy in Phy-DRL can well address the challenge induced by a model mismatch between the linear model and the real nonlinear dynamics model. Specifically, in our residual policy,  the model-based action policy can guide the exploration of DRL agents during training. Meanwhile, the DRL policy learns to effectively deal with uncertainties and compensate for the model mismatch of the model-based action policy.
>
> Because our Phy-DRL does not adopt a model for state prediction. To have a fair comparison, our baseline models of DRL also do have the inside model for state prediction, i.e., they are model-free. For this reason, our previous experiments did not include comparisons with model-based DRL.  Following the reviewer's suggestion, the updated paper includes comparisons with model-based DRL. They are presented in Appendix K.7 of the updated paper (also available at [link 1 (anonymous hosting and browsing)](https://www.dropbox.com/scl/fi/yi1u5ui3jlpayf9f230ys/commbrl.pdf?rlkey=fq9ttlu0fzasy4d46kqq2jhd5&dl=0)).
>
> $$$$
> [1]  Moerland, T. M., Broekens, J., Plaat, A., \& Jonker, C. M. (2023). Model-based reinforcement learning: A survey. Foundations and Trends® in Machine Learning, 16(1), 1-118.
>
> [2]  Janner, M., Fu, J., Zhang, M., \& Levine, S. (2019). When to trust your model: Model-based policy optimization. Advances in neural information processing systems, 32.

---

> ### Author Response · Authors · 2023-11-16
> **Response to Reviewer BXs3: Part II**
>
> **Q1: How does NN input augmentation help embed what physics knowledge into AC?**
>
> $\textbf{Answer:}$ NN Input Augmentation, i.e., Algorithm 1, does not directly contribute to embedding physics knowledge into AC. Its function is to augment each NN input such that the augmented input vector can embrace all the non-missing and non-redundant monomials of a Taylor series (given the order of series). One motivation behind the augmentation is that according to Taylor's Theorem, the Taylor series can approximate arbitrary nonlinear functions, with controllable accuracy via controlling series order. This augmentation thus empowers our DNN with the controllable accuracy for approximating action-value function and action policy. The second motivation is our proposed safety-embedded reward $r(s(k), a(k))$ is a typical Taylor series; the action-value and action policy are functions of it. If using the Taylor series to approximate the action-value function and the action policy, we can also discover some hidden knowledge.
>
> $$$$
>
>  **Q2: how does NN editing help embed what knowledge into AC?**
>
> $\textbf{Answer:}$ We would like to use a toy example to answer the question. Assume ground-truth action policy can be described by
>
> $\pi\left(s\right) = [0,{{w_1}},{{w_2}},0,{{w_3}},{{w_4}},0,{{w_5}},0,0] \cdot \underbrace{[
> 1,v,w,\zeta,{v^2},vw,v\zeta,{w^2},w\zeta,{\zeta ^2}]^{\top}}_{ = {\mathfrak{m}}(s)} + p(s) , ~~~(\text{e}1)$
>
> where ${p}(s)$ is unknown model mismatch, and ${w_1}, \ldots, w_{15}$ are learning weights. The NN editing, including link editing and action editing, will embed the knowledge included in system matrix in (e1), with additional given knowledge that  ``$p(s)$ does not have monomials included in ${\mathfrak{m}}(s)$ in (e1)", to actor network inside. Specifically, for the end-to-end DNN mapping (denoted by $y = \widehat{\pi}\left(s \right)$), referring to (1), the link editing i) removes all link connections with node representations  $1$, $\zeta$, $v\zeta$, $w\zeta$, $\zeta^2$, and ii) maintains link connections with $v$, $w$, $v^2$, $vw$ and  $w^2$. Meanwhile, the action editing guarantees the usages of action functions in all cascade layers do not introduce monomials of $\zeta$ to the mapping $y = \widehat{\pi}\left(s \right)$, because the ground-truth model does not depend on $\zeta$.
>
> $$$$
>
>  **Q3: How do algorithms (1)(2) achieve NN input aug and NN editing?**
>
> $\textbf{Answer:}$ For Algorithm 1 for achieving NN input augmentation, we have added more explanations and motivation to the updated paper. To better understand how Algorithm 1 achieves this and also how it can embrace all the non-missing and non-redundant monomials of a Taylor series, the updated paper includes an illustration example shown in Figure 8 in Appendix F (also available at [link 2 (anonymous hosting and browsing)](https://www.dropbox.com/scl/fi/9qkj7widgadx0q23gxqvv/fnb.pdf?rlkey=zgr6j2i3u7r6n1mw52w2xx46s&dl=0)).

---

> ### Author Response · Authors · 2023-11-16
> **Response to Reviewer BXs3: Part II (Continued)**
>
> To show how Algorithm 2 achieves NN editing, the updated paper includes a flowchart of NN editing in a single PhyN layer, which is shown in Figure 9 in Appendix I.1 (also available at [link 3 (anonymous hosting and browsing)](https://www.dropbox.com/scl/fi/f273ddb1c2nyam5dkkbfs/flowchart.pdf?rlkey=xb32x3t2cdfk5fb8og0a80kyf&dl=0)). Specifically, given the node-representation vector from Algorithm 1, the raw weight matrix obtained via the gradient descent algorithm is edited via link editing to embed assigned physics knowledge, resulting in an edited matrix: $W_t  \leftarrow K_t + U_t = K_t + M_t  \odot W_t$.  We note here the ${K}_t$,  generated in Line 8 and Line 14 of Algorithm 2, includes all the parameters in the knowledge set. While the weight-masking matrix ${M}_t$, generated in Line 9 and Line 15, is used to generate uncertainty matrix ${U}_t$ to include all the parameters excluded from the knowledge set. This is achieved by freezing the parameters of raw ${W}_t$ (from gradient descent algorithm) included in the knowledge set to zeros, through $M_t  \odot W_t$.
>
> Further, Figure 10 in Appendix I.2 (also available at [link 4 (anonymous hosting and browsing)](https://www.dropbox.com/scl/fi/fwlevg52xhqfate7hkz40/asdexp.pdf?rlkey=bedg1yfjim9epu5zofrz6dxrv&dl=0)) depicts the flowchart of NN editing in cascade PhyNs. As an example, we consider actor network, whose ground-truth model is
>
> $\pi\left(s\right) = A_\pi \cdot  {\mathfrak{m}}(s, r_\pi) + p(s) . ~~~(\text{e}2)$
>
>
> We note Lines 5-9 of Algorithm 2 mean that  $A_\pi = K_1 + M_1 \odot A_\pi$, leveraging which and the letting $r_1 = r_\pi$, $x = s$, and $y = \pi\left({s}\right)$, the ground-truth model (e2)  can be rewritten as
>
> $y =  (K_1 + M_1 \odot A_\pi) \cdot \mathfrak{m}(s,r_\pi) + {p}(s) =  K_1 \cdot \mathfrak{m}(x, r_\pi) + (M_1 \odot A_\pi) \cdot \mathfrak{m}(s,r_1) + {p}(s).  ~~~(\text{e}3)$
>
>
> We obtain from Line 19 of Algorithm 2 that the output of the first PhyN layer is
>
> $y_1=  K_1  \cdot \mathfrak{m}(s, r_1) +  a_1 \odot \text{act}(U_1 \cdot {\mathfrak{m}}(s,r_1)). ~~~(\text{e}4)$
>
> Recalling that $K_1$ includes all the entries of $A_\pi$ in knowledge set, while the $U_1$ includes remainders, we conclude from (e3) and (e4) that the available physics knowledge pertaining to the ground-truth model has been embedded to the first PhyN layer.
> As Figure 10 in Appendix I.2 (also available at [link 4 (anonymous hosting and browsing)](https://www.dropbox.com/scl/fi/fwlevg52xhqfate7hkz40/asdexp.pdf?rlkey=bedg1yfjim9epu5zofrz6dxrv&dl=0)) shows the knowledge embedded in the first layer shall be passed down to the remaining cascade PhyNs and preserved therein, such that the end-to-end critic and actor network can strictly comply with the physics knowledge. This knowledge passing is achieved by the block matrix $K_p$ generated in Line 14 of Algorithm 2, thanks to which, the output of $t$-th PhyN layer satisfies
>
> $[y]_{1:\text{len}(y)}$
>
> = $K_1 \cdot \mathfrak{m}(s, r_1) + [a_t \odot \text{act}(U_t \cdot \mathfrak{m}(\bar{y},  r_t ))]_{1:\text{len}(y)},  ~~\forall t \in \{2,\ldots,p\})$
>
> where $\bar{y} = y_{t-1}$. Meanwhile, the $U_t = M_t \odot W_t$ means the masking matrix $M_t$ generated in Line 15 of Algorithm 2 is to remove the spurious correlations in the cascade PhyNs, which is depicted by the cutting link operation in Figure 10 in Appendix I.2  (also available at [link 4 (anonymous hosting and browsing)](https://www.dropbox.com/scl/fi/fwlevg52xhqfate7hkz40/asdexp.pdf?rlkey=bedg1yfjim9epu5zofrz6dxrv&dl=0)).

---

> > ### Comment · Reviewer_BXs3 · 2023-11-16
> >
> > I appreciate the responses from the authors. The section 6 is now much better than the original submitted version. With the toy example added, it is now easy to follow and understand how input augmentation and NN editing contribute to physics knowledge embedding and how they are achieved by Algorithms 1 and 2.
> >
> > I will go through the paper again later and am willing to increase my score to an 8 if no further questions are found.

---

> > > ### Author Response · Authors · 2023-11-18
> > > **Reply to Reviewer BXs3**
> > >
> > > Dear Reviewer BXs3,
> > >
> > > We have updated the paper. The current version is ready for your review. We look forward to your new feedback.
> > >
> > > Sincerely,
> > >
> > > Authors of paper.

---

> > > > ### Comment · Reviewer_BXs3 · 2023-11-20
> > > >
> > > > raise my score to 8. Thanks.

---

> > > > > ### Author Response · Authors · 2023-11-21
> > > > > **Thank you**
> > > > >
> > > > > Dear Reviewer,
> > > > >
> > > > > Thanks a lot for your precious time in reviewing our work and insightful comments. We would like to thank you for your recognition.
> > > > >
> > > > > Best regards,
> > > > >
> > > > > Authors

---

> ### Author Response · Authors · 2023-11-16
> **Reply to Reviewer BXs3**
>
> Dear Reviewer BXs3,
>
> We appreciate your quick reply and positive feedback. The current updated paper is not our final version. We are adding more experiments and explanations to the paper. We will let you know when the final version is ready.
>
> Sincerely,
>
> Authors

---

### Official Review · Reviewer_4Lie · 2023-11-01

**Soundness:** 2 fair
**Presentation:** 1 poor
**Contribution:** 2 fair
**Rating:** 8
**Confidence:** 2

**Summary:**

In this paper, the authors introduce the "Phy-DRL" framework which integrates physics knowledge into deep reinforcement learning (DRL) for safety-critical autonomous systems. Key features of Phy-DRL include:

An action policy that combines both data-driven learning and physics-based modeling.
A reward system embedded with safety considerations.
A neural network (NN) which is guided by physics modeling.
Theoretical benefits of Phy-DRL are that it offers mathematically provable safety guarantees and aligns well with physics knowledge for computing action values. The authors tested their framework on a cart-pole system and a quadruped robot, finding that Phy-DRL offers safety benefits over purely data-driven DRL while requiring fewer learning parameters and enabling stable training.

**Strengths:**

Addressing Safety in DRL: The paper introduces a DRL framework specifically designed for safety-critical autonomous systems, highlighting the significance of adhering to physical laws in AI applications.

Mathematically-Provable Safety Guarantee: One of the notable features of the Phy-DRL is its mathematically provable safety guarantee, which is crucial for real-world applications where safety is paramount. (although I am taking the claim of authors at face value and couldn't do a thorough analysis of the mathematical proofs and theorems given in the paper)

**Weaknesses:**

The thing I feel which is missing is where is the definition of invariant embeddings.
What are they invariant to? How is adding a residual to a model-based making it invariant (and again invariant to what? state space?) Where is the invariant embedding principles coming from? (please cite any papers)

The lack of real-world results is also concerning, given the efforts by the community to test robots in the real world.

The presentation also needs to be improved. Specifically, Important terms and at least some experiment related information should be added in introduction to make the paper and its application easier to understand.

**Questions:**

Please clarify what is meant by invariant embeddings here.

How does Phy-DRL differ from existing DRL frameworks in terms of its integration with physics knowledge? Is this a completely new framework, or an extension of existing ones?

Could you provide more insight into the "residual action policy"? How does it ensure invariance with respect to training?

---

> ### Author Response · Authors · 2023-11-17
> **Response to Reviewer 4Lie: Part I**
>
> We thank Reviewer 4Lie for the positive feedback and insightful comments. The detailed responses are provided below.
>
> $$$$
>
>  **W1: The thing I feel which is missing is where is the definition of invariant embeddings. What are they invariant to? How is adding a residual to a model-based making it invariant (and again invariant to what? state space?) Where is the invariant embedding principles coming from? (please cite any papers).**
>
> $\textbf{Answer:}$ The invariant embedding is our proposed concept. In this paper, an `invariant' refers to a prior policy, prior knowledge, or a designed property, that is independent of agent training.
>
> The residual action policy is one invariant-embedding design.  In Phy-DRL,  the terminal action $a(k)$ has the residual form:
> $a(k) = a_{\text{drl}}(k) + a_{\text{phy}}(k)$, where the $a_{\text{drl}}(k)$ denotes a date-driven action from DRL. The $a_{\text{phy}}(k)$ is a model-based action computed according to  $a_{\text{phy}}(k) = F \cdot s(k)$, where the computation of $F$ is based on the model knowledge $(A, B)$. The model-based action policy is invariant because the $F$ depends on model knowledge $(A, B)$ (obtained offline) only, completely independent of agent training.
>
> The `invariant embedding' in this paper refers to embedding a prior policy, prior knowledge, or a designed property into DRL. These embedded things are already invariant and known, so the agent in training does not have the redundant mission of making these things invariant.
>
> The word 'invariant embedding principles' is confusing. In the updated paper, it has been updated with `invariant-embedding designs.'
>
> $$$$
>
>  **W2: The lack of real-world results is also concerning, given the efforts by the community to test robots in the real world.**
>
> $\textbf{Answer:}$ For the experiments of a quadruped robot, we have the corresponding real (R\&D) A1 Explorer quadruped robot from Unitree. In our project, we train the robot in a simulator and then deploy the $\underline{trained}$ policy to the real robot. We cannot directly perform training on real robots. Otherwise, the robot can be easily damaged (e.g., the robot falls and the motor burns). Therefore, all the experimental results related to training (such as our claimed fast training towards safety guarantee) are obtained in the simulator. The simulator we are using is built on Pybullet, and released on [GitHub](https://github.com/yxyang/locomotion_simulation). The simulator package includes an interface for direct sim-to-real transfer and has a strong record of deploying a trained policy to the same version of the A1 Explorer quadruped robot with us.
>
> The experiments about the trained policies (for validating our derived safety guarantee) also need to be performed in the simulator. The critical reason is the sim-to-real gap. If a violation of safety occurs in a real robot, it can be caused by the sim-to-real gap, rather than our proposed Phy-DRL framework.
>
>  **Beyond Scope of This Paper: addressing the sim-to-real gap via safe continued learning.** To thoroughly demonstrate Phy-DRL in the real robot, our first step is to address the sim-to-real gap. To achieve this, we propose to develop a safe continued learning framework, which is built on neural Simplex [1]. In this framework, we will deploy the well-trained package (rather than a trained policy) from the simulator to the real robot and continue the training there. The neural Simplex will work as a fault-tolerant software architecture.  If a safety violation occurs during the continued training, the Simplex triggers the switching to the high-assurance mode (a protection strategy) for temporarily suspending training. The Simplex will trigger switching to continued training when safety violations or concerns disappear. The continued training will stop if our derived condition of safety guarantee is satisfied in the real robot.  The development of a safe continued learning framework for addressing the sim-to-real gap is beyond the scope of the paper.
>
> [1] Phan, D. T., Grosu, R., Jansen, N., Paoletti, N., Smolka, S. A., \& Stoller, S. D. (2020). Neural simplex architecture. In NASA Formal Methods: 12th International Symposium, NFM 2020, Moffett Field, CA, USA, May 11–15, 2020, Proceedings 12 (pp. 97-114).
>
> $$$$
>
>  **W3: The presentation also needs to be improved. Specifically, Important terms and at least some experiment related information should be added in introduction to make the paper and its application easier to understand.**
>
> $\textbf{Answer:}$ We have added an overview, explanations, and motivations to the updated paper. Due to the page limit, some experimental results are included in the appendices. The updated paper has highlighted connection links with them.

---

> ### Author Response · Authors · 2023-11-17
> **Response to Reviewer 4Lie: Part II**
>
> **Q1: Please clarify what is meant by invariant embeddings here.**
>
> $\textbf{Answer:}$ In this paper, an 'invariant' refers to a prior policy, prior knowledge, or a designed property, that is independent of agent training. `invariant embedding' refers to embedding such an invariant into DRL. For example, integrating a data-driven action policy with a model-based action policy yields our residual action policy. The model-based action policy is $a_{\text{phy}}(k) = F \cdot s(k)$, where $F$ completely depends on model knowledge $(A, B)$ (obtained offline), i.e., it is independent of agent training. Therefore, the residual action policy belongs to an invariant-embedding design. In addition, our proposed safety-embedded reward has an always-holding property: $\mathbf{P} -  \overline{\mathbf{A}}^\top \cdot \mathbf{P} \cdot \overline{\mathbf{A}} \succ 0$ (independent of agent training). So, our proposed safety-embedded reward also belongs to invariant-embedding design. We have added the clarification to the updated paper.
>
> $$$$
>
>  **Q2: How does Phy-DRL differ from existing DRL frameworks in terms of its integration with physics knowledge? Is this a completely new framework, or an extension of existing ones?**
>
> $\textbf{Answer:}$ Our proposed Phy-DRL is built on a deep deterministic policy gradient algorithm [2]. Phy-DRL has very distinguished designs, which are very different from the existing model-free and model-based DRLs.  Differentiating from model-based RLs that adopt a model for state prediction, our Phy-DRL uses a model for 1) the automatic construction of a safety-embedded reward for guiding the exploration of action policy, with an aim of safety guarantee during training, and 2) the automatic computation of model-based action policy, which also guides the exploration of DRL agents during training. Because of the concurrent safety-embedded reward and residual action policy, Phy-DRL features mathematically-provable safety guarantee and fast training towards a safety guarantee.
>
> Another distinguished design is our proposed physics-knowledge-enhanced DNN for DRL's critic and actor networks. The new networks have two innovations in neural architecture: i) Neural Network (NN) Input Augmentation (empowering controllable model accuracy through embedding Taylor series), and ii) Physics-Model-Guided NN Editing (for embedding available physics knowledge about action-value function and action policy into critic and actor networks). Because of this design, Phy-DRL's critic and actor networks feature remarkably fewer learning parameters, while offering enhanced model accuracy and strict compliance with available knowledge.
>
> [2] Achiam, J., \& Morales, M. (2020). Deep deterministic policy gradient.
>
> $$$$
>
>  **Q3: Could you provide more insight into the "residual action policy"? How does it ensure invariance with respect to training?**
>
> $\textbf{Answer:}$ In Phy-DRL,  the terminal action $a(k)$ has the residual form: $a(k) = a_{\text{drl}}(k) + a_{\text{phy}}(k)$, where the $a_{\text{drl}}(k)$ denotes a date-driven action from DRL, while the $a_{\text{phy}}(k)$ is a model-based action computed according to $a_{\text{phy}}(k) = F \cdot s(k)$, where the computation of $F$ is based on the model knowledge $(A, B)$.
>
> The keyword ``residual" stems from Phy-DRL's working mechanism: the model-based action policy can guide the exploration of DRL agents during training. Meanwhile, the agent training is to learn or search for a data-driven policy to effectively deal with uncertainties and compensate for the model mismatch existing in the model-based action policy. The model mismatch exists because the model-based action policy is based on model knowledge $(A, B)$ (representing a linear model), while the dynamics model of real systems is nonlinear.
>
>
> The keyword ``invariant" in residual policy stems from model-based action policy $a_{\text{phy}}(k) = F \cdot s(k)$. The $F$ completely depends on model knowledge $(A, B)$ (obtained offline), independent of agent training. In other words, model-based action policy is a prior design, not influenced by training.

---

> > ### Author Response · Authors · 2023-11-19
> > **Discussion deadline approaching**
> >
> > Dear Reviewer 4Lie,
> >
> > The discussion period is coming to an end, and we hope that we have addressed all of your comments. Could you take a look at our responses and updated paper, and let us know if you have any follow-up questions? We will be happy to answer them.
> >
> > Kind regards,
> >
> > the Authors

---

> > > ### Comment · Reviewer_4Lie · 2023-11-20
> > > **Thanks for the comprehensive response.**
> > >
> > > Hi Authors,
> > > Thanks for the comprehensive response. I have increased my rating to an 8.

---

> ### Author Response · Authors · 2023-11-21
> **Thank you**
>
> Dear Reviewer,
>
> Thanks a lot for your precious time in helping us improve our paper. We would like to thank you for your recognition.
>
> Best regards,
>
> Authors

---

### Official Review · Reviewer_GsDG · 2023-11-01

**Soundness:** 2 fair
**Presentation:** 2 fair
**Contribution:** 3 good
**Rating:** 6
**Confidence:** 3

**Summary:**

Purely data driven learning has a pitfall of tending to violate known conditions of the environment. To address this, prior work has looked at modeling the known transition dynamics of the problem setting and using this to regularize the behavior of the data-driven model. This paper introduces a new technique (Phy-DRL) to improve safety assurance in Reinforcement Learning (RL) systems by means of invariant embeddings tackling three separate issues : using physics knowledge in learning, using safety violation information, and action policies violating known physics laws. Experimental results show the effectiveness of the approach in utilizing existing knowledge in both Cart-Pole and a complex quadrupedal robot.

**Strengths:**

- Considers safety information and known system dynamics in the final policy.
- Introduces novel scheme for neural network policy and action-value function editing using the known dynamics of the environment.

**Weaknesses:**

- The paper addresses the case of RL with partially known environment dynamics but does not adequately consider comparative approaches such as  model-based RL  [1, 2]  or ODE-regularized RL [3]  in all the experiments.
- The effect of each of the invariant embeddings is not shown empirically. An included ablation analysis would be useful to determine this.
- Presentation could be improved slightly, perhaps by including some pseudo code in the Appendix.

[1]  End-to-End Safe Reinforcement Learning through Barrier Functions for Safety-Critical Continuous Control Tasks, Cheng et al., AAAI 2019

[2] Lyapunov Design for Robust and Efficient Robotic Reinforcement Learning, Westenbroek et al., CoRL 2022

[3] Model-based Reinforcement Learning for Semi-Markov Decision Processes with Neural ODEs, Du et al., NeuRIPS 2020

**Questions:**

1. Fig. 4 does not have comparisons with any modern methods for MBRL or physics based RL. Is there a reason why this was omitted?
2. Is ODE-based regularization [3] significantly more restrictive than the proposed approach? Why is a direct comparison not considered?

---

> ### Author Response · Authors · 2023-11-17
> **Response to Reviewer GsDG: Part I**
>
> We thank Reviewer GsDG for the positive feedback and insightful comments. Our answers to the questions will address the concerns. Let us know if our response clarifies the unclear points.
>
> $$$$
>
>
> **W1: The paper addresses the case of RL with partially known environment dynamics but does not adequately consider comparative approaches such as model-based RL [1, 2] or ODE-regularized RL [3] in all the experiments.**
>
> $\textbf{Answer:}$ To address the comment, we would like to clarify the conceptual distinctions between models in our Phy-DRL and standard model-based RL (MBRL). In MBRL, the term ''model" typically denotes the ''world model." Throughout policy optimization or training, the agent engages with this world model, whether known or learned. The agent's performance is intricately tied to the accuracy of this model, as emphasized in [r1]. Conventional practice entails updating the world model using data derived from the true Markov Decision Process (MDP) to enhance policy optimization towards optimality. The model in MBRL is typically used for state prediction. In our proposed Phy-DRL, we introduce an alternative approach to using an MDP model. The model in Phy-DRL is only used for the automatic construction of safety-embedded reward and the computing of the model-based action policy for delivering residual action policy.  In other words, our Phy-DRL does not adopt the model for state prediction. In addition, the residual action policy in Phy-DRL can address the challenge induced by a model mismatch between the model and the real nonlinear dynamics model. Specifically, the model-based action policy can guide the exploration of DRL agents during training. Meanwhile, the training stage is to learn a data-driven policy to effectively deal with uncertainties and compensate for the model mismatch of the model-based action policy.
>
> Appendix K.7 of the updated paper includes new experimental results of Phy-DRL with and without a model for state prediction, and DRL with and without a model for state prediction. For convenience, the experimental results are also available at [link 1 (anonymous hosting and browsing)](https://www.dropbox.com/scl/fi/yi1u5ui3jlpayf9f230ys/commbrl.pdf?rlkey=fq9ttlu0fzasy4d46kqq2jhd5&dl=0).
>
> We now summarize the contributions of the suggested works of literature and compare them with our work.
>
>
>  **Comparisons with [3].**  The work in [3] concentrates on constructing a precise world model for policy optimization with minimal data and optimizing the time schedule to reduce interaction rates. Our approach is different as it requires only a linear model for constructing the residual policy, and we directly train the agent within the true MDP. Using linear model knowledge and given many safety conditions, our Phy-DRL can automatically and simultaneously construct the safety-embedded reward and the model-based action policy in our residual diagram.
>
>  **Comparisons with [1].** The work in [1] proposes an architecture using Control Barrier Function (CBF) controllers as residual policies to ensure safety. It presumes the availability of an invariant safe set, which may be non-trivial to establish in complex systems. Given many (high-dimensional) safety conditions or regulations for formulating a safety set, our Phy-DRL can automatically compute a safety envelope (i.e., a subset of the safety set), which is then used for the automatic construction of (one-dimensional) safety-embedded reward.  Subsequently, we train a Phy-DRL agent to search for a data-driven action policy, which works with a model-based action policy for rendering the safety envelope invariant (also the definition of safety guarantee).
>
> Importantly, our model-based action policy and DRL action policy share the common objective of achieving a safety guarantee. In comparison, the framework proposed in [1] faces a potential conflict, as the CBF controller prioritizes safety while the DRL controller aims to drive the system to the target state, necessitating careful parameter tuning to balance the two controllers.
>
> Besides, the reward of RL guides the exploration of action policy during training, a safety-embedded reward is crucial for DRL to search for policies with a safety guarantee. To handle this, our Phy-DRL first allows us to simplify the system dynamics model to be an analyzable and verifiable linear one. The linear model knowledge is then used to automatically construct concurrent safety-embedded rewards and residual action policies for fast training towards safety guarantee.

---

> ### Author Response · Authors · 2023-11-17
> **Response to Reviewer GsDG: Part I (Continued)**
>
> **Comparisons with [2].** The work in [2] is seminal since it discovered that if DRL’s reward is a control Lyapunov function (CLF), a well-trained DRL can exhibit a mathematically-provable stability guarantee. Intuitively, if a CLF reward is constructed with consideration of safety conditions, a well-trained DRL with such a CLF reward can exhibit a mathematically-provable safety guarantee. For this reason, our paper presents a systematic comparison with the work in [2]. In the updated paper, all the pure DRL models have a CLF reward.
>
> We also perform analysis and experimental comparisons on our safety-embedded reward and CLF reward, which are presented in Appendix L.7 (analysis report is also available at [link 2 (anonymous hosting and browsing)](https://www.dropbox.com/scl/fi/fr2g3rgihj97q65lnl6df/rewarddiff.pdf?rlkey=ixi90w13mge7meiqmhdx6m61a&dl=0)).  In summary, the difference is our reward decouples known invariant and unknown for learning (i.e., the data-driven component only learns the unknown), while the CLF reward mixes known invariant and unknown (i.e., the data-driven part component learns both unknown and known invariant). Decoupling known invariant and unknown for learning can be a critical reason our safety-embedded reward leads to fast training, compared with CLF reward.
>
> $$$$
>
> [1] End-to-End Safe Reinforcement Learning through Barrier Functions for Safety-Critical Continuous Control Tasks, Cheng et al., AAAI 2019.
>
> [2] Lyapunov Design for Robust and Efficient Robotic Reinforcement Learning, Westenbroek et al., CoRL 2022.
>
> [3] Model-based Reinforcement Learning for Semi-Markov Decision Processes with Neural ODEs, Du et al., NeuRIPS 2020.
>
> [r1] Moerland, T. M., Broekens, J., Plaat, A., \& Jonker, C. M. (2023). Model-based reinforcement learning: A survey. Foundations and Trends in Machine Learning, 16(1), 1-118.

---

> ### Author Response · Authors · 2023-11-17
> **Response to Reviewer GsDG: Part II**
>
> **W2: The effect of each of the invariant embeddings is not shown empirically. An included ablation analysis would be useful to determine this.**
>
> $\textbf{Answer:}$ The updated paper has the ablation analysis. Due to the page limit, some parts are included in the Appendices.  To improve the clarity of the paper, we have added information indicating ablation analysis and connecting links to the corresponding content in the Appendices. The critical clarifications are presented below.
>
> **Effect of concurrent residual action policy and safety-embedded reward.**   The experiments on a cart-pole system in Section 7.1 are only to demonstrate the effect of concurrent residual action policy and safety-embedded reward. In other words, the only difference between the DRL models and our Phy-DRL model is that our Phy-DRL has the concurrent residual action policy and safety-embedded reward, while the DRL models have CLF rewards. All their actor and critic networks are implemented as the same multi-layer perceptron, with four fully connected layers. The output dimensions of all critic and actor networks are 256, 128, 64, and 1, respectively.  All the activation functions of the first three neural layers are ReLU, while the output of the last layer is the Tanh function for the actor-network and Linear for the critic network. Such information is included in Appendix K.5 in the updated paper.
>
> **Effect of safety-embedded reward.** The effect of sole safety-embedded reward is demonstrated through analysis and comparing training. However, due to the page limit, the experiment and analysis on the effect of sole safety-embedded reward are presented in Appendix L.7. In the experiment, we consider two Phy-DRL models. Their only difference is the reward; one is our safety-embedded reward, while the other is the CLF reward. All models' other structures and configurations are the same.
>
> **Effect of NN Editing.** The effect of NN Editing is viewed from training. The experiment is in Appendix L.6 of the updated paper. For the considered models, their only difference exists in critic and actor networks. Specifically, one model's critic and actor networks are built on our proposed networks with NN editing. The other model's critic and actor networks are built on the same networks, but they do not have NN editing. All models' other structures and configurations are the same.
>
> $$$$
>
>  **W3: Presentation could be improved slightly, perhaps by including some pseudo code in the Appendix.**
>
> $\textbf{Answer:}$ The updated paper now includes two pseudo codes: Algorithm 1 for NN Input Augmentation and Algorithm 2 for NN Editing. Also, following Review BXs3's suggestions, more detailed explanations and comments have been added to Algorithm 1. However, Algorithm 2 needs a large space for a detailed overview, explanations, and comments. Due to the paper limit, they are included in Appendix I.

---

> ### Author Response · Authors · 2023-11-17
> **Response to Reviewer GsDG: Part III**
>
> **Q1: Fig. 4 does not have comparisons with any modern methods for MBRL or physics based RL. Is there a reason why this was omitted?**
>
> $\textbf{Answer:}$ We would like first to recall that model-based RL involves using a  model (whether learned or known) to predict future states for optimizing policies. Within the model-based RL framework, the model is used primarily to generate states, allowing the agent to query the Markov Decision Process at any desired state-action pair to train the action policy [r2].
>
> We previously did not consider the comparisons with model-based DRL, because of the limitation of model-based RL, as discussed in [r3]: the performance is constrained by modeling errors or model mismatch. Our application systems, such as autonomous vehicles and quadruped robots, have system-environment interaction dynamics. These systems have environmental factors (e.g., road friction force and latency) that are often hard to model or measure, especially in dynamic and unforeseen operating environments. Therefore, relying on a model for only state prediction (in model-based RL) may lead to sub-optimal performance and safety violations potentially. Because of the concerns, in our proposed Phy-DRL, we introduce an alternative approach for using a model: Phy-DRL uses the model only for the automatic constructions of safety-embedded reward and residual action policy. In other words, our Phy-DRL does not adopt the model for state prediction.
>
> In addition, we note the residual action policy in Phy-DRL can well address the challenge induced by a model mismatch between our used model and the real nonlinear dynamics model. Specifically, In our residual policy,  the model-based action policy can guide the exploration of DRL agents during training. Meanwhile, the agent training is to learn a data-driven action policy that can effectively deal with uncertainties and compensate for the model mismatch of the model-based action policy.
>
> Because our Phy-DRL does not adopt a model for state prediction. To have a fair comparison, our baseline models of DRL also do have the inside model for state prediction, i.e., they are model-free. For this reason, our previous experiment section did not include comparisons with model-based DRL.
>
> To better answer the question, the updated paper also includes the comparisons with model-based DRL, which are presented in Appendix K.7 (also available at [link 1 (anonymous hosting and browsing)](https://www.dropbox.com/scl/fi/yi1u5ui3jlpayf9f230ys/commbrl.pdf?rlkey=fq9ttlu0fzasy4d46kqq2jhd5&dl=0)).
>
> [r2]  Moerland, T. M., Broekens, J., Plaat, A., \& Jonker, C. M. (2023). Model-based reinforcement learning: A survey. Foundations and Trends® in Machine Learning, 16(1), 1-118.
>
> [r3]  Janner, M., Fu, J., Zhang, M., \& Levine, S. (2019). When to trust your model: Model-based policy optimization. Advances in neural information processing systems, 32.
>
>
> $$$$
>
>  **Q2: Is ODE-based regularization [3] significantly more restrictive than the proposed approach? Why is a direct comparison not considered?**
>
> $\textbf{Answer:}$ We appreciate the reviewer's insights into the work in [3]. The work utilizes Neural ODEs to model continuous-time dynamics, aiming to enhance modeling accuracy for better policy optimization. The work can significantly address the limitations due to model mismatch between the MDP model (for state prediction) and the real nonlinear dynamics model.  Our focus of Phy-DRL lies in integrating model knowledge of real dynamics into model-free DRL for delivering the automatic constructions of safety-embedded reward and residual action policy.  In this way, Phy-DRL features fast training towards safety guarantee and mathematically-provable safety guarantee. These two approaches thus differ in motivations and design aims.
>
> Our newly added experiments in Appendix K.7 of the updated paper (also available at [link 1 (anonymous hosting and browsing)](https://www.dropbox.com/scl/fi/yi1u5ui3jlpayf9f230ys/commbrl.pdf?rlkey=fq9ttlu0fzasy4d46kqq2jhd5&dl=0)) show that incorporating MDP model into Phy-DRL for state prediction does not improve system performance, viewed from training  and testing of safety guarantee. The reviewer's question is very insightful for developing an ODE-based Phy-DRL framework: Phy-DRL + Neural ODEs (for representing an environment and state predictions).  But there exist a few open problems that will constitute our future investigations. For example, compared with the MDP models, Neural ODEs have super performance, but it has nonlinear activation functions and a large space of learning parameters. So, will having Neural ODEs lower training towards safety guarantee?  Neural ODEs have many latent variables. Are these latent variables sensitive to spurious correlations? If yes, for safety-critical autonomous systems, will incorporating Neural ODEs into Phy-DRL degrade system performance from the perspective of safety guarantee?

---

> > ### Comment · Reviewer_GsDG · 2023-11-20
> > **Thanks for the response**
> >
> > Thanks to the authors for their detailed response.  I appreciate the clarification of the differences of Phy-RL with Model-based RL and its key motivations (combining known linear system dynamics with data-driven systems). I also find the model-based RL comparison experiments to be valuable addition.
> >
> > Regarding the unknown model mismatch, are there any experiments showing how the algorithm performs when there is a large mismatch from the true dynamics? I would expect this to be the case in many complicated environments and such a result would be crucial for the method's general applicability.
> >
> > One final note, the paper as-is is quite dense at the moment, it may be useful to relegate some text to the appendix (maybe some of the algorithm pseudocodes) then spacing text appropriately.

---

> > > ### Author Response · Authors · 2023-11-22
> > > **Final Day**
> > >
> > > Dear Reviewer GsDG,
> > >
> > > We hope that our last reply can answer your new questions for you.  We are on the final day of discussion (deadline: Nov. 22).  We would like to know if you have any further points.  We will be fully committed to discussing more.
> > >
> > >
> > > Kind regards,
> > >
> > > the Authors

---

> ### Author Response · Authors · 2023-11-19
> **Discussion deadline approaching**
>
> Dear Reviewer GsDG,
>
> The discussion period is coming to an end, and we hope that we have addressed all of your comments. Could you take a look at our responses and updated paper, and let us know if you have any follow-up questions? We will be happy to answer them.
>
> Kind regards,
>
> the Authors

---

> ### Author Response · Authors · 2023-11-21
> **Reply to Reviewer GsDG**
>
> Dear Reviewer GsDG,
>
> We appreciate your positive feedback, in-depth questions, and important suggestions.
>
> $$$$
>
> **Q： Regarding the unknown model mismatch, are there any experiments showing how the algorithm performs when there is a large mismatch from the true dynamics? I would expect this to be the case in many complicated environments and such a result would be crucial for the method's general applicability.**
>
> **A：** We would like to answer the questions from two perspectives: 1) experiment and 2) constructions of safety-embedded reward and residual action policy, detailed below.
>
> **1) Experiment.** To convincingly demonstrate the general applicability or robustness of Phy-DRL, we performed the out-of-distribution testing of trained Phy-DRL on a quadruped robot. For the quadruped robot, the linear model is obtained by ignoring friction force and letting raw, yaw, and pitch be zeros. The linear model has a large mismatch since the real robot system is highly nonlinear.  Meanwhile, the robot has system-environment interaction dynamics.  So, out-of-distribution testing environments can induce varying and large model mismatches.
>
> The experimental results are shown in Figure 4 (a)--(d) of the updated paper.   The robot is training in the environment: snow road and velocity-reference command: 0.5 m/s. Figure 4 (a): out-of-distribution testing environment:  snow road and velocity-reference command: 1 m/s. Figure 4 (b): testing environment: the same as the training environment. Figure 4 (c): out-of-distribution testing environment:  wet road and velocity-reference command: -1.4 m/s. Figure 4 (d): out-of-distribution testing environment:  wet road and velocity-reference command: -0.4 m/s. Figure 4 shows that in all testing environments, Phy-DRL successfully
> constraints the robot’s states to a safety set. If given more reasonable velocity commands (Figure 4 (b) and (d)), Phy-DRL can successfully constrain system states to the safety envelope.
>
>
> **2) Constructions of Safety-embedded Reward and Residual Action Policy.** We recall that our proposed safety-embedded reward is  $$r({s}(k), {a}(k)) = s^\top(k) \cdot {\overline{{A}}^\top \cdot {P} \cdot \overline{{A}} } \cdot {s}(k) - {{s}^\top(k+1)} \cdot {P} \cdot {s}(k+1), \text{with } \overline{{A}} \buildrel \Delta \over = {A} + {{B} } \cdot {{F}},$$ and the model-based action policy in residual policy diagram is ${a}_{\text{phy}}(k) = F \cdot s(k)$. We note that $A$ and $B$ are our known linear model knowledge. Then, observing the formulas of safety-embedded reward and model-based action policy, we can discover the constructions of safety-embedded reward and residual action policy equate the computations of matrices $F$ and $P$. To have them, we need first to solve the problem:
>
> ${\text{argmin}_{Q,R}} ~ ( {\log \det \left( {{Q}^{ - 1}} \right)}) \text{ subject to (7) and (11),}~~~~(\text{e}1)$
>
> where Equations (7) and (11) in the paper include the transformed safety conditions, and linear molde knowledge $A$ and $B$. There are multiple toolboxes for solving (e1), such as MATLAB’s LMI and Python’s CVXPY. With the obtained $Q$ and $R$ at hand, $F$ and $P$ are then obtained as  $F = R \cdot Q^{-1}$ and $P = Q^{-1}$.
>
> In summary, our Phy-DRL allows us to simplify the nonlinear system dynamics model to be an analyzable and verifiable linear one for delivering the automatic constructions of concurrent model-based action policy and safety-embedded reward, through solving (e1). However, the model mismatch between the analyzable and verifiable linear model and the real nonlinear dynamics model cannot be arbitrarily large. The upper bound on the model mismatch is hidden in solving (e1). In other words, a very large model mismatch can lead to infeasible solutions of (e1). If (e1) is infeasible, MATLAB’s LMI or Python’s CVXPY will output a warning.  This also means that when we simplify the nonlinear dynamics for an analyzable and verifiable linear model, we must guarantee that the optimal problem (e1) is feasible.
>
> $$$$
>
> **S: One final note, the paper as-is is quite dense at the moment, it may be useful to relegate some text to the appendix (maybe some of the algorithm pseudocodes) then spacing text appropriately.**
>
>
> **A：** We agree with the reviewer the current version is quite dense. This problem can be easily addressed if our paper is finally accepted for publication. Because, in the final version, one paragraph of demonstration links can be updated with just one link. But to be anonymous in the discussion stage, we cannot use the simple link to replace the long paragraph.

---

> ### Comment · Reviewer_GsDG · 2023-11-22
>
> Thanks for addressing my questions. The paper has many good ideas but I am inclined to keep my evaluation for two major reasons:
> - **Model Mismatch with Linear Model not fully qualified**: The experiments on the quadruped robot with the linear model ignoring friction and setting robot angles as zero is promising for the methods real-world applicability. I would have liked a more formal statement on when the model mismatch is too great for the method to work well (in expectation).
> - **Provable safety is mostly empirical and on a limited model of the environment**: To the best of my knowledge, the calculations of a provably-safe trajectory are within the Linear Model of the environment (hence the LMIs). While real-world trajectories that satisfy the safety sub-reward property are `safe' in practice and stay within the safety envelope, this does not mention anything about general deployed policies in the real world (in expectation or otherwise). Without a precise measure of model mismatch (missing from the paper), I fear this guarantee might be of limited utility.

---

> ### Author Response · Authors · 2023-11-22
> **Reply to Reviewer GsDG**
>
> Dear Reviewer,
>
> We appreciate you for the very critical questions. The discussion deadline is approaching, but we would like to share two clarifications for your questions.
>
> $$$$
>
> **Q1: Model Mismatch with Linear Model not fully qualified: The experiments on the quadruped robot with the linear model ignoring friction and setting robot angles as zero is promising for the methods real-world applicability. I would have liked a more formal statement on when the model mismatch is too great for the method to work well (in expectation).**
>
> **Answer:** We will add a remark right after Theorem 5.3 (derived conditions on safety guarantee), formally stating that a large model mismatch our proposed Phy-DRL cannot tolerate if the model mismatch leads to infeasible LMIs. The statement generally holds (not only for our two experiment systems). Here, we do not need to quantify model mismatch; we only need to check if the LMIs are feasible or not, using the available toolbox.
>
> $$$$
>
> **Q2: Provable safety is mostly empirical and on a limited model of the environment: To the best of my knowledge, the calculations of a provably-safe trajectory are within the Linear Model of the environment (hence the LMIs). While real-world trajectories that satisfy the safety sub-reward property are `safe' in practice and stay within the safety envelope, this does not mention anything about general deployed policies in the real world (in expectation or otherwise). Without a precise measure of model mismatch (missing from the paper), I fear this guarantee might be of limited utility.**
>
> **Answer:** The linear model in our Phy-DRL is only leveraged for the automatic constructions of safety-embedded reward and residual action policy. Our derived conditions of safety guarantee -- $r(\mathbf{s}(k), \mathbf{a}(k)) \ge \alpha - 1$ is generic (a sufficient condition for safety guarantee), and is not on a limited model of the environment. Because the $r(\cdot)$ is our defined safety-embedded reward for a real plant (not a linear model), and the $\mathbf{s}(k)$ and $\mathbf{a}(k)$ are real-time sensor outputs (system states and actions) of a real plant. Besides, the derived condition has the following two usages.
>
> **1) Training Stopping.** During training stage, we will output and observe real-time $r(\mathbf{s}(k), \mathbf{a}(k))$. If $r(\mathbf{s}(k), \mathbf{a}(k)) \ge \alpha - 1$ always hold, according to our Theorem 5.3, system safety is guaranteed, training can stop.
>
> **2) Safety Evaluation of Trained Policy.** The condition $r(\mathbf{s}, \mathbf{a}) \ge \alpha - 1$ can be also used for the evaluation of a trained policy deployed to a real plant. Sampling any state $\mathbf{s}$ inside a safety envelope, if $r(\mathbf{s}, \mathbf{a}) \ge \alpha - 1$ always holds, the trained action policy has a safety guarantee according to our safety definition.

---

### Author Response · Authors · 2023-11-18
**An Overview of Major Revision and General Response**

We thank our reviewers for their positive feedback, thoughtful reviews, and insightful comments. So that you know – the paper has been revised accordingly. We first outline the major revisions and then the general responses.

$$$$

## Major Revisions

**Section 3: Design Overview: Invariant Embeddings.** Invariant and invariant embedding are our proposed concepts. We have added the definitions of 'invariant' and  'invariant embedding' to this section.

**Section 6: Invariant Embedding 3: Physics-Knowledge-Enhanced DNN.** We have reorganized the section to include more detailed intuition/motivations, connecting paragraphs, and a design overview. Toy examples have been added for a better understanding of a bunch of symbols, and also the working mechanisms of NN input augmentation and NN editing.

**Appendix D.2: Extension: Explanation of Fast Training Toward Safety Guarantee.** The experimental results demonstrate that compared with DRL, our proposed Phy-DRL features fast training towards safety guarantee. In this newly added appendix, we have leveraged the proof path of Theorem 5.3  to reveal the driving factor of the fast training: concurrent residual action policy and safety-embedded reward.

**Appendix K.7: Testing Comparisons: Model for State Prediction?** This is a newly added appendix, which includes new experimental results of Phy-DRL with and without a model for state prediction, and DRL with and without a model for state prediction. The experiments also include relevant information on sample efficiency and wall-clock time.


$$$$

## General Responses

Compared with model-based and model-free DRL frameworks, our Phy-DRL has two critical differences:


**1) Model Usage.** Phy-DRL allows us to simplify the system dynamics model to be an analyzable and verifiable linear one. This linear model is not used for state prediction. Instead, it delivers the concurrent residual action policy and safety-embedded reward.

**2) Physics-Knowledge-Enhanced DNN.** Phy-DRL's critic and actor networks are built on our proposed physics-knowledge-enhanced DNN. The DNN has two innovations in neural architecture: NN Input Augmentation and  NN Editing.

Because of three invariant-embedding designs (residual action policy, safety-embedded reward, and physics-knowledge-enhanced DNN), Phy-DRL features four distinguished traits:
* Automatic construction of (one-dimensional) safety-embedded reward, given many (high-dimensional) safety conditions or regulations.
* Mathematically-provable safety guarantee.
* Fast training towards safety guarantee, compared with pure (model-based and model-free) DRL.
* Strict compliance with available knowledge about the action-value function and action policy.

---

> ### Comment · Reviewer_BXs3 · 2023-11-20
>
> I personally feel this is a good paper. The readability was a problem in the initial manuscript (this problem was discussed a lot by different reviewers as I checked) but the writing is greatly improved in the updated version. The approach is novel, sound, and very interesting, at least to me.

---

### Meta-Review · Area_Chair_NYTp · 2023-12-24

**Metareview:**

Summary:
The paper a new technique (Phy-DRL) towards safety in Reinforcement Learning (RL) leveraging invariant embeddings tackling three separate issues : using physics knowledge in learning, using safety violation information, and action policies violating known physics laws.

Strength and weaknesses: The paper presents a framework for Mathematically-Provable Safety Guarantee in Deep RL. While important in subject, the presentation is dense and at times confusing. Furthermore much of the details are in an expanded appendix

**Justification For Why Not Higher Score:**

The manuscript lacked in presentation which resulted in hesitations. While this has improved during rebuttal, it remains a concern.

**Justification For Why Not Lower Score:**

All Reviewers are in agreement of the novelty and technical value of the method.

---

### Decision · Program_Chairs · 2024-01-16

Accept (spotlight)